# Learning Optimal Representations with the Decodable Information Bottleneck

**Yann Dubois**
Facebook AI Research
yannd@fb.com

**Douwe Kiela**
Facebook AI Research
dkiela@fb.com

**David J. Schwab**
Facebook AI Research
CUNY Graduate Center
dschwab@fb.com

**Ramakrishna Vedantam**
Facebook AI Research
ramav@fb.com

## Abstract

We address the question of characterizing and finding optimal representations for supervised learning. Traditionally, this question has been tackled using the Information Bottleneck, which compresses the inputs while retaining information about the targets, in a decoder-agnostic fashion. In machine learning, however, our goal is not compression but rather generalization, which is intimately linked to the predictive family or decoder of interest (e.g. linear classifier). We propose the Decodable Information Bottleneck (DIB) that considers information retention and compression from the perspective of the desired predictive family. As a result, DIB gives rise to representations that are optimal in terms of expected test performance and can be estimated with guarantees. Empirically, we show that the framework can be used to enforce a small generalization gap on downstream classifiers and to predict the generalization ability of neural networks.

## 1 Introduction

A fundamental choice in supervised machine learning (ML) centers around the data representation from which to perform predictions. While classical ML uses predefined encodings of the data [1–5] recent progress [6, 7] has been driven by learning such representations. A natural question, then, is what characterizes an "optimal" representation — in terms of generalization — and how to learn it.

The standard framework for studying generalization, statistical learning theory [8], usually assumes a fixed dataset/representation, and aims to restrict the predictive functional family $\mathcal{V}$ (e.g. linear classifiers) such that empirical risk minimizers (ERMs) generalize.[1] Here, we turn the problem on its head: we ask whether it is possible to enforce generalization by changing the representation of the inputs such that ERMs in $\mathcal{V}$ perform well, irrespective of the complexity of $\mathcal{V}$.

A common approach to representation learning consists of jointly training the classifier and representation by minimizing the empirical risk (which we call J-ERM). By only considering empirical risk, J-ERM is optimal in the infinite data limit (consistent; [10]), but the resulting representations do not favor classifiers that will generalize from finite samples. In contrast, the information bottleneck (IB) method [11] aims for representations that have *minimal* information about the inputs to avoid over-fitting, while having *sufficient* information about the labels [12]. While conceptually appealing and used in a range of applications [13–16], IB is based on Shannon's mutual information, which

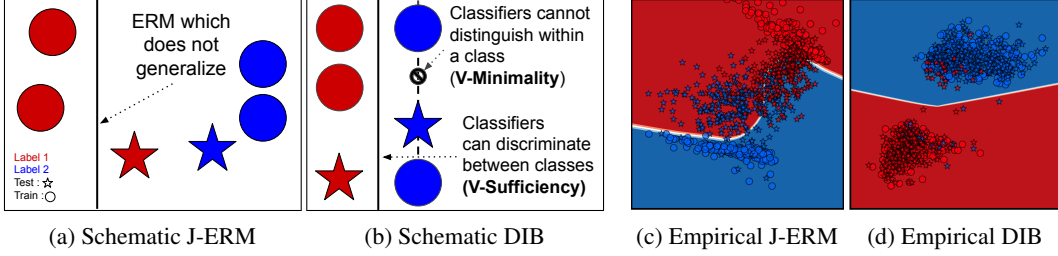

| (a) Schematic J-ERM | (b) Schematic DIB | (c) Empirical J-ERM | (d) Empirical DIB |

Figure 1: **Left two plots**: illustration of representations learned by joint empirical risk minimization (J-ERM) and our decodable information bottleneck (DIB), for classifiers $\mathcal{V}$ with linear vertical decision boundaries. (a) For representations learned by J-ERM, there may exist an ERM that does not generalize; (b) Representations learned by DIB ensure that any ERM will generalize to the test set ($\mathcal{V}$-minimality) . **Right two plots**: 2D representations encoded by an multi-layer perceptron (MLP) for odd-even classification of 200 MNIST [22] examples. The white decision boundary corresponds to a classifer which was trained to perform well on train but bad on test (see Sec. 4.2). (c) J-ERM allows such classifiers that cannot generalize; (d) DIB ensures that there are no such classifiers in $\mathcal{V}$.

was developed for communication theory [17] and does not take into account the predictive family $\mathcal{V}$ of interest. As a result, IB's sufficiency requirement does not ensure the existence of a predictor $f \in \mathcal{V}$ that can perform well using the learned representation; [2] while its minimality term is difficult to estimate, making IB impractical without resorting to approximations [18–21].

We resolve these issues by introducing the *decodable information bottleneck* (DIB) objective, which recovers minimal sufficient representations relative to a predictive family $\mathcal{V}$. Intuitively, it ensures that classifiers in $\mathcal{V}$ can predict labels ($\mathcal{V}$-sufficiency) but cannot distinguish examples with the same label ($\mathcal{V}$-minimality), as illustrated in Fig. 1. Our main contributions can be summarized as follows:

- We generalize notions of minimality and sufficiency to consider predictors $\mathcal{V}$ of interest.
- We prove that such representations are optimal — every downstream ERM in $\mathcal{V}$ reaches the best achievable test performance — and can be learned with guarantees using DIB.
- We experimentally demonstrate that using our representations can increase the performance and robustness of downstream classifiers in average and worst case scenarios.
- We show that the generalization ability of a neural network is highly correlated with the degree of $\mathcal{V}$-minimality of its hidden representations in a wide range of settings (562 models).

## 2 Problem Statement and Background

Throughout this paper, we provide a more informal presentation in the main body, and refer the reader to the appendices for more precise statements. For details about our notation, see Appx. A.

### 2.1 Problem Set-Up: Representation Learning as a Two-Player Game

Consider a game between Alice, who selects a classifier $f \in \mathcal{V}$, and Bob, who provides her with a representation to improve her performance. We are interested in Bob's optimal choice. Specifically:

1. Alice decides *a priori* on: a predictive family $\mathcal{V}$, a task of interest that consists of classifying labels Y given inputs X, and a score/loss function $S$ measuring the quality of her predictions.
2. Given Alice's selections, Bob trains an encoder $P_{Z \mid X}$ to map inputs X to representations Z.
3. Using Bob's encoder and a dataset $\mathcal{D} \overset{\text{i.i.d.}}{\sim} P_{X,Y}^M$ of $M$ input-output pairs $(x, y)$, Alice selects a classifier $\hat{f}$ from all ERMs $\hat{\mathcal{V}}(\mathcal{D}) := \arg\min_{f \in \mathcal{V}} \hat{R}(f, Z; \mathcal{D})$, where $\hat{R}(f, Z; \mathcal{D}) := \frac{1}{M} \sum_{y,x \in \mathcal{D}} E_{z \sim P_{Z \mid x}}[S(y, f[z])]$ is an estimate of the risk $R(f, Z) = E_{\mathcal{D}}\left[\hat{R}(f, Z; \mathcal{D})\right]$.

Our goals are to: (i) characterize *optimal* representations $Z^*$ that minimize Alice's expected loss $R(\hat{f}, Z)$; (ii) derive an objective $\mathcal{L}$ that can be optimized to approximate the optimal encoder $P_{Z^* \mid X}$.

We assume that: (i) sample spaces $\mathcal{Y}, \mathcal{Z}, \mathcal{X}$ are finite; (ii) $S(y, f[z])$ is the log loss $-\log f[z](y)$, where $f[z](y)$ approximates $P_{Y|Z}(y|z)$ as in [23]; (iii) the family $\mathcal{V}$ satisfies mild constraints that hold for practical classifiers such as neural networks. See Appx. B for all assumptions.

## 2.2 Sufficiency, Minimality, and the Information Bottleneck (IB)

We review IB, an information theoretic method for supervised representation learning. IB is built upon the intuition that a representation Z should be maximally informative about Y (sufficient), but contain no additional information about X (minimal) to avoid possible over-fitting. Specifically, the set of **sufficient** representations $\mathcal{S}$ and **minimal sufficient** $\mathcal{M}$ representations are defined as:[3]

$$\mathcal{S} := \arg \max_{Z'} I[Y; Z'] \quad \text{and} \quad \mathcal{M} := \arg \min_{Z' \in \mathcal{S}} I[X; Z'] \tag{1}$$

The IB criterion (to minimize) can then be interpreted [12] as the Lagrangian relaxation of Eq. (1):

$$\mathcal{L}_{IB} := - I[Y; Z] + \beta * I[X; Z] \tag{2}$$

Despite its intuitive appeal, IB suffers from the following theoretical and practical issues: (i) **Lack of optimality** guarantees for $Z \in \mathcal{M}$. Generalization bounds based on $I[Z; X]$ are a step towards such guarantees [12, 26] but current bounds are still vacuous [27]. The strong performance of invertible neural networks [28, 29] also shows that a small $I[X; Z]$ is not required for generalization; (ii) $\mathcal{L}_{IB}$ is **hard to estimate** with finite samples [12, 30]. One has to either restrict the considered setting [11, 31, 32], or optimize variational [18–20] or non-parametric [21] bounds; (iii) $\mathcal{L}_{IB}$ is **invariant to bijections** and thus does not favor simple decision boundaries [33] that can be achieved by a $f \in \mathcal{V}$.

We stipulate that these known issues stem from a common cause: IB uses mutual information, which is agnostic to the predictive family $\mathcal{V}$ of interest. To remedy this, we leverage the recently proposed $\mathcal{V}$-information [23] to formalize the notion of $\mathcal{V}$-minimal $\mathcal{V}$-sufficient representations.

## 2.3 $\mathcal{V}$-information

From a predictive perspective, mutual information $I[Y; Z]$ corresponds to the difference in expected log loss when predicting Y with or without Z using the best possible probabilistic classifier.

$$I[Y; Z] := H[Y] - H[Y|Z] = H[Y] - E_{z,y \sim P_{Z,Y}} \left[ -\log P_{Y \mid Z} \right] \tag{3}$$

$$= H[Y] - \inf_{f \in \mathcal{U}} E_{z,y \sim P_{Z,Y}} [-\log f[z](y)] \quad \text{Strict Properness [34]} \tag{4}$$

where $\mathcal{U}$ is the collection of all predictors from $\mathcal{Z}$ to distributions over $\mathcal{Y}$, which we call *universal*. As the optimization in Eq. (4) is over $\mathcal{U}$, $I[Y; Z]$ measures information that might not be "usable" by $f \in \mathcal{V} \subset \mathcal{U}$. Xu et al.'s [23] resolve this issue by introducing $\mathcal{V}$-information $I_{\mathcal{V}}[Z \to Y]$ to only consider the information that can be decoded by a predictors of interest $f \in \mathcal{V}$ instead of $f \in \mathcal{U}$.[4]

$$I_{\mathcal{V}}[Z \to Y] := H[Y] - H_{\mathcal{V}}[Y \mid Z] = H[Y] - \inf_{f \in \mathcal{V}} E_{z,y \sim P_{Z,Y}} [-\log f[z](y)] \tag{5}$$

$I_{\mathcal{V}}[Z \to Y]$ has useful properties, it: recovers $I[Y; Z]$ for $\mathcal{V} = \mathcal{U}$, is non-negative, and is zero when Z is independent of Y. Importantly, $\mathcal{V}$-information is easier to estimate than Shannon's information; indeed, it corresponds to estimating the risk ($H_{\mathcal{V}}[Y \mid Z]$) and thus inherits [23] probably approximately correct (PAC; [35]) estimation bounds that depend on the (Rademacher [36]) complexity of $\mathcal{V}$.

## 3 Methods

In this section, we define $\mathcal{V}$-minimal $\mathcal{V}$-sufficient representations, prove that they are optimal in the two-player representation learning game, and discuss how to approximately learn them in practice.

### 3.1 $\mathcal{V}$-Sufficiency and Best Achievable Performance

Let us study Alice's best risk $\min_{f \in \mathcal{V}} \mathrm{R}(f, \mathrm{Z})$ using a representation Z. This tight lower bound on her performance looks strikingly similar to $\mathrm{H}_{\mathcal{V}}[\mathrm{Y} \mid \mathrm{Z}]$, which is controlled by $\mathrm{I}_{\mathcal{V}}[\mathrm{Z} \to \mathrm{Y}]$ (see Eq. (5)). As a result, if Bob maximizes $\mathrm{I}_{\mathcal{V}}[\mathrm{Z} \to \mathrm{Y}]$, he will ensure that Alice can achieve the lowest loss.

**Definition 1.** A representation Z is said to be $\mathcal{V}$**-sufficient** if it maximizes $\mathcal{V}$-information with the labels. We denote all such representations as $\mathcal{S}_{\mathcal{V}} := \arg\max_{\mathrm{Z}'} \mathrm{I}_{\mathcal{V}}[\mathrm{Z}' \to \mathrm{Y}]$.

**Proposition 1.** Z is $\mathcal{V}$-sufficient $\iff$ there exists $f^* \in \mathcal{V}$ whose test loss when predicting from Z is the best achievable risk, i.e., $\mathrm{R}(f^*, \mathrm{Z}) = \min_{\mathrm{Z}} \min_{f \in \mathcal{V}} \mathrm{R}(f, \mathrm{Z})$.

Although the previous proposition may seem trivial, it bears important implications, namely that contrary to the sufficiency term of IB one should maximize $\mathrm{I}_{\mathcal{V}}[\mathrm{Z} \to \mathrm{Y}]$ rather than $\mathrm{I}[\mathrm{Y}; \mathrm{Z}]$ when predictors live in a constrained family $\mathcal{V}$. Indeed, ensuring sufficient $\mathrm{I}[\mathrm{Y}; \mathrm{Z}]$ does not mean that there is a classifier $f \in \mathcal{V}$ that can "decode" that information. [5] We prove our claims in Appx. C.2.

### 3.2 $\mathcal{V}$-minimality and Generalization

We have seen that $\mathcal{V}$-sufficiency ensures that Alice *could* achieve the best loss. In this section, we study what representations Bob should chose to guarantee that Alice's ERMs *will* perform optimally by ensuring that any ERM generalizes beyond the training set.

IB suggests minimizing the information $\mathrm{I}[\mathrm{Z}; \mathrm{X}]$ between the representation Z and inputs X to avoid over-fitting. We, instead, argue that only the information that can be decoded by $\mathcal{V}$ matters, and would thus like to minimize $\mathrm{I}_{\mathcal{V}}[\mathrm{Z} \to \mathrm{X}]$. However, the latter is not defined as X does not generally take value in (t.v.i.) $\mathcal{Y}$, the sample space of $\mathcal{V}$'s co-domain. For example, in a $32 \times 32$ image binary classification, $\mathcal{Y} = \{0, 1\}$ but $\mathcal{X} = [0, \dots, 256]^{1024}$ so classifiers $f \in \mathcal{V}$ cannot predict $x \in \mathcal{X}$. To circumvent this, we decompose X into a collection of r.v.s N that t.v.i. $\mathcal{Y}$, so that $\mathrm{I}_{\mathcal{V}}[\mathrm{Z} \to \mathrm{N}]$ is well defined. Specifically, let $\mathrm{X}_y, \mathrm{Z}_y$ be "conditional r.v." s.t. $P_{\mathrm{X}_y} = P_{\mathrm{X}|y}$, $P_{\mathrm{Z}_y} = P_{\mathrm{Z}|y}$, $P_{\mathrm{X}_y, \mathrm{Z}_y} = P_{\mathrm{X}, \mathrm{Z}|y}$. We define the $y$ decomposition of X as r.v.s that arise by all possible labelings of $\mathrm{X}_y$:[6]

$$\mathrm{Dec}(\mathrm{X}, y) := \{\mathrm{N} \mid \exists t' : \mathcal{X} \to \mathcal{Y} \; s.t. \; \mathrm{N} = t'(\mathrm{X}_y)\} \tag{6}$$

We can now define the average $\mathcal{V}$-information between Z and the $y$ decompositions of X as:

$$\mathrm{I}_{\mathcal{V}}[\mathrm{Z} \to \mathrm{Dec}(\mathrm{X}, \mathcal{Y})] := \frac{1}{|\mathcal{Y}|} \sum_{y \in \mathcal{Y}} \frac{1}{|\mathrm{Dec}(\mathrm{X}, y)|} \sum_{\mathrm{N} \in \mathrm{Dec}(\mathrm{X}, y)} \mathrm{I}_{\mathcal{V}}[\mathrm{Z}_y \to \mathrm{N}] \tag{7}$$

$\mathrm{I}_{\mathcal{V}}[\mathrm{Z} \to \mathrm{Dec}(\mathrm{X}, \mathcal{Y})]$ essentially measures how well predictors in $\mathcal{V}$ can predict arbitrary labeling $\mathrm{N} \in \mathrm{Dec}(\mathrm{X}, y)$ of examples with the same underlying label $y$. Replacing the minimality term $\mathrm{I}[\mathrm{X}; \mathrm{Z}]$ by $\mathrm{I}_{\mathcal{V}}[\mathrm{Z} \to \mathrm{Dec}(\mathrm{X}, \mathcal{Y})]$ we get our notion of $\mathcal{V}$-minimal $\mathcal{V}$-sufficient representations.

**Definition 2.** Z is $\mathcal{V}$**-minimal** $\mathcal{V}$-sufficient if it is $\mathcal{V}$-sufficient *and* has minimal average $\mathcal{V}$-information with $y$ decompositions of X. We denote all such Z as $\mathcal{M}_{\mathcal{V}} := \arg\min_{\mathrm{Z} \in \mathcal{S}_{\mathcal{V}}} \mathrm{I}_{\mathcal{V}}[\mathrm{Z} \to \mathrm{Dec}(\mathrm{X}, \mathcal{Y})]$.

Intuitively, a representation is $\mathcal{V}$-minimal if no predictor in $\mathcal{V}$ can assign different predictions to examples with the same label. Consequently, predictors will not be able to distinguish train and test examples and must thus perfectly generalize. In this case, predictors will perform optimally as there is at least one which does ($\mathcal{V}$-sufficiency; Prop. 1). We formalize this intuition in Appx. C.3:

**Theorem 1.** (Informal) Let $\mathcal{V}$ be a predictive family, $\mathcal{D} \overset{\text{i.i.d.}}{\sim} P_{\mathrm{X}, \mathrm{Y}}^M$ a dataset, and assume labels Y are a deterministic function of the inputs $t(\mathrm{X})$. If $\mathrm{Z} \in \mathcal{M}_{\mathcal{V}}$ be $\mathcal{V}$-minimal $\mathcal{V}$-sufficient, then the expected test loss of any ERM $\hat{f} \in \hat{\mathcal{V}}(\mathcal{D})$ is the best achievable risk, i.e., $\mathrm{R}(\hat{f}, \mathrm{Z}) = \min_{\mathrm{Z}} \min_{f \in \mathcal{V}} \mathrm{R}(f, \mathrm{Z})$.

As all ERMs reach the best risk, so does their expectation, i.e., any $\mathrm{Z} \in \mathcal{M}_{\mathcal{V}}$ is optimal. We also show in Appx. C.4 that $\mathcal{V}$-minimality and $\mathcal{V}$-sufficiency satisfy the following properties:

**Function** $H(\mathcal{V}, \mathbf{x}, \mathbf{y}, P_{Z|X})$:

  $\mathbf{z} \leftarrow$ sample once from each $P_{Z|\mathbf{x}}$

  **return** $\inf_{f \in \mathcal{V}} \sum_{z,y \in \mathbf{z},\mathbf{y}} \frac{-\log f[z](y)}{|\mathcal{D}|}$

**Function** $\hat{\mathcal{L}}_{\mathrm{DIB}}(\mathcal{V}, \mathcal{D}, P_{Z|X}, K, \beta)$:

  $(\mathbf{x}, \mathbf{y}), \mathcal{L}_{\mathcal{V}\min}, \mathcal{Y} \leftarrow \mathcal{D}, 0, \mathrm{unique}(\mathbf{y})$

  $\mathcal{L}_{\mathcal{V}\mathrm{suff}} \leftarrow H(\mathcal{V}, \mathbf{x}, \mathbf{y}, P_{Z|X})$

  **for** $y$ *in* $\mathcal{Y}$ **do**

    $\mathbf{x}_y \leftarrow \mathbf{x}[\mathbf{y} == y]$

    **for** $k \leftarrow 1$ **to** $K$ **do**

      $\mathbf{n} \leftarrow \mathrm{random\_choice}(\mathcal{Y}, \mathrm{size}=|\mathbf{x}_y|)$

      $\mathcal{L}_{\mathcal{V}\min} \mathrel{+}= \frac{H(\mathcal{V}, \mathbf{x}_y, \mathbf{n}, P_{Z|X})}{|\mathcal{Y}| * K}$

  **return** $(const) + \mathcal{L}_{\mathcal{V}suff} - \beta * \mathcal{L}_{\mathcal{V}min}$

(a) Pseudo-code for $\hat{\mathcal{L}}_{\mathrm{DIB}}(\mathcal{D})$

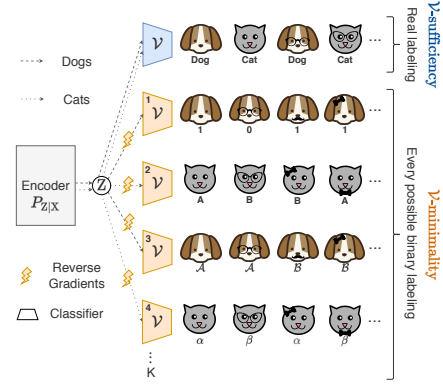

(b) DIB with neural networks

Figure 2: **Practical DIB** (a) Pseudo-code to compute the $\hat{\mathcal{L}}_{\mathrm{DIB}}(\mathcal{D})$; (b) Illustration of DIB to train a neural encoder. $\mathcal{V}$-sufficiency corresponds to the standard log loss. $\mathcal{V}$-minimality heads are trained to classify $K$ arbitrary labeling within each class but the gradients w.r.t. the encoder are reversed so that Z cannot be used for that task. Each head has different parameters but the same architecture $\mathcal{V}$.

**Proposition 2.** Let $\mathcal{V} \subseteq \mathcal{V}^+$ be two families and $\mathcal{U}$ the universal one. If labels are deterministic:

- **Recoverability** The set of $\mathcal{U}$-minimal $\mathcal{U}$-sufficient representations corresponds to the minimal sufficient representations that t.v.i. in the domain of $\mathcal{U}$, i.e., $\mathcal{M}_{\mathcal{U}} = \mathcal{M} \cap \mathcal{Z}$.

- **Monotonicity** $\mathcal{V}^+$-minimal $\mathcal{V}$-sufficient representations are $\mathcal{V}$-minimal $\mathcal{V}$-sufficient, i.e., $\arg\max_{Z \in \mathcal{S}_{\mathcal{V}}} I_{\mathcal{V}^+}[Z \rightarrow \mathrm{Dec}(X, \mathcal{Y})] \subseteq \mathcal{M}_{\mathcal{V}}$.

- **Characterization** $Z \in \mathcal{M}_{\mathcal{V}} \iff Z \in \mathcal{S}_{\mathcal{V}}$ and $I_{\mathcal{V}}[Z \rightarrow \mathrm{Dec}(X, \mathcal{Y})] = 0$.

- **Existence** At least one $\mathcal{U}$-minimal $\mathcal{U}$-sufficient representation always exists, i.e., $|\mathcal{M}_{\mathcal{V}}| > 0$.

The recoverability shows that our notion of $\mathcal{V}$-minimal $\mathcal{V}$-sufficiency is a generalization of minimal sufficiency. As a corollary, IB's representations are optimal when Alice is unconstrained in her choice of predictors $\mathcal{V} = \mathcal{U}$. The monotonicity implies that minimality with respect to (w.r.t.) a larger $\mathcal{V}^+$ is also optimal. Finally, the characterizations property gives a simple way of testing for $\mathcal{V}$-minimality.

### 3.3 Practical Optimization and Estimation

In the previous section we characterized optimal representations $Z^* \in \mathcal{M}_{\mathcal{V}}$. Unfortunately, Bob cannot learn these $Z^*$ as it requires the underlying distribution $P_{X,Y}$. We will now show that he can nevertheless approximate $Z^*$ in a sample- and computationally- efficient manner.

**Optimization**. Learning $Z \in \mathcal{M}_{\mathcal{V}}$ requires solving a constrained optimization problem. Similarly to IB, we minimize the *decodable information bottleneck* (DIB), a Lagrangian relaxation of Def. 2:

$$\mathcal{L}_{\mathrm{DIB}} := -I_{\mathcal{V}}[Z \rightarrow Y] + \beta * I_{\mathcal{V}}[Z \rightarrow \mathrm{Dec}(X, \mathcal{Y})] \qquad (8)$$

Notice that each $I_{\mathcal{V}}[Z \rightarrow \cdot]$ has an internal optimization. In particular $I_{\mathcal{V}}[Z \rightarrow \mathrm{Dec}(X, \mathcal{Y})]$ turns the problem into a min (over Z) - max (over $f \in \mathcal{V}$) optimization, which can be hard to optimize [38, 39]. We empirically compare methods for optimizing $\mathcal{L}_{\mathrm{DIB}}$ in Appx. E.2 and show that joint gradient descent ascent performs well if we ensure that the norm of the learned representation cannot diverge.

**Estimation**. A major benefit of $\mathcal{L}_{\mathrm{DIB}}$ over $\mathcal{L}_{\mathrm{IB}}$ is that it can be estimated with guarantees using finite samples. Namely, if Bob has access to a training set $\mathcal{D} \overset{\text{i.i.d.}}{\sim} P_{X,Y}^M$, he can estimate $\mathcal{L}_{\mathrm{DIB}}$ reasonably well. In practice, we: (i) use $\mathcal{D}$ to estimate all expectations over $P_{X,Y}$; (ii) use samples from Bob's encoder $z \sim P_{Z|x}$; (iii) estimate the average over $N \in \mathrm{Dec}(X, y)$ in Eq. (7) using $K$ samples. Figure 2a shows a (naive) algorithm to compute the resulting estimate $\hat{\mathcal{L}}_{\mathrm{DIB}}(\mathcal{D})$. Despite these approximations, we show in Appx. C.5 that $\mathcal{L}_{\mathrm{DIB}}$ inherits $\mathcal{V}$-information's estimation bounds [23].

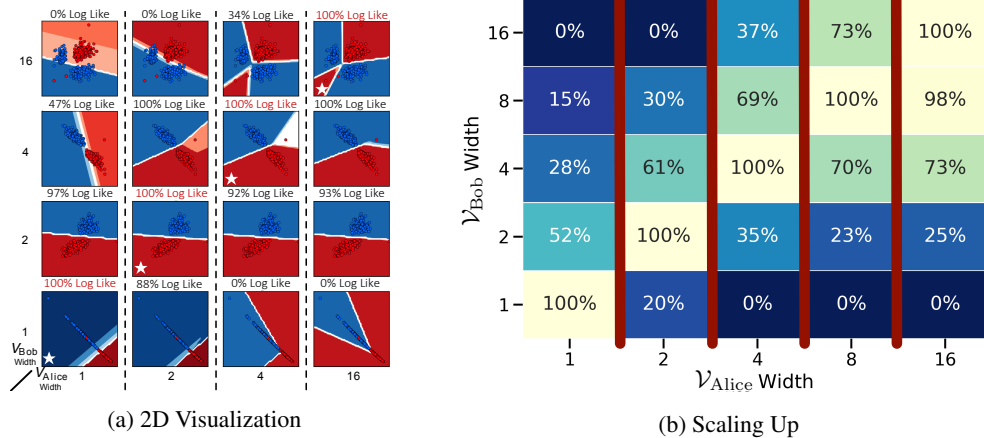

(a) 2D Visualization          (b) Scaling Up

Figure 3: **Optimality of $\mathcal{V}$-sufficiency.** Plots of Alice's best possible performance with different $\mathcal{V}_{Bob}$-sufficient representations. The log likelihood is column-wise scaled from 0 to 100, and vertical separators are present to discourage between-column comparison. The predictive families are MLPs with varying widths. (a) Samples of Bob's 2D representations along with Alice's decision boundaries for an odd-even CIFAR100 binary classification; (b) Same scaled log likelihood but using 8D representations, the standard CIFAR100 dataset, and averaging over 5 runs.

**Proposition 3.** (Informal) Let $\mathfrak{R}_{|\mathcal{D}|}$ denote the $|\mathcal{D}|$ samples Rademacher complexity. Assuming the loss is always bounded $|\log f[x](y)| < C$ then with probability at least $1 - \delta$, $\hat{\mathcal{L}}_{\mathrm{DIB}}(\mathcal{D})$ described in Fig. 2a approximates $\mathcal{L}_{\mathrm{DIB}}$ with error less than $2\mathfrak{R}_{|\mathcal{D}|}(\log \circ \mathcal{V}) + \beta \log|\mathcal{Y}| + C\sqrt{\frac{2\log\frac{1}{\delta}}{M}}$.

The fact that the estimation error in Prop. 3 grows with the (Rademacher) complexity of $\mathcal{V}$, shows that the error is largest for $\mathcal{V} = \mathcal{U}$ corresponding to $\mathcal{L}_{\mathrm{IB}}$. We also see a trade-off in Alice's choice of $\mathcal{V}$. A more complex $\mathcal{V}$ means the estimation of $\hat{\mathcal{L}}_{\mathrm{DIB}}(\mathcal{D})$ is harder for Bob (Prop. 3), but Alice's prediction will improve (smaller $\min_{\mathrm{Z}} \min_{f \in \mathcal{V}} \mathrm{R}(f, \mathrm{Z})$; Theorem 1).

**Case study: neural networks.** Suppose that $\mathcal{V}$ is a specific neural architecture, the encoder $P_{\mathrm{Z}|\mathrm{X}}$ is parametrized by a neural network $q_\theta$, and we are interested in cat-dog classification. As shown in Fig. 2b, training $q_\theta$ with DIB corresponds to fitting $q_\theta$ with multiple classification heads, each having exactly the same architecture $\mathcal{V}$ but different parameters. The $\mathcal{V}$-sufficiency head (in blue) tries to classify cats and dogs. Each of the $K$ (typically 3-4, see Appx. E.4) $\mathcal{V}$-minimality heads (in orange) ensure that the representation cannot be used to classify an arbitrary (fixed) labeling of cats or dogs. In practice, the encoder and heads are trained jointly but gradients from $\mathcal{V}$-minimality heads are reversed. The $\mathcal{V}$-minimality losses are also multiplied by a hyper-parameter $\beta$.

## 4 Experiments

We evaluate our framework in practical settings, focusing on: (i) the relation between $\mathcal{V}$-sufficiency and Alice's best achievable performance; (ii) the relation between $\mathcal{V}$-minimality and generalization; (iii) the consequence of a mismatch between $\mathcal{V}_{Alice}$ and the functional family $\mathcal{V}_{Bob}$ w.r.t. which Z is sufficient or minimal — especially in IB's setting $\mathcal{V}_{Bob} = \mathcal{U}$; (iv) the use of our framework to predict generalization of trained networks. Many of our experiments involve sweeping over the complexity of families $\mathcal{V}^- \subseteq \mathcal{V} \subseteq \mathcal{V}^+$, we do this by varying widths of MLPs — with $\mathcal{V} \to \mathcal{U}$ in the infinite width limit [40, 41]. Alternative ways of sweeping over $\mathcal{V}$ are evaluated in Appx. E.1.

### 4.1 $\mathcal{V}$-sufficiency: Optimal Representations When the Data Distribution is Known

We study optimal representations when Alice has access to the data distribution $P_{\mathrm{Z} \times \mathrm{Y}}$. Alice's risk $\mathrm{R}(f, \mathrm{Z})$ in such setting is important as it is a tight lower bound on her performance in practical settings (see Sec. 3.1). We consider the following setting: Bob trains a ResNet18 encoder [42] by maximizing

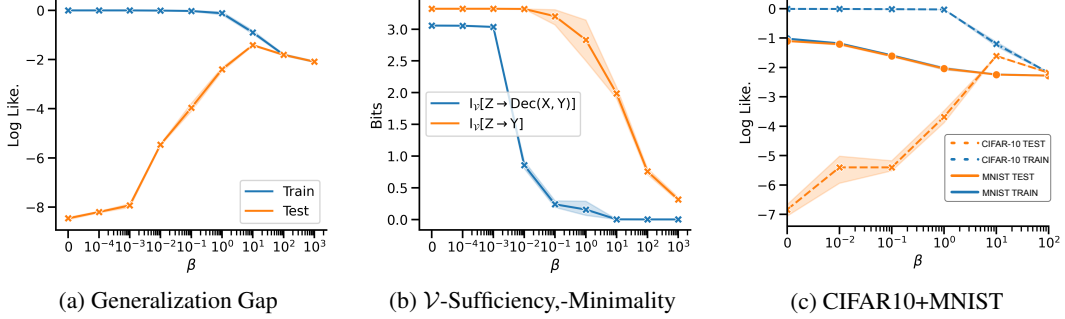

(a) Generalization Gap    (b) $\mathcal{V}$-Sufficiency,-Minimality    (c) CIFAR10+MNIST

Figure 4: **Effect of DIB on generalization** Left two plots (CIFAR10): Impact of DIB's $\beta$ on: (a) the train and test performance of Alice worst ERM; (b) the $\hat{\mathrm{I}}_{\mathcal{V}}[Z \to \mathrm{Dec}(X, \mathcal{Y}); \mathcal{D}]$ and $\hat{\mathrm{I}}_{\mathcal{V}}[Y \to Z; \mathcal{D}]$ of Bob's representation Z. As $\beta$ increases, Z becomes $\mathcal{V}$-minimal which increases the Alice's test performance until Z is far from $\mathcal{V}$-sufficient. Right plot (CIFAR10+MNIST): Same as (a) but images contain overlaid digits as distractors (see Appx. D.2). $\mathcal{V}$-minimality avoids over-fitting by removing spurious MNIST information. The shaded areas indicate $95\%$ bootstrap confidence interval on 5 runs.

$\mathrm{I}_{\mathcal{V}_{\mathrm{Bob}}}[Z \to Y]$, Alice freezes it and trains her own classifier $f \in \mathcal{V}_{Alice}$ using the underlying $P_{Z \times Y}$, i.e., $f$ is trained and evaluated on the *same* dataset. See Appx. D.1 for experimental details.

**Which $\mathcal{V}$-sufficiency should Bob chose?** Proposition 1 tells us that Bob's optimal choice is $\mathcal{V}_{Bob} = \mathcal{V}_{Alice}$. If he opts for a larger family $\mathcal{V}_{Alice} \subseteq \mathcal{V}_{Bob}$, the representation Z is unlikely to be decodable by Alice. If $\mathcal{V}_{Bob} \subseteq \mathcal{V}_{Alice}$, he will unnecessarily constrain Z. We first consider a setting that can be visualized: classifying the parity of CIFAR100 class index [43] using 2D representations. Figure 3a shows samples from Z and Alice's decision boundaries. To highlight the optimal $\mathcal{V}_{Bob}$ for a given $\mathcal{V}_{Alice}$, we scale the performance of each column from 0 to 100 in the figure. As predicted by Prop. 1, the best performance is achieved at $\mathcal{V}_{Alice} = \mathcal{V}_{Bob}$. The worst predictions arise when $\mathcal{V}_{Alice} \subseteq \mathcal{V}_{Bob}$, as the representations cannot be separated by Alice's classifier (e.g. $\mathcal{V}_{Bob}$ width 16 and $\mathcal{V}_{Alice}$ width 1). This suggests that IB's sufficiency (infinite width $\mathcal{V}_{Bob} = \mathcal{U}$) is undesirable when $\mathcal{V}_{Alice}$ is constrained. Figure 3b shows similar results in 8D across 5 runs. See Appx. E.8 for more settings.

## 4.2 $\mathcal{V}$-minimality: Optimal Representations for Generalization

Theorem 1 states that $\mathcal{V}$-minimality ensures all ERMs can generalize well. We investigate whether this is still approximately the case in practical settings, i.e., when Bob optimizes $\hat{\mathcal{L}}_{\mathrm{DIB}}(\mathcal{D})$.

**Experimental Details.** Our claim concerns *all* ERMs $\hat{\mathcal{V}}^*(\mathcal{D})$, which cannot be supported by training a few $\hat{f} \in \hat{\mathcal{V}}^*(\mathcal{D})$. Instead, we evaluate the ERM that performs worst on the test set (Worst ERM), i.e., $\arg \max_{f \in \hat{\mathcal{V}}^*(\mathcal{D})} \mathrm{R}(f, Z)$. We do so by optimizing the following Lagrangian relaxation $\arg \min_{f \in \mathcal{V}} \hat{\mathrm{R}}(f, Z; \mathcal{D}) - \gamma \mathrm{R}(f, Z)$ (see Appx. E.7). As our theory does not impose constraints on Z, we need an encoder close to a universal function approximator. We use a 3-MLP encoder with around 21M parameters and a 1024 dimensional Z. Since we want to investigate the generalization of ERMs resulting from Bob's criterion, we do not use (possibly implicit) regularizers such as large learning rate [44]. For more experimental details see Appx. D.1.

**What is the impact of DIB's $\beta$ ?** We train representations on CIFAR10 with various $\beta$ to investigate the effect of $\hat{\mathrm{I}}_{\mathcal{V}}[Z \to \mathrm{Dec}(X, \mathcal{Y}); \mathcal{D}]$ and $\hat{\mathrm{I}}_{\mathcal{V}}[Y \to Z; \mathcal{D}]$ (Fig. 4b) on Alice's performance (Fig. 4a). Increasing $\beta$ results in a decrease in $\hat{\mathrm{I}}_{\mathcal{V}}[Z \to \mathrm{Dec}(X, \mathcal{Y}); \mathcal{D}]$ which monotonically shrinks the train-test gap. This suggests that, although our theory only applies for $\mathcal{V}$-minimality, $\mathrm{I}_{\mathcal{V}}[Z \to \mathrm{Dec}(X, \mathcal{Y})]$ is tightly linked to generalization even when it is non-zero. After a certain threshold ($\beta = 10$) the generalization gains come at a large cost in $\mathrm{I}_{\mathcal{V}}[Y \to Z]$, which controls the best achievable loss. This shows that a trade-off (controlled by $\beta$) between $\mathcal{V}$-minimality (generalization) and $\mathcal{V}$-sufficient (lower bound) arises when Bob has to estimate $\mathcal{L}_{\mathrm{DIB}}$ using finite samples $\hat{\mathcal{L}}_{\mathrm{DIB}}(\mathcal{D})$.

**$\mathcal{V}$-minimality and robustness to spurious correlations**. We overlay MNIST digits as a distractor on CIFAR10 (see Appx. D.2). We run the same experiments as with CIFAR10, but we additionally train an ERM from $\mathcal{V}$ to predict MNIST labels, i.e., test whether Z contains decodable information

about MNIST. Figure 4c shows that as $\beta$ increases, predicting MNIST becomes harder. Indeed decreasing $\hat{I}_\mathcal{V}[Z \to \mathrm{Dec}(X, \mathcal{Y}); \mathcal{D}]$ removes all $\mathcal{V}$-information in Z which is not useful for predicting Y. As a result, Alice's ERM must generalize better as it cannot over-fit spurious patterns.[7]

Table 1: Alice's worst and average case log loss given different representation schemes used by Bob (lower is better). Standard errors are across 5 runs.

| | No Reg. | Stoch. Rep. | Dropout | Wt. Dec. | VIB | $\mathcal{V}^-$-DIB | $\mathcal{V}^+$-DIB | DIB |
|---|---|---|---|---|---|---|---|---|
| Worst | $10.23 \pm .13$ | $8.61 \pm .05$ | $1.90 \pm .00$ | $10.25 \pm .03$ | $1.82 \pm .02$ | $1.54 \pm .03$ | $1.94 \pm .33$ | $\mathbf{1.41} \pm .01$ |
| Avg. | $4.62 \pm .00$ | $4.34 \pm .04$ | $1.49 \pm .00$ | $4.96 \pm .03$ | $1.76 \pm .01$ | $1.47 \pm .01$ | $1.74 \pm .18$ | $\mathbf{1.38} \pm .01$ |

**Which $\mathcal{V}$-minimality should Bob chose?** We study the effect of $Z \in \mathcal{S}_{\mathcal{V}_{Alice}}$ being minimal w.r.t. families which are larger ($\mathcal{V}^+$-DIB), smaller ($\mathcal{V}^-$-DIB), and equal to $\mathcal{V}_{Alice}$. In theory, optimal representations would be $\mathcal{V}_{Alice}$-minimal (Theorem 1), which are achieved by DIB and $\mathcal{V}^+$-DIB (Monotonicity). $\mathcal{V}^+$-DIB should nevertheless be harder to estimate than DIB (Prop. 3). In the last 3 columns of Table 1 we indeed observe that DIB performs best. $\mathcal{V}^+$-DIB performs worse, suggesting that IB's minimality is undesirable in practice. We also minimize a known lower bound of $I[Z; X]$ (VIB; [19]) and find that it performs worse than DIB.[8] We show results for different $\beta$ in Appx. E.10.

**Comparison to traditional regularizers**. To ensure that the previous experimental gains support our theory and are not necessarily true for other regularizers, we test different regularizers on Bob and see whether they also learn representations that ensure Alice's ERM will generalize. In Table 1, we show the results of: (i) No regularization; (ii) Stochastic representations (DIB with $\beta = 0$); (iii) Dropout [45]; (iv) Weight decay. We find that DIB significantly outperforms other regularizers, which supports our claims that $\mathcal{V}$-minimality is well-suited for enforcing generalization. We emphasize that we evaluate the regularizers in a setting which is closer to our theory: two-stage game, no implicit regularizers, and evaluated on log likelihood. We show in Appx. E.11 that DIB is a descent regularizer in standard classification settings but performs a little worse than dropout.

### 4.3 Probing Generalization in Deep Learning

Methods that predict or correlate with the generalization of neural networks have been of recent theoretical and practical interest [46–51], as they can shed light on the inductive biases in deep learning [52–54] and prescribe better training procedures [55–59]. Having empirically shown a strong link between the degree of $\mathcal{V}$-minimality and generalization (Fig. 2b), it is natural to ask whether it can predict the generalization of a trained model. Specifically, consider the first $L$ layers as an encoder from inputs X to representations $Z_L$, and subsequent layers as a classifier in $\mathcal{V}_L$. We hypothesize that $\hat{I}_{\mathcal{V}_L}[Z_L \to \mathrm{Dec}(X, \mathcal{Y}); \mathcal{D}]$ correlates well with the generalization of the network.

To test this, we follow Jiang et al. [50] and train convolutional networks (CNN) with varying hyperparameters (depth, width, dropout, batch size, weight decay, learning rate, dimensionality of Z) and retain those that reach $0.01$ empirical risk. From this set of 562 models, we measure Kendall's rank correlation [60] between $\hat{I}_{\mathcal{V}_L}[Z_L \to \mathrm{Dec}(X, \mathcal{Y}); \mathcal{D}]$ and the generalization gap of each CNN, i.e., the difference between their train and test performance. For experimental details see Appx. D.3.

Table 2: Rank correlation $\tau$ between different measures and generalization gap (in terms of accuracy $\tau_{acc}$ and log loss $\tau_{logloss}$) of 562 CNNs. $\mathcal{V}_L$ denotes our $\hat{I}_{\mathcal{V}_L}[Z_L \to \mathrm{Dec}(X, \mathcal{Y}); \mathcal{D}]$.

| | $\mathcal{V}_L^+$ | $\mathcal{V}_L^-$ | $\mathcal{V}_L$ | Entropy | Path Norm | Var. Grad. | Sharp. Mag. |
|---|---|---|---|---|---|---|---|
| $\tau_{acc.}$ [50] | | | | 0.148 | 0.373 | 0.311 | **0.484** |
| $\tau_{acc}$ (ours) | **0.482** | 0.391 | 0.471 | 0.234 | 0.347 | 0.332 | 0.385 |
| $\tau_{logloss}$ (ours) | **0.505** | 0.435 | 0.498 | 0.164 | 0.357 | 0.167 | 0.233 |

**Does $\mathcal{V}$-minimality correlate with generalization?** In the last five columns of Table 2, we compare our results ($\mathcal{V}_L$) to the best generalization measure from each categories investigated in [50]:

the entropy of the output [61], the path norm [62], the variance of the gradients after training (Var. Grad. ; [50]), and the "sharpness" of the minima (Sharp. Mag.; [57]).[9] As hypothesized, $\hat{I}_{\mathcal{V}_L}[Z_L \to \text{Dec}(X, \mathcal{Y}); \mathcal{D}]$ correlates with generalization and even outperforms the baselines. Similarly to Table 1 we also evaluate minimality with respect to a family larger ($\mathcal{V}_L^+$) or smaller ($\mathcal{V}_L^-$) than $\mathcal{V}_L$. Surprisingly, $\mathcal{V}_L^+$ performs better than $\mathcal{V}_L$, which might be because larger networks can help optimization of sub-networks $\mathcal{V}_L \subseteq \mathcal{V}_L^+$ as suggested by the Lottery Ticket Hypothesis [63].

To the best of our knowledge $\mathcal{V}$-minimality is the first measure of generalization of a network that only considers a single internal representation $Z_L$. This could be of particular interest in transfer learning, as it can predict how well any model of a certain architecture will generalize when using a specific pretrained encoder. As $\mathcal{V}$-minimality is a property of a representation rather than the architecture, we show in Appx. E.12 that it can be meaningfully compared *across* different architectures and datasets.

## 5   Other Related Work

**Generalized information, game theory and Bayes decision theory**. If you need a distribution $P_X^*$ to act as a representative $\Gamma \subseteq \mathcal{P}(\mathcal{X})$ you should follow the maximum entropy (MaxEnt) principle [64, 65] to minimize the worst-case log loss [66, 67]. Grünwald et al. [68] generalized MaxEnt to different losses by framing the problem as an adversarial game between nature and a decision maker. Robust supervised learning [69] can also be framed in a way that suggests to maximize *conditional* entropy [70, 71]. This line of work focuses on prediction rules (Alice). Our framing (Sec. 2.1) extends this literature by incorporating a co-operative agent (Bob), which learns representations to minimize the worst-case loss of the decision maker (Alice). Although [10, 72] also studied representations using generalized information, they focused on consistency rather than generalization.

**Extended sufficiency and minimality**. Linear sufficiency is well studied [73–75] but only considers linear encoders and predictors and is used for estimation rather than predictions. In ML, Cvitkovic and Koliander [76] incorporated the encoder's family (Bob) to characterize achievable Z. This is complementary to our incorporation of the decoder's family $\mathcal{V}$ (Alice) to characterize optimal Z.

**Kernel Learning**. There is a large literature in learning kernels [77–80] for support vector machines [2], which implicitly learns a data representation [81]. The learning is either done by minimizing estimates [82, 83] or bounds of the generalization error [84–88]. The major advantage of our work is that we are not restricted to predictors $\mathcal{V}$ that can be "kernelized" and provide an optimality proof.

## 6   Conclusion and Future Work

In this work, we propose a prescriptive theory for representation learning. We first characterize optimal representations $Z^*$ for supervised learning, by defining minimal sufficient representation with respect to a family of classifiers $\mathcal{V}$. These representations $Z^*$ guarantee that any downstream empirical risk minimizer $f \in \mathcal{V}$ will incur minimal expected test loss, by ensuring that $f$ can correctly predict labels but cannot distinguish examples with the same label. We then provide the decodable information bottleneck objective to learn $Z^*$ with PAC-style guarantees. We empirically show that using $Z^*$ can improve the performance and robustness of image classifiers. We also demonstrate that our framework can be used to predict generalization in neural networks.

In addition to supporting our theory, our experiments raise interesting questions for future work. First, results in Sec. 4.2 suggest that performance is causally related with the degree of $\mathcal{V}$-minimality of a representation, even though we only prove it for "perfect" $\mathcal{V}$-minimality. A natural question, then, is whether generalization bounds can be derived for approximate $\mathcal{V}$-minimality. Second, the high correlation between generalization in neural networks and the degree $\mathcal{V}$-minimality (Table 2) suggest that it might be an important quantity to study for understanding generalization in deep learning.

More generally, our work shows that information theory in theoretical and applied ML can benefit from incorporating the predictive family $\mathcal{V}$ of interest. For example, we believe that many issues of mutual information [89] in self-supervised learning [90–92], and IB [33, 93, 94] in IB's theory of deep learning [14, 95] could be solved by taking into account $\mathcal{V}$. By extending $\mathcal{V}$-information to arbitrary r.v. (through decompositions) we hope to enable its use in those and many other domains.

## Broader Impact

Our work takes the perspective that an "optimal" representation is one such that any classifier that fits the training data should generalize well to test. In terms of potential practical benefits, it is possible that using our optimal representations, one can alleviate the effort of hyperparameter search and selection currently required to tune deep learning models. This could be a step towards democratizing machine learning to sections of the society without large computational resources – since hyperparameter search is often computationally expensive. We do not anticipate that our work will advantage or disadvantage any particular group.

## Acknowledgments and Disclosure of Funding

We would like to thank: Naman Goyal for early feedback and best engineering practices; Brandon Amos for support concerning min-max optimization; Stephane Deny for suggesting to look for the Worst ERM; Emile Mathieu, Chris Maddison, Sho Yaida, and Max Nickel for helpful discussions and feedback; Jakob Foerster for the name "decodable" information bottleneck; Dan Roy for suggesting to use the term "distinguishability" to understand $\mathcal{V}$-minimality; and Ari Morcos for tips to help Yann Dubois writing papers. DJS was partially supported by the NSF through the CPBF PHY-1734030, a Simons Foundation fellowship for the MMLS, and by the NIH under R01EB026943.

## Footnotes

[1]Rather than defining learning in terms of deterministic hypotheses $h \in \mathcal{H}$, we consider the more general [9] case of probabilistic predictors $\mathcal{V}$, in order to make a link with information theory.

[2] As an illustration, IB invariance to bijections suggests that a non-linearly entangled representation is as good as a linearly separable one if there is a bijection between them, even when classifying using a logistic regression.

[3]As shown in Appx. C.1.1, this common [12, 24] definition is a generalization of minimal sufficient *statistics* [25] to stochastic statistics $Z = T(X, \epsilon)$ where $\epsilon$ is a source of noise independent of X.

[4][23] also replace $H[Y]$ with $H_{\mathcal{V}}[Y \mid \varnothing]$. This requires that any $f \in \mathcal{V}$ can be conditioned on the empty set $\varnothing$. We keep $H[Y]$ for simplicity and show that both are equivalent in our setting (Appx. C.1.2).

[5]Notice that $\mathrm{I}_{\mathcal{V}}[\mathrm{Z} \to \mathrm{Y}]$ corresponds to the variational lower bound on $\mathrm{I}[\mathrm{Y}; \mathrm{Z}]$ used by Alemi et al. [19]. We view $\mathrm{I}_{\mathcal{V}}[\mathrm{Z} \to \mathrm{Y}]$ as the correct criterion rather than an estimate of $\mathrm{I}[\mathrm{Y}; \mathrm{Z}]$.

[6]Such (deterministic) labelings are also called "random labelings" [37] as they are semantically meaningless.

[7]Achille and Soatto [20] show that this happens for minimal Z. The novelty is that we obtain similar results when considering $\mathcal{V}$-minimality, which is less stringent (Prop. 2) and does not require the intractable $I[X; Z]$.

[8] VIB is hard to compare to DIB as it is unclear w.r.t. which family, if any, VIB's solutions are minimal.

[9] We report Jiang et al.'s [50] results since our experiments and Sharp. Mag. differs slightly from theirs.

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
