[Supplementary Material]

In the following appendices we: (i) Formalize our notation in Appx. A; (ii) State and discuss our assumptions in Appx. B; (iii) State and prove our theoretical results in Appx. C; (iv) Provide details for reproducing our results in Appx. D; (v) Provide and discuss additional results that shed light on many of our design choices Appx. E.

# A    Notation

Letters that are upper-case non-italic Y, calligraphic $\mathcal{Y}$, and lower-case $y$, represent, respectively, a random variable (r.v.), its associated codomain, and a realization of it. When necessary to be explicit, we will say that Y takes value in (t.v.i.) $\mathcal{Y}$. Conditional distribution will be denoted $P_{Y\,|\,Z} \in \mathcal{P}(\mathcal{Y}|\mathcal{Z})$, and the image of $z$ as $P_{Y\,|\,z} \in \mathcal{P}(\mathcal{Y})$, where $\mathcal{P}(\mathcal{Y})$ denotes the collection of all probability measures on $\mathcal{Y}$ with its $\sigma$-algebra and $\mathcal{P}(\mathcal{Y}|\mathcal{Z}) := \{P : \mathcal{Z} \to \mathcal{P}(\mathcal{Y})\}$ is used as a shorthand. The composition of a function $f$ with a random variable Z will be denoted $f(Z)$. Expectations will be written as: $E_{y\sim P_Y}[y] := \int y\,dP_Y$. Independence between two r.v.s will be denoted with $\cdot \perp \cdot$. The indicator function is denoted as $\mathbb{1}[\cdot]$. The cardinality of a set is denoted by $|\cdot|$. The preimage of $\{x\}$ by $f$ will be denoted $f^{\leftarrow}(\{x\})$. Finally, a hat $\hat{\cdot}$ will be used to refer to empirical estimates: (i) $\hat{Y}$ is an approximation of Y (so $P_Y, P_{\hat{Y}} \in \mathcal{P}(\mathcal{Y})$); (ii) $\hat{P}_Y$ denotes an empirical distribution of Y; (iii) Functionals with expectations taken over empirical distributions inherit the hat (e.g. $\hat{H}[Y\,|\,X]$).

Letters X,Z,Y are respectively used to refer to the input, representation and target of a predictive task. We use $X_y, Z_y$ to respectively denote an input and a representations that have the same distribution as X,Z conditioned on $y$, i.e., $P_{X_y} = P_{X|y}$ and $P_{Z_y} = P_{Z|y}$. We denote by $\mathcal{V}$ any predictive family, i.e, $\mathcal{V} \in \mathcal{P}(\mathcal{Y}|\mathcal{Z})$ and satisfies the assumptions in Appx. B.2. The largest such set if the universal predictive family $\mathcal{U} := \mathcal{P}(\mathcal{Y}|\mathcal{Z})$. The probability of $y \in \mathcal{Y}$ given $z \in \mathcal{Z}$ as predicted by a classifier $f \in \mathcal{V}$ is denoted $f[z](y)$ to distinguish it from the underlying conditional probability $P_{Y|Z}(y|z)$. We are interested in minimizing the expected loss of a classifier $f \in \mathcal{V}$, also called risk $R(f, Z) := E_{y,x\sim P_{Y,X}}\big[E_{z\sim P_{Z\,|\,x}}[S(y, f[z])]\big]$. In practice we will be given a training set of $M$ input-target pairs $\mathcal{D} \overset{\text{i.i.d.}}{\sim} P_{X,Y}^M$, in which case we can estimate the risk using the empirical risk $\hat{R}(f, Z; \mathcal{D}) := \frac{1}{M}\sum_{y,x\in\mathcal{D}} E_{z\sim P_{Z\,|\,x}}[S(y, f[z])]$. The set of ERMs are denoted as $\hat{\mathcal{V}}(\mathcal{D}) := \arg\min_{f\in\mathcal{V}} \hat{R}(f, Z; \mathcal{D})$. Finally, we will denote the best achievable risk for $\mathcal{V}$ as $R^*(\mathcal{V}) := \min_Z \min_{f\in\mathcal{V}} R(f, Z)$.

# B    Assumptions

## B.1    Generic Assumptions

We make a some assumptions throughout our paper to have concise statements. First let us discuss generic assumptions about the setting we are studying:

**At least one example per class**  We assume that every training set has at least one example per label. This is generally true in modern ML, where $|\mathcal{D}| \gg |\mathcal{Y}|$. Theorem 1 would not hold without it, as ERMs could not perform optimally without having examples to learn from.

**Logarithmic score**  We only consider the log loss $S(y, f[z]) := -\log f[z](y)$ as it is the most common scoring rule. Indeed, it is (essentially) the only *strictly proper* (strictly minimized by the underlying distribution $P_{Y\times Z}$) and *local* (depending only on predicted probability of the observed event $P_{Y\times Z}(y|Z)$) scoring rule [96], making it computationally attractive. The framework can likely be extended to any proper scoring rule (e.g. pseudo-likelihood, Brier score, kernel scoring rule) by considering generalized predictive entropy [34, 68, 97].

**Finite sample spaces**  We restrict ourselves to finite $|\mathcal{X}|, |\mathcal{Y}|, |\mathcal{Z}|$, so as to avoid the use of measure theory and axiomatic set theory, which would obscure the main points of the paper. While this assumption holds in computational ML (due to the use of digital computer or the fact that we can always restrict the sample spaces to the finite examples seen in our training and testing set), it is unsatisfactory from a theoretical standpoint and the general case should be investigated in future work. We conjecture that Theorem 1 extends to the uncountable case.

**At least as many representations as labels**  The sample space of representations is at least as large as the one for labels: $|\mathcal{Z}| \geq |\mathcal{Y}|$. This holds in practice where there are usually less than a 1000

possible labels $\mathcal{Y}$ while even a single dimensional Z can often (depending on computer) take $2^{32} \approx 4 * 10^6$ values.

**Multi-class classification** The sample space of the target is $\mathcal{Y} = [0, \ldots, |\mathcal{Y}| - 1]$.

## B.2 Assumptions on Functional Families

Now let us discuss the assumptions that we make about functional families. The following assumptions hold for many functional families that are used in practice, including neural networks, logistic regression, and decision tree classifiers.

**Invariance of $\mathcal{V}$ to label permutations** All predictive families are invariant to permutation, i.e., $\forall \mathcal{V}, \ \forall \pi : \mathcal{Y} \to \mathcal{Y}, \ \forall f \in \mathcal{V}, \ \exists f' \in \mathcal{V}$ s.t. $\forall z \in \mathcal{Z}, \forall y \in \mathcal{Y}$ we have $\forall f[z](y) = f'[z](\pi(y))$. This holds in practice (neural networks, decision trees, ...) as we usually do not want predictors to depend on the order of labels, e.g. $\mathcal{Y} = \{\text{``cat''}, \text{``dog''}\}$ or $\mathcal{Y} = \{\text{``dog''}, \text{``cat''}\}$. We use this assumption to simplify the proof of Theorem 1.

**Non-empty preimage of labels** We consider predictive families that have a non empty preimage for each label: $\forall \mathcal{V}, \exists f \in \mathcal{V}$, s.t. $\forall y \in \mathcal{Y}, \ \exists z \in \mathcal{Z}$ we have $f[z](y) = 1$. This is usually true in ML. In neural networks, this can be achieved by making the weights of your last layer very large such that the softmax will give the label a probability of 1 (achieved due to floating point representation). We use this assumption to show that when the label is deterministic, $\mathrm{R}^*(\mathcal{V}) = 0$.

**Arbitrary constant prediction of $\mathcal{V}$** We assume that in all functional families there always is a predictor which predicts any constant output regardless of the input: $\forall \mathcal{V}, \ \forall P_Y \in \mathcal{P}(\mathcal{Y}), \ \exists f \in \mathcal{V}$ s.t. $\forall z \in \mathcal{Z}$ we have $f[z] = P_Y$. This is typically true in classification, when the last layer parametrizes a categorical distribution. In neural networks this can be achieved by setting all weights to $0$ and then the bias of the last layer (softmax) to the desired values. Notice that this not true in the general case (regression and countable infinite sample space), in which case the assumption can be relaxed to optional ignorance as in [23]. We use this assumption to simplify the definition of $\mathcal{V}$-information in Prop. 5.

**Monotonic biasing of $\mathcal{V}$** We assume that all functional families are closed under "monotonic biasing towards a prediction $y$". Formally, $\forall f' \in \mathcal{V}, \ \forall y \in \mathcal{Y}, \ \forall z \in \mathcal{Z}, \ \forall p \in [0, 1], \exists g \in \mathcal{V}$ s.t. $g[z](y) = p$ and $\forall z', z'' \in \mathcal{Z}, \forall y' \in \mathcal{Y}$ we have $\mathrm{sign}(f'[z'](y') - f'[z''](y')) = \mathrm{sign}(g[z'](y') - g[z''](y'))$. In other words, it is possible to construct a $g \in \mathcal{V}$ that assigns to a (single) pair $z, y$ the probability $p$ of your choice and preserves the order — if $z, y$ was assigned a higher probability than $z', y$ by $f'$ then the same holds for $g$. Such assumption holds for neural networks, as it is always possible to construct $g$ by modifying the bias term of the final softmax layer. This assumption is crucial for the proof of Theorem 1.

## B.3 Assumptions for the Theorem

We make an additional assumptions for Theorem 1 and Prop. 2.

**Deterministic Labeling** We assume that labels are deterministic functions of the data $\exists t : \mathcal{X} \to \mathcal{Y}$ s.t. $Y = t(X)$. This is generally true in ML datasets where every example is only seen once and thus every example is given a single label with probability 1. This does not necessarily hold in the real world. We use this assumption to simplify the proofs, we believe that it is not necessary for the theorem to hold but should be investigated in future work.

# C  Theoretical Results and Proofs

## C.1  Background

### C.1.1  Minimal Sufficient Statistics

In the following, we clarify the link between minimal sufficient *statistics* [25] and *representations* [12, 24] of inputs X. The difference between a representation Z in IB and a statistic $T(X)$, is that the mapping between the inputs X and the representation Z can be stochastic — specifically a representation is a statistic of the input and independent noise $\epsilon \perp X$, i.e., $Z = T(X, \epsilon)$. We now

prove that for (deterministic) statistics, the notion of minimal sufficient representation is equivalent to that of predictive minimal sufficient statistics [98].

**Definition 3** (Minimal Sufficient Representations). A representation $Z = T(X, \epsilon)$ is:

- **Sufficient for** Y if $Z \in \mathcal{S} := \arg\max_{Z'} I[Y; Z']$

- **Minimal Sufficient for** Y if $Z \in \mathcal{M} := \arg\min_{Z' \in \mathcal{S}} I[X; Z']$

**Definition 4** (Sufficient Statistic). A statistic $Z = T(X)$ is predictive sufficient for Y if $Y - Z - X$ forms a Markov Chain.

**Lemma 1** (Equivalence of Sufficiency). Let Z be a statistic $T(X)$ or a representation $T(X, \epsilon)$ of X, then Z is predictive sufficient for Y by Def. 4 $\iff$ Z is sufficient for Y by Def. 3.

*Proof.* We prove the following for statistics $Z = T(X)$ but the same proof holds for representations $T(X, \epsilon)$. For both directions we use the fact that for any statistics $\max_{Z'} I[Y; Z'] = I[Y; X]$. Indeed, $Y - X - Z$ constitutes a Markov Chain as $Z = T(X)$. From the data processing inequality (DPI) we have $I[Y; X] \geq I[Y; Z]$, where the equality is achieved by using the identity statistic $Z = X$.

($\implies$) Suppose Z is sufficient by Def. 4. Since Z is a statistic we again have $I[Y; X] \geq I[Y; Z]$. From Def. 4, we also have $Y - Z - X$ which implies (DPI) $I[Y; Z] \geq I[Y; X]$. Due to the upper and lower bound we must have $I[Y; X] = I[Y; Z]$, which is equivalent to Def. 3.

($\impliedby$) Assume that Z is sufficient by Def. 3. Using the chain rule of information we have

$$
\begin{aligned}
I[Y; Z, X] &= I[Y; Z, X] \\
I[Y; Z] + I[Y; X \mid Z] &= I[Y; X] + I[Y; Z \mid X] \\
\max_{Z'} I[Y; Z'] + I[Y; X \mid Z] &= I[Y; X] + I[Y; Z \mid X] \qquad\qquad \text{Def. 3} \\
I[Y; X \mid Z] &= I[Y; Z \mid X] \\
I[Y; X \mid Z] &= 0 \qquad\qquad\qquad\qquad\qquad\quad Y - X - Z
\end{aligned}
$$

The fourth line comes from $\arg\max_{Z'} I[Y; Z'] = I[Y; X]$. The last line holds as Z is a statistic of X. $I[Y; X \mid Z] = 0$ implies that $Y \perp X \mid Z$ so $Y - Z - X$ is a Markov Chain, which concludes the proof. □

**Definition 5** (Minimal Sufficient Statistic). A sufficient statistic Z is *minimal* if for any other sufficient statistic $Z'$, there exists a function $g$ such that $Z = g(Z')$.

**Proposition 4** (Equivalence of Minimal Sufficiency). Let $Z = T(X)$ be a (deterministic) statistic, then Z is minimal (by Def. 5) and sufficient for Y (by Def. 4) $\iff$ Z is minimal sufficient by Def. 3.

*Proof.* From Lemma 1 we know that the sufficiency requirements are equivalent in Def. 5 and Def. 3. We now need to prove that the minimality requirements are also equivalent.

($\implies$) Let Z be minimal by Def. 5, then for all other sufficient $Z'$ we have the Markov Chain $X - Z' - Z$. From the DPI, $I[X; Z] \leq I[X; Z']$. This completes the first direction of the proof.

($\impliedby$) We will prove it by contrapositive. Suppose $Z := T(X) \in \mathcal{S}$ is not minimal by Def. 5, i.e. there exists a sufficient statistic $Z' := T'(X) \in \mathcal{S}$ s.t. no function $g$ satisfies $T(X) = g(T'(X))$. Then the binary relation $\{(T'(x), T(x)) \mid x \in \mathcal{X}\}$ is not univalent, therefore the converse relation $\{(T(x), T'(x)) \mid x \in \mathcal{X}\}$ is not injective. As a result, there exists a non injective function $\tilde{g}$ such that $T'(X) = \tilde{g}(T(X))$. From the DPI we have $I[X; Z'] < I[X; Z]$ with a strict inequality due to the non injectivity of $\tilde{g}$. So Z is not minimal by Def. 3, thus concluding the proof. □

We emphasize that the second implication ($\impliedby$) does *not* hold in the case of a representation $Z = T(X, \epsilon)$. Indeed, Def. 5 is not really meaningful for "stochastic" representations.

### C.1.2 Replacing $H_{\mathcal{V}}[Y \mid \varnothing]$ by $H[Y]$

Due to our "arbitrary constant prediction of $\mathcal{V}$" assumption, we can replace $H_{\mathcal{V}}[Y \mid \varnothing]$ by $H[Y]$ in Xu et al.'s [23] definition of $\mathcal{V}$-information.

**Proposition 5.** For all predictive families $\mathcal{V}$ we have $H_{\mathcal{V}}[Y \mid \varnothing] = H[Y]$.

*Proof.* Denote $\mathcal{V}_{\varnothing} \subset \mathcal{V}$ the subset of $f$ that satisfy $f[x] = f[\varnothing]$, $\forall x \in \mathcal{X}$.

$$
\begin{aligned}
H_{\mathcal{V}}[Y \mid \varnothing] &:= \inf_{f \in \mathcal{V}} E_{z,y \sim P_{Z,Y}}[-\log f[\varnothing](y)] \\
&= \inf_{f \in \mathcal{V}_{\varnothing}} E_{z,y \sim P_{Z,Y}}[-\log f[\varnothing](y)] \\
&= \inf_{f \in \mathcal{V}_{\varnothing}} E_{z,y \sim P_{Z,Y}}[-\log f[z](y)] \\
&= E_{y \sim P_Y}[-\log P_Y] \qquad\qquad \text{Properness and Arbitrary Const. Pred.} \\
&= H[Y]
\end{aligned}
$$

The penultimate line uses the properness of the log loss (best unconditional predictor of y is $P_Y$) and our assumption regarding "arbitrary constant prediction", which implies that there exists $f \in V$ s.t. $\forall z \in \mathcal{Z}$ we have $f[z] = P_Y$. $\qquad\square$

### C.2 $\mathcal{V}$-Sufficiency

In this subsection, we prove our claims in Sec. 3.1. First, let us show that $H_{\mathcal{V}}[Y \mid Z]$ is indeed the best achievable risk for Z.

**Lemma 2.** For any predictive family $\mathcal{V}$, $\min_{f \in \mathcal{V}} R(f, Z) = H_{\mathcal{V}}[Y \mid Z]$.

*Proof.* This directly come from the definition of predictive information:

$$
\begin{aligned}
H_{\mathcal{V}}[Y \mid Z] &:= \inf_{f \in \mathcal{V}} E_{z,y \sim P_{Z,Y}}[-\log f[z](y)] \\
&= \inf_{f \in \mathcal{V}} E_{y \sim P_Y}\left[E_{z \sim P_{Z|y}}[-\log f[z](y)]\right] \\
&= \inf_{f \in \mathcal{V}} E_{y \sim P_Y}\left[E_{x \sim P_{X|y}}\left[E_{z \sim P_{Z|x}}[-\log f[z](y)]\right]\right] \qquad Y - X - Z \\
&= \inf_{f \in \mathcal{V}} E_{y,x \sim P_{Y,X}}\left[E_{z \sim P_{Z \mid x}}[-\log f[z](y)]\right] \\
&= \inf_{f \in \mathcal{V}} R(f, Z) \qquad\qquad\qquad\qquad\qquad\qquad\qquad \text{Def. Risk} \\
&= \min_{f \in \mathcal{V}} R(f, Z) \qquad\qquad\qquad\qquad\qquad\qquad\qquad \text{Finite Sample Space}
\end{aligned}
$$

$\square$

Proposition 1 is a trivial corollary of the previous lemma.

**Proposition 1.** Z is $\mathcal{V}$-sufficient $\iff$ there exists $f^* \in \mathcal{V}$ whose test loss when predicting from Z is the best achievable risk, i.e., $R(f^*, Z) = \min_Z \min_{f \in \mathcal{V}} R(f, Z)$.

*Proof.*

$$
\begin{aligned}
\mathcal{S}_{\mathcal{V}} &:= \arg\max_Z I_{\mathcal{V}}[Z \to Y] \\
&= \arg\max_Z H[Y] - H_{\mathcal{V}}[Y \mid Z] \\
&= \arg\min_Z H_{\mathcal{V}}[Y \mid Z] \qquad\qquad\qquad \text{Const. } H[Y] \\
&= \arg\min_Z \min_{f \in \mathcal{V}} R(f, Z) \qquad\qquad\quad \text{Lemma 2}
\end{aligned}
$$

$\square$

Let us now show that when the label is deterministic $\forall V,\ \mathrm{R}^*(\mathcal{V}) = 0$. This may be counterintuitive, but the following proof shows that we are simply shifting the burden of classification from the classifier to the encoder — which is unconstrained.

**Proposition 6.** Assume that labels are a deterministic function of the data $\exists t : \mathcal{X} \to \mathcal{Y}$ s.t. $\mathrm{Y} = t(\mathrm{X})$, then for any predictive family $\mathcal{V}$ the best achievable risk is $\min_{\mathrm{Z}} \min_{f \in \mathcal{V}} \mathrm{R}(f, \mathrm{Z}) = 0$.

*Proof.* First notice that $\min_{\mathrm{Z}} \min_{f \in \mathcal{V}} \mathrm{R}(f, \mathrm{Z}) \geq 0$ due to the non-negativity of the log loss. We show that the inequality is an equality by constructing a representation $\mathrm{Z}^*$ and a $f \in V$ such that $\mathrm{R}(f, \mathrm{Z}^*) = 0$. Intuitively, we do so by finding "buckets" of Z that correspond to a certain label and then having an encoder which essentially classifies each input $x$ to the correct bucket. Formally:

Let $f^{\leftarrow}(\{y\}) := \{z \in \mathcal{Z} \text{ s.t. } f[z](y) = 1\}$ denote the preimage of a deterministic label by a classifier $f$. By the "Non-empty Preimage of Labels" assumption we know that $\forall \mathcal{V}$ there exists $f \in \mathcal{V}$ s.t. $\forall y \in \mathcal{Y}$, the preimage is non-empty $|f^{\leftarrow}(\{y\})| \geq 0$. Let $f$ be one of those predictors. We construct the desired $\mathrm{Z}^*$ by setting its probability mass function $\forall z \in \mathcal{Z}, x \in \mathcal{X}$ as a uniform distribution over the $f$ preimage of the label of $x$ (deterministic label assumption $\mathrm{Y} = t(x)$) .

$$P_{\mathrm{Z}^* \mid \mathrm{X}}(z|x) := \begin{cases} \frac{1}{|f^{\leftarrow}(\{t(x)\})|} & \text{if } z \in f^{\leftarrow}(\{t(x)\}) \\ 0 & \text{else} \end{cases} \tag{9}$$

We now show that the risk $\mathrm{R}(f, \mathrm{Z}^*)$ is indeed 0:

$$\begin{aligned}
\mathrm{R}(f, \mathrm{Z}^*) &:= \mathbb{E}_{y,x \sim P_{\mathrm{Y},\mathrm{X}}} \left[ \mathbb{E}_{z \sim P_{\mathrm{Z} \mid x}}[- \log f[z](y)] \right] \\
&= \sum_{y \in \mathcal{Y}} \sum_{x \in \mathcal{X}} \sum_{z \in \mathcal{Z}} P_{\mathrm{Y}}(y) P_{\mathrm{X} \mid \mathrm{Y}}(x \mid y) P_{\mathrm{Z}^* \mid \mathrm{X}}(z \mid x) [- \log f[z](y)] \\
&= \sum_{y \in \mathcal{Y}} \sum_{x \in \mathcal{X}} \sum_{z \in f^{\leftarrow}(\{t(x)\})} P_{\mathrm{Y}}(y) P_{\mathrm{X} \mid \mathrm{Y}}(x|y) \frac{- \log f[z](y)}{|f^{\leftarrow}(\{t(x)\})|} && \text{Eq. (9)} \\
&= \sum_{y \in \mathcal{Y}} \left[ \sum_{z \in f^{\leftarrow}(\{y\})} P_{\mathrm{Y}}(y) \frac{- \log f[z](y)}{|f^{\leftarrow}(\{y\})|} \right] \left[ \sum_{x \in \mathcal{X}} P_{\mathrm{X} \mid \mathrm{Y}}(x|y) \right] \\
&= \sum_{y \in \mathcal{Y}} \left[ \sum_{z \in f^{\leftarrow}(\{y\})} P_{\mathrm{Y}}(y) \frac{- \log 1}{|f^{\leftarrow}(\{y\})|} \right] * 1 && \text{Def. } f \\
&= 0
\end{aligned}$$

The fourth line uses $y = t(x)$, thus removing the dependence with X. The penultimate line uses $\forall z \in f^{\leftarrow}(\{y\}), f[z](y) = 1$ which is the defining property of the selected $f$. $\qquad\square$

### C.3 Theorem

We will now prove the main result of our paper, namely that *any* ERM that uses a $\mathcal{V}$-minimal $\mathcal{V}$-sufficient representation will reach the best achievable test loss.

**Theorem.** Suppose Y is a deterministic labeling $t(\mathrm{X})$. Let $\mathcal{V} \in \mathcal{P}(\mathcal{Y}|\mathcal{Z})$ be a predictive family satisfying the assumptions in Appx. B.2. Under the assumptions stated in Appx. B.1 , we have that: if Z is a $\mathcal{V}$-minimal $\mathcal{V}$-sufficient representation of X for Y, then any ERM on any dataset will achieve the best achievable risk, i.e.

$$\mathrm{Z} \in \mathcal{M}_{\mathcal{V}} \implies \forall M \geq |\mathcal{Y}|,\ \mathcal{D} \overset{\text{i.i.d.}}{\sim} P_{\mathrm{X},\mathrm{Y}}^M,\ \forall \hat{f} \in \hat{\mathcal{V}}(\mathcal{D}) \text{ we have } \mathrm{R}(\hat{f}, \mathrm{Z}) = \min_{\mathrm{Z}} \min_{f \in \mathcal{V}} \mathrm{R}(f, \mathrm{Z})$$

#### C.3.1 Lemmas for Theorem 1

In this subsection we show three simple lemmas that are useful for proving Theorem 1. First we show that in the deterministic label setting, $\mathcal{V}$-sufficiency implies that the representaion space can be partitioned by $\mathcal{Y}$, i.e., the supports of each $Z_y$ are non-overlapping.

**Lemma 3.** Assume Y is a deterministic labeling $t(X)$. Then $Z \in \mathcal{S}_\mathcal{V} \implies \forall y \neq y'$, $y, y' \in \mathcal{Y}$ we have $\mathrm{supp}(P_{Z|y}) \cap \mathrm{supp}(P_{Z|y'}) = \varnothing$.

*Proof.* Let us prove it by contrapositive. Namely, we will show $\mathrm{supp}(P_{Z|y}) \cap \mathrm{supp}(P_{Z|y'}) \neq \varnothing \implies Z \notin \mathcal{S}_\mathcal{V}$. $\mathrm{supp}(P_{Z|y}) \cap \mathrm{supp}(P_{Z|y'}) \neq \varnothing$ implies that $\exists y \neq y' \in \mathcal{Y}, \exists z \in \mathcal{Z}$ s.t. $P_{Z|Y}(z|y') \neq 0$ and $P_{Z|Y}(z|y) \neq 0$. Using Bayes rule (and the fact that $P_Y$ has support for all labels), that means $P_{Y|Z}(y'|z) \neq 0$ and $P_{Y|Z}(y|z) \neq 0$ so $\nexists y \in \mathcal{Y}$ s.t. $P_{Y|Z}(y'|z) = 1$. Due to the finite sample space assumption and monotonicity of $\mathcal{V}$ predictive entropy, this implies $0 < \mathrm{H}[Y \mid Z] = \mathrm{H}_\mathcal{U}[Y \mid Z] \leq \mathrm{H}_\mathcal{V}[Y \mid Z]$. From Prop. 6 we conclude that $Z \notin \mathcal{S}_\mathcal{V}$ as desired. $\qquad\square$

We now show the simple fact that, if some classifier achieves zero test loss then being an ERM is equivalent to achieving zero training loss.

**Lemma 4.** Let $\mathcal{D} \overset{\text{i.i.d.}}{\sim} P_{X,Y}^M$ be a training dataset. Suppose $\exists f \in \mathcal{V}$ s.t. $\mathrm{R}(f, Z) = 0$, then: $\hat{f} \in \hat{\mathcal{V}}^*(\mathcal{D}) \iff \hat{\mathcal{R}}(\hat{f}, Z; \mathcal{D}) = 0$.

*Proof.* As $\mathrm{R}(f, Z)$ is an expectation of a non-negative discrete r.v., it is equal to zero if and only if $\forall z \in \mathrm{supp}(Z)$, $\forall y \in \mathrm{supp}(P_{Y|Z})$ we have $\log f[z](y) = 0$. $\hat{\mathcal{R}}(f, Z; \mathcal{D})$ is also a weighted average (discrete expectation) of $\log f[z](y) = 0$ over a subset of the previous support $\hat{z} \in \mathrm{supp}(\hat{Z}) \subseteq \mathrm{supp}(Z)$, $\forall \hat{y} \in \mathrm{supp}(\hat{Y}) \subseteq \mathrm{supp}(Y)$ so we conclude that $\hat{\mathcal{R}}(f, Z; \mathcal{D}) = 0$. As the minimal training loss is always zero and the risk cannot be less than zero (non negativity of log loss and finite sample space) the definition of ERMs becomes $\hat{\mathcal{V}}(\mathcal{D}) := \arg\min_{f \in \mathcal{V}} \hat{\mathrm{R}}(f, Z; \mathcal{D}) = \{f \in \mathcal{V} \text{ s.t. } \hat{\mathrm{R}}(f, Z; \mathcal{D}) = 0\}$ as desired. $\qquad\square$

A representation is $\mathcal{V}$-minimal $\mathcal{V}$-sufficient if and only if it has no $\mathcal{V}$-information with any of the terms in any $y$ decomposition of X.

**Lemma 5.** Assume Y is a deterministic labeling $t(X)$, then:

$$\forall Z \in \mathcal{M}_\mathcal{V} \iff Z \in \mathcal{S}_\mathcal{V} \text{ and } \forall y \in \mathcal{Y}, \forall N \in \mathrm{Dec}(X, y) \text{ we have } \mathrm{I}_\mathcal{V}[Z_y \to N] = 0.$$

*Proof.*
( $\impliedby$ ) Due to the non negativity of $\mathcal{V}$-information, we have $\forall Z' \in \mathcal{S}_\mathcal{V}$, $0 = \mathrm{I}_\mathcal{V}[Z_y \to N] \leq \mathrm{I}_\mathcal{V}[Z'_y \to N]$ so Z reaches the minimal achievable value in each term and thus also on their expectation $\mathrm{I}_\mathcal{V}[Z \to \mathrm{Dec}(X, \mathcal{Y})] = 0$. We thus conclude that $Z \in \mathcal{M}_\mathcal{V} := \arg\min_{Z \in \mathcal{S}_\mathcal{V}} \mathrm{I}_\mathcal{V}[Z \to \mathrm{Dec}(X, \mathcal{Y})]$.

( $\implies$ ) Let us show that there is at least one $Z \in \mathcal{M}_\mathcal{V}$ s.t. $\forall y \in \mathcal{Y}, \forall N \in \mathrm{Dec}(X, y)$, $\mathrm{I}_\mathcal{V}[Z_y \to N] = 0$, from which we will conclude that they all have to satisfy the previous property in order to minimize $\mathrm{I}_\mathcal{V}[Z \to \mathrm{Dec}(X, \mathcal{Y})]$. Let us consider $Z^*$ as defined in Eq. (9). Notice that $P_{Z_y^* \mid X_y}(z|x) = \frac{1}{|f^\leftarrow(\{t(x)\})|} = \frac{1}{|f^\leftarrow(\{y\})|} = P_{Z_y^*}(z_y)$, where the last equality comes from the fact that $z_y$ is associated with a single label $y$. An other way of saying it, is that $X_y - y - Z_y^*$ forms a Markov Chain, but $y$ is a constant. We thus conclude $\forall y \in \mathcal{Y}$ $Z_y^* \perp X_y$. By definition of $y$ decomposition of X (Eq. (6)), we also know that $\forall N \in \mathrm{Dec}(X, y)$, $\exists t' : \mathcal{X} \to \mathcal{Y}$ s.t. $N = t'(X_y)$, from which we conclude that $Z_y^* \perp N$. Due to the independence property of $\mathcal{V}$-information, we have $\forall y \in \mathcal{Y}$, $\forall N \in \mathrm{Dec}(X, y), \mathrm{I}_\mathcal{V}[Z_y \to N] = 0$ as desired. As we found one $Z^* \in \mathcal{M}_\mathcal{V}$ s.t. this is true, it must be for all $Z \in \mathcal{M}_\mathcal{V}$. Indeed, due to the positivity property it is the only way of reaching the minimal $\mathrm{I}_\mathcal{V}[Z \to \mathrm{Dec}(X, \mathcal{Y})] = 0$. $\qquad\square$

### C.3.2 Proof Intuition

The main difficulty in the proof is that $\mathcal{V}$-minimality removes information using deterministic labeling while the predictors $f \in \mathcal{V}$ are probabilistic.[10] As a result the proof is relatively long, here is a rough outline:

Figure 5: Intuition behind the construction of $g \in \mathcal{V}$ in the proof of Theorem 1. The plot schematically represents the all the representations $\mathcal{Z}_y$ associated with a label $y = 1$. The representations $\mathcal{Z}_y^t$ (green) are associated with training examples, $\mathcal{Z}_y^e$ (yellow) are associated with test / evaluation examples, $\mathcal{Z}_y^c$ (blue) are those that yield correct predictions of $y = 1$ by $f'$, $\mathcal{Z}_y^w$ (orange) are those that yield wrong predictions of $y = 1$ by $f'$.

1. As the theorem is about *all* ERMs, use a proof by contrapositive to only talk about a single ERM $f'$ that performs optimally on train but not on test.

2. Construct a random variable $\tilde{N}$ which labels train examples as 1 and test as 0. Show that $\tilde{N} \in \mathrm{Dec}(X, y)$.

3. Using the "monotonic biasing of $\mathcal{V}$" assumption, construct $g \in \mathcal{V}$ from $f' \in \mathcal{V}$ by monotonically biasing the predictions towards the "test" label $\tilde{N} = 0$ s.t. $g[z](0) = P_{\tilde{N}}(0)$ for every representation $z^{(c)}$ that are perfectly labelled by $f'$ (as shown in blue in Fig. 5).

4. Show that $g$ predicts $\tilde{N}$ better than the marginal distribution $P_{\tilde{N}}$ for representations $z^{(w)}$ that are not perfectly labelled by $f'$ (as shown in orange in Fig. 5), while predicting as well as $P_{\tilde{N}}$ for $z^{(c)}$. Conclude that $g$ predicts $\tilde{N}$ better than $P_{\tilde{N}}$.

5. Show that the previous point entails $\mathrm{I}_\mathcal{V}\left[Z \to \tilde{N}\right] \neq 0$. Conclude by Lemma 5 that $Z \notin \mathcal{M}_\mathcal{V}$ as desired.

### C.3.3    Formal Proof

*Proof.* If $Z \in \mathcal{M}_\mathcal{V}$ is $\mathcal{V}$-minimal $\mathcal{V}$-sufficient then by definition it is also $\mathcal{V}$-sufficient, we thus restrict our discussion to $\mathcal{V}$-sufficient representations. As $Z$ is $\mathcal{V}$-sufficient, $\exists f \in V$ s.t. $\mathrm{R}(f, Z) = \min_Z \min_{f \in \mathcal{V}} \mathrm{R}(f, Z) = 0$. The first equality comes from Prop. 1. The second equality comes from Prop. 6 and the deterministic assumption $Y = t(X)$. As there is some function $f$ with risk $\mathrm{R}(f, Z) = 0$, Lemma 4 tells us that being an ERM is equivalent to having zero empirical risk $\hat{f} \in \hat{\mathcal{V}}(\mathcal{D}) \iff \hat{\mathrm{R}}(\hat{f}, Z; \mathcal{D}) = 0$. We thus only need to prove that $\mathcal{V}$-minimality implies that every function with zero empirical risk will get zero actual risk, i.e., they all generalize: $Z \in \mathcal{M}_\mathcal{V} \implies \forall \mathcal{D} \overset{\text{i.i.d.}}{\sim} P_{X,Y}^M, \forall f \in \mathcal{V}$ s.t. $\hat{\mathrm{R}}(f, Z; \mathcal{D}) = 0$ will achieve $\mathrm{R}(f, Z) = 0$.

We will prove this statement by contrapositive, namely that the existence of a predictor with $0$ empirical risk but larger actual risk implies that the representation is not $\mathcal{V}$-minimal: $\exists \mathcal{D} \overset{\text{i.i.d.}}{\sim} P_{X,Y}^M, \exists f'$ s.t. $\hat{\mathrm{R}}(f', Z; \mathcal{D}) = 0 \wedge \mathrm{R}(f', Z) > 0 \implies Z \notin \mathcal{M}_\mathcal{V}$. For ease of notation let us assume that we are in a binary classification setting $|\mathcal{Y}| = 2$. We will later show how to reduce the multi-classification setting to the binary one.

Let $f'$ be a function with $\hat{\mathrm{R}}(f', Z; \mathcal{D}) = 0 \wedge \mathrm{R}(f', Z) > 0$. As the risk is positive, there must be some label $\exists y \in \mathcal{Y}$ s.t. when predicting from examples labeled as $y$ the expected loss will be positive $\mathrm{R}_y(f', Z) := \mathrm{E}_{x \sim P_{X|Y}}\left[\mathrm{E}_{z \sim P_{Z|X}}[-\log f'[z](y)]\right] > 0$. Due to the deterministic labeling assumption and Lemma 3, we can study one single such label $y$ without considering any other label $y' \in \mathcal{Y}$ (neither the examples nor the representations interact between labels). Without loss of generality — due to the invariance of $\mathcal{V}$ to label permutations — let us assume that this label is $y = 1$.

Let $\mathcal{X}_y^t$ be the set of examples (associated with $y$) seen during training and $\mathcal{X}_y^e = \mathcal{X}_y \setminus \mathcal{X}_y^t$ those only during test (eval). Let $\mathcal{Z}_y^w := \{z_y^{(w)} \in \mathcal{Z}_y | f'[z_y](y) \neq 1\}$ be the representations who are (wrongly)

not predicted $y$ by $f'$ with a probability of 1. Let $\mathcal{Z}_y^c := \mathcal{Z}_y \setminus \mathcal{Z}_y^w$ be the set of representations that are (correctly) labeled $y$ by $f'$ with a probability of 1. Notice that both these sets are non empty $|\mathcal{Z}_y^w| > 0$ and $|\mathcal{Z}_y^c| > 0$ because respectively $\mathrm{R}_y(f', \mathrm{Z}) > 0$ and $\hat{\mathrm{R}}_y(f', \mathrm{Z}, \mathcal{D}) = 0$.

Let $\tilde{\mathrm{N}} := \mathbb{1}[\mathrm{X}_y \in \mathcal{X}_y^t]$ be a binary "selector" of training examples. Notice that $\tilde{\mathrm{N}} \in \mathrm{Dec}(\mathrm{X}, y)$, as $\mathbb{1}[\cdot \in \mathcal{X}_y^t]$ is a deterministic function from $\mathcal{X}_y \to \mathcal{Y}$. We want to show that $\mathrm{H}_{\mathcal{V}}\left[\tilde{\mathrm{N}} \mid \mathrm{Z}_y\right] < \mathrm{H}\left[\tilde{\mathrm{N}}\right]$. To do so we have to find a function $g$ whose risk when predicting $\tilde{\mathrm{N}}$ is smaller than the entropy $\mathrm{H}\left[\tilde{\mathrm{N}}\right]$. Notice that $f'$ is close to being the desirable $g$, but not quite. [11]

We construct the desired $g$ by starting with $f'$ and monotonically increasing the probability of predicting $\tilde{\mathrm{N}} = 0$ s.t. $\forall z^{(c)} \in \mathcal{Z}_y^c$ we have $g[z^{(c)}] = P_{\tilde{\mathrm{N}}}$ as seen in Fig. 5. Specifically, from the "monotonic biasing of $\mathcal{V}$" we know that for $f' \in \mathcal{V}$, $y = 0$, some $z^{(c)} \in \mathcal{Z}_y^c$, and $p = P_{\tilde{\mathrm{N}}}$ there exists $g \in \mathcal{V}$ s.t. $g[z^{(c)}](0) = P_{\tilde{\mathrm{N}}}(0)$ and $\forall z', z'' \in \mathcal{Z}, \forall y' \in \mathcal{Y}$ we have $\mathrm{sign}(f'[z'](y') - f'[z''](y')) = \mathrm{sign}(g[z'](y') - g[z''](y'))$. Notice that $\forall \tilde{z}^{(c)} \in \mathcal{Z}_y^c$ we have $g[z](0) = P_{\tilde{\mathrm{N}}}(0)$ due to the ordering requirement of monotonic biasing. Indeed, we have just shown that this is true for one such $z^c$ and by construction $\forall \tilde{z}^{(c)} \in \mathcal{Z}_y^c$ we have $\mathrm{sign}(f'[z^{(c)}](0) - f'[\tilde{z}^{(c)}](0)) = \mathrm{sign}(0 - 0) = 0$ so $0 = \mathrm{sign}(g[z^{(c)}](0) - g[\tilde{z}^{(c)}](0)) = \mathrm{sign}(P_{\tilde{\mathrm{N}}}(0) - g[\tilde{z}^{(c)}](0))$ from which we conclude that $g[\tilde{z}^{(c)}](0) = P_{\tilde{\mathrm{N}}}(0)$ and $g[\tilde{z}^{(c)}](1) = P_{\tilde{\mathrm{N}}}(1)$ as we are in a binary setting.

Due to ordering requirement of monotonic biasing $\forall z^{(w)} \in \mathcal{Z}_y^w$ we have $g[z^{(w)}](0) > P_{\tilde{\mathrm{N}}}(0)$ as seen in Fig. 5. Indeed, $\forall z^{(w)} \in \mathcal{Z}_y^w$ we have $f'[z^{(w)}](0) > 0$ by construction $(f'[z^{(w)}](1) \neq 1)$ so by the ordering requirement $\mathrm{sign}(f'[z^{(c)}](0) - f'[z^{(w)}](0)) = \mathrm{sign}(-f'[z^{(w)}](0)) = -1 = \mathrm{sign}(g[z^{(c)}](0) - g[z^{(w)}](0)) = \mathrm{sign}(P_{\tilde{\mathrm{N}}}(0) - g[z^{(w)}](0))$ from which we conclude that $g[z^{(w)}](0) > P_{\tilde{\mathrm{N}}}(0)$.

In other words, $g$ predicts training examples $\tilde{\mathrm{N}} = 0$ in the same way as $P_{\tilde{\mathrm{N}}}$ but testing examples better than $P_{\tilde{\mathrm{N}}}$. From here it should be clear that $\mathrm{H}_{\mathcal{V}}\left[\tilde{\mathrm{N}} \mid \mathrm{Z}_y\right] < \mathrm{H}\left[\tilde{\mathrm{N}}\right]$, which we prove below for completeness.

$$
\begin{aligned}
\mathrm{I}\left[\mathrm{Z}_y \to \tilde{\mathrm{N}}\right] &:= \mathrm{H}\left[\tilde{\mathrm{N}}\right] - \mathrm{H}_{\mathcal{V}}\left[\tilde{\mathrm{N}} \mid \mathrm{Z}_y\right] \\
&= \sup_{f \in \mathcal{V}} \mathrm{E}_{x,n,z \sim P_{\mathrm{X}_y, \tilde{\mathrm{N}}, \mathrm{Z}_y}}\left[\log \frac{f[z](n)}{P_{\tilde{\mathrm{N}}}(n)}\right] \\
&\geq \mathrm{E}_{x,n,z \sim P_{\mathrm{X}_y, \tilde{\mathrm{N}}, \mathrm{Z}_y}}\left[\log \frac{g[z](n)}{P_{\tilde{\mathrm{N}}}(n)}\right] && \text{Use } g \\
&= \mathrm{E}_{x,n \sim P_{\mathrm{X}_y, \tilde{\mathrm{N}}}}\left[\sum_{z \in \mathcal{Z}_y^c} P_{\mathrm{Z}_y \mid \mathrm{X}_y}(z|x) \log \frac{g[z](n)}{P_{\tilde{\mathrm{N}}}(n)}\right] \\
&\quad + \mathrm{E}_{x,n \sim P_{\mathrm{X}_y, \tilde{\mathrm{N}}}}\left[\sum_{z \in \mathcal{Z}_y^w} P_{\mathrm{Z}_y \mid \mathrm{X}_y}(z|x) \log \frac{g[z](n)}{P_{\tilde{\mathrm{N}}}(n)}\right] && \mathcal{Z}_y^c := \mathcal{Z}_y \setminus \mathcal{Z}_y^w \\
&= 0 + \mathrm{E}_{x,n \sim P_{\mathrm{X}_y, \tilde{\mathrm{N}}}}\left[\sum_{z \in \mathcal{Z}_y^w} P_{\mathrm{Z}_y \mid \mathrm{X}_y}(z|x) \log \frac{g[z](n)}{P_{\tilde{\mathrm{N}}}(n)}\right] && \forall z \in \mathcal{Z}_y^c, g[z] = P_{\tilde{\mathrm{N}}}(n) \\
&= \sum_{x \in \mathcal{X}_y^t} \sum_{z \in \mathcal{Z}_y^w} P_{\mathrm{Z}_y, \mathrm{X}_y}(z,x) \log \frac{g[z](1)}{P_{\tilde{\mathrm{N}}}(1)}
\end{aligned}
$$

$$+ \sum_{x \in \mathcal{X}_y^e} \sum_{z \in \mathcal{Z}_y^w} P_{Z_y, X_y}(z, x) \log \frac{g(z)[0]}{P_{\tilde{N}}(0)} \qquad\qquad \text{Def. } \tilde{N} \text{ and } \mathcal{X}_y^e = \mathcal{X}_y \setminus \mathcal{X}_y^t$$

$$= 0 + \sum_{x \in \mathcal{X}_y^e} \sum_{z \in \mathcal{Z}_y^w} P_{Z_y, X_y}(z, x) \log \frac{g(z)[0]}{P_{\tilde{N}}(0)} \qquad\qquad \hat{R}(f', Z; \mathcal{D}) = 0$$

$$> 0 \qquad\qquad\qquad\qquad {\color{orange}\forall z \in \mathcal{Z}_y^w, g[z](0) > P_{\tilde{N}}(0)}$$

Where the penultimate line comes form the fact that all train example $x \in \mathcal{X}_y^t$ must get encoded to some $z \in \mathcal{Z}_y^c$ as the empirical risk of $f'$ is 0. We conclude that $I\left[Z_y \to \tilde{N}\right] \neq 0$, so by Lemma 5, $Z \notin \mathcal{M}_{\mathcal{V}}$ which concludes the proof for the binary case.

For the multi-classification setting, the same proof holds by taking $f'$ and effectively reducing it to a binary classifier. This is possible by starting from $f'$ and monotonically biasing it to construct $f'_{binary}$ which predicts with zero probability for all but two labels $y, y' \in \mathcal{Y}$. One of those labels (say $y$) has to be the correct label (to ensure that $f'_{binary}$ still reaches 0 empirical risk), while the other $y'$ can be any label s.t. $\exists z \in \mathcal{Z}_y$ with $f'[z](y') \neq 0$. Such a $y'$ always exists as the risk of $f'$ is not 0. Due to the monotonic biasing, $\forall z \in \mathcal{Z}_y$ we have $f'_{binary}[z](y) \geq f'[z](y)$ and $f'_{binary}[z](y') \geq f'[z](y')$. As a result $f'_{binary}$ still reaches 0 empirical risk but non zero actual risk, it can thus be used to construct the desired $g$ as before.

$\square$

### C.4  $\mathcal{V}$-minimal $\mathcal{V}$-sufficient Properties

In this section we prove Prop. 2. We first show that universal sufficient representation corresponds to the subset of sufficient representations that t.v.i. in the domain of predictors in $\mathcal{U}$.

**Lemma 6** (Recoverability of sufficiency). Let $\mathcal{U}$ be the universal family, then $\mathcal{S}_{\mathcal{U}} = \mathcal{S} \cap \mathcal{Z}$

*Proof.* In the following we abuse notation by using $Z \in \mathcal{Z}$ to denote that $Z$ t.v.i. $\mathcal{Z}$. Let us denote $\Omega = \bigcup \mathcal{Z}$ as the set of all possible finite sample spaces. From the recoverability property of $\mathcal{V}$-information, we know that $I_{\mathcal{U}}[Z \to Y] = I[Y; Z]$ so $\arg\max_{Z \in \mathcal{Z}} I_{\mathcal{U}}[Z \to Y] = \arg\max_{Z \in \mathcal{Z}} I[Y; Z]$. Suppose that $\mathcal{S} \cap \mathcal{Z}$ is non empty, then $\mathcal{S} \cap \mathcal{Z} = (\arg\max_{Z \in \Omega} I[Y; Z]) \cap \mathcal{Z} = \arg\max_{Z \in \mathcal{Z}} I[Y; Z] = \mathcal{S}_{\mathcal{U}}$ as desired. To show that $\mathcal{S} \cap \mathcal{Z}$ is indeed non empty, notice that one can effectively learn the sufficient representation $Z = Y$ due to the assumption $|\mathcal{Y}| \leq |\mathcal{Z}|$ (when $|\mathcal{Y}| < |\mathcal{Z}|$, restrict the support to $\mathcal{Y}$). $\square$

Let us characterize the set of minimal sufficient representations in terms of independence.

**Lemma 7** (Characterization of Minimal Sufficient Representations). Suppose $Y$ is a deterministic labeling $t(X)$, then the set of minimal sufficient representations $\mathcal{M}$ correspond to $\{Z \in \mathcal{S} | \ Z \perp X|Y \}$.

*Proof.*

$$\mathcal{M} := \arg\min_{Z \in \mathcal{S}} I[X; Z]$$
$$= \arg\min_{Z \in \mathcal{S}} I[X; Z] + I[Z; Y|X] \qquad\qquad Y - X - Z$$
$$= \arg\min_{Z \in \mathcal{S}} I[X, Y; Z] \qquad\qquad \text{Chain Rule}$$
$$= \arg\min_{Z \in \mathcal{S}} I[X; Z|Y] + I[Z; Y] \qquad\qquad \text{Chain Rule}$$
$$= \arg\min_{Z \in \mathcal{S}} I[X; Z|Y] \qquad\qquad \text{Sufficiency}$$

The last line uses the fact that $I[Z; Y]$ is a constant as the optimization is contrained to sufficient $Z \in \mathcal{S}$. Notice that $I[X; Z|Y] \geq 0$ and we know that it can reach zero when the labels are deterministic, for example with $Z = Y$. So $\mathcal{M} = \{Z \in \mathcal{S} | I[Y; Z|Y] = 0\} = \{Z \in \mathcal{S} | \ Z \perp X|Y \}$ which concludes the proof. $\square$

Similarly, let us characterize $\mathcal{U}$-minimal $\mathcal{U}$-sufficient representations in terms of independence.

**Lemma 8** (Characterization of $\mathcal{U}$-Minimal $\mathcal{U}$-Sufficient Representations). Suppose Y is a deterministic labeling $t(X)$, then $\mathcal{U}$-minimal $\mathcal{U}$-sufficient representations $\mathcal{M}_{\mathcal{U}}$ correspond to $\{Z \in \mathcal{S}_{\mathcal{U}}| \, \forall y \in \mathcal{Y}, \forall N \in \text{Dec}(X, y), \ Z \perp N \ \}$

*Proof.* Because of Lemma 5, $\forall y \in \mathcal{Y}, \forall N \in \text{Dec}(X, y)$ we have $I_{\mathcal{V}}[Z_y \to N] = 0$. As $\mathcal{U}$-information recovers MI that means $N \perp Z_y$. So $\mathcal{M}_{\mathcal{U}} = \{Z \in \mathcal{S}_{\mathcal{U}}| \, \forall y \in \mathcal{Y}, \forall N \in \text{Dec}(X, y), \ Z \perp N \ \}$ as desired. $\square$

**Lemma 9** (Monotonicity of $\mathcal{V}$-information). Let $\mathcal{V} \subseteq \mathcal{V}^+$ be two predictive families, then $\forall Z$ t.v.i $\mathcal{Z}$ and $\forall Y$ t.v.i. $\mathcal{Y}$ we have $I_{\mathcal{V}^+}[Z \to Y] \leq I_{\mathcal{V}}[Z \to Y]$.

*Proof.* Let us start from the monotonicity of the predictive entropy [23], which comes directly from the fact that we are optimizing over a larger functional family:

$$H_{\mathcal{G}}[Y|Z] \geq H_{\mathcal{G}}[Y|Z] \qquad \qquad \text{Monotonicity } \mathcal{V}\text{-ent.}$$
$$H[Y] - H_{\mathcal{G}}[Y|Z] \leq H[Y] - H_{\mathcal{G}}[Y|Z]$$
$$I_{\mathcal{G}}[Z \to Y] \leq I_{\mathcal{V}}[Z \to Y] \qquad \qquad \text{Prop. 5}$$

$\square$

**Proposition 2.** Let $\mathcal{V} \subseteq \mathcal{V}^+$ be two families and $\mathcal{U}$ the universal one. If labels are deterministic:

- **Recoverability** The set of $\mathcal{U}$-minimal $\mathcal{U}$-sufficient representations corresponds to the minimal sufficient representations that t.v.i. in the domain of $\mathcal{U}$, i.e., $\mathcal{M}_{\mathcal{U}} = \mathcal{M} \cap \mathcal{Z}$.

- **Monotonicity** $\mathcal{V}^+$-minimal $\mathcal{V}$-sufficient representations are $\mathcal{V}$-minimal $\mathcal{V}$-sufficient, i.e., $\arg\max_{Z \in \mathcal{S}_{\mathcal{V}}} I_{\mathcal{V}^+}[Z \to \text{Dec}(X, \mathcal{Y})] \subseteq \mathcal{M}_{\mathcal{V}}$.

- **Characterization** $Z \in \mathcal{M}_{\mathcal{V}} \iff Z \in \mathcal{S}_{\mathcal{V}}$ and $I_{\mathcal{V}}[Z \to \text{Dec}(X, \mathcal{Y})] = 0$.

- **Existence** At least one $\mathcal{U}$-minimal $\mathcal{U}$-sufficient representation always exists, i.e., $|\mathcal{M}_{\mathcal{V}}| > 0$.

*Proof.*

**Existence** In Eq. (9) we show how to construct such a $Z \in \mathcal{M}_{\mathcal{V}}$.

**Characterization** As $I_{\mathcal{V}}[Z \to \text{Dec}(X, \mathcal{Y})]$ is an average over a non negative (positivity of $\mathcal{V}$-information) $I_{\mathcal{V}}[Z_y \to N]$ it is equal to zero if and only if all the terms are zero: $I_{\mathcal{V}}[Z \to \text{Dec}(X, \mathcal{Y})] = 0 \iff \forall y \in \mathcal{Y}, \forall N \in \text{Dec}(X, y)$ we have $I_{\mathcal{V}}[Z_y \to N] = 0$. We conclude the proof using Lemma 5.

**Monotonicty** Following the same steps as the proof in ( $\implies$ ) of Lemma 5, we get that $\mathcal{V}^+$-minimal $\mathcal{V}$-sufficient representation implies $\forall y \in \mathcal{Y}, \forall N \in \text{Dec}(X, y)$ we have $I_{\mathcal{V}^+}[Z_y \to N] = 0$. Using the monotonocity of $\mathcal{V}$-information in our setting (Lemma 9) we have $0 = I_{\mathcal{V}^+}[Z_y \to N] \leq I_{\mathcal{V}}[Z_y \to N]$. As $\mathcal{V}$-information is always positive, we conclude that $\forall y \in \mathcal{Y}, \forall N \in \text{Dec}(X, y), I_{\mathcal{V}}[Z_y \to N] = 0$. By Lemma 5, we conclude that Z is $\mathcal{V}$-minimal $\mathcal{V}$-sufficient as desired.

**Recoverability** Using Lemma 6 we know that $\mathcal{S}_{\mathcal{U}} = \mathcal{S} \cap \mathcal{Z}$ so the domain of optimization for minimality and $\mathcal{U}$-minimality is the same. Using Lemma 7 and Lemma 8 we only need to show that $\forall Z \in \mathcal{S}_{\mathcal{U}}$ we have $Z \perp X|Y \iff \forall y \in \mathcal{Y}, \forall N \in \text{Dec}(X, y)$ we have $N \perp Z_y$. As a reminder, $Z_y$ and $X_y$ have distribution $P_{X_y} = P_{X|y}$, $P_{Z_y} = P_{Z|y}$ and $P_{X_y, z_y} = P_{X, Z|y}$ by definition.

( $\implies$ ) Starting from minimality $Z \perp X|Y$ so $\forall y \in \mathcal{Y}$ we have $P_{X_y, Z_y} = P_{X, Z|y} = P_{X|y} * P_{Z|y} = P_{X_y} * P_{Z_y}$ from which we conclude that $Z_y \perp X_y$. As $N = t'(X_y)$, for all $N \in \text{Dec}(X, y)$ we have $Z_y \perp N$ as desired.

( $\Longleftarrow$ ) Let us prove it by contrapositive. I.e. we show that $Z \not\perp X|Y \implies \exists y \in \mathcal{Y}, \exists N \in \text{Dec}(X, y)$ s.t. $N \not\perp Z_y$. As $Z \not\perp X|Y$ then $\exists \tilde{x} \in \mathcal{X}, \tilde{y} \in \mathcal{Y}$ s.t. $P_{Z|X,Y}(\cdot \,|\, \tilde{x}, \tilde{y}) \neq P_{Z|Y}(\cdot \,|\, \tilde{y})$. Let us define $\tilde{N} = \mathbb{1}[X_{\tilde{y}} = \tilde{x}_{\tilde{y}}]$, as $\mathbb{1}[X_{\tilde{y}} = \cdot]$ is a deterministic function from $\mathcal{X} \to \mathcal{Y}$, we have $\tilde{N} \in \text{Dec}(X, \tilde{y})$. Then:

$$
\begin{aligned}
P_{Z_{\tilde{y}}|\tilde{N}}(\cdot \,|\, 1) &= P_{Z_{\tilde{y}}|X_{\tilde{y}}}(\cdot \,|\, \tilde{x}_{\tilde{y}}) && \text{Def. } \tilde{N} \\
&= P_{Z|X,Y}(\cdot \,|\, \tilde{x}, \tilde{y}) && \text{Def. } Z_{\tilde{y}} \\
&\neq P_{Z|Y}(\cdot \,|\, \tilde{y}) && \text{Def. } \tilde{x}, \tilde{y} \\
&= P_{Z_{\tilde{y}}} && \text{Def. } Z_{\tilde{y}}
\end{aligned}
$$

As $P_{Z_{\tilde{y}}|\tilde{N}}(\cdot \,|\, 1) \neq P_{Z_{\tilde{y}}}$ we conclude that $\exists \tilde{y} \in \mathcal{Y}, \exists N \in \text{Dec}(X, \tilde{y})$ s.t. $N \not\perp Z_{\tilde{y}}$ as desired.

$\square$

**Corollary 1.** Suppose Y is a deterministic labeling $t(X)$. Let $\mathcal{U}$ be the universal predictive family. Under the assumptions stated in Appx. B.1, we have that: if Z is a minimal sufficient representation of X for Y that t.v.i. $\mathcal{Z}$, then any ERM on any dataset will achieve zero risk, i.e.

$$
Z \in \mathcal{M} \cap \mathcal{Z} \implies \forall M \geq |\mathcal{Y}|, \, \mathcal{D} \overset{\text{i.i.d.}}{\sim} P_{X,Y}^M, \, \forall \hat{f} \in \hat{\mathcal{U}}(\mathcal{D}) \text{ we have } R(\hat{f}, Z) = 0
$$

*Proof.* First notice that $\mathcal{U}$ is unconstrained and thus satisfies the assumptions in Appx. B.2. As a result, Theorem 1 tells us that $Z \in \mathcal{M}_{\mathcal{U}} \implies \forall M \geq |\mathcal{Y}|, \, \mathcal{D} \overset{\text{i.i.d.}}{\sim} P_{X,Y}^M, \, \forall \hat{f} \in \hat{\mathcal{U}}(\mathcal{D})$ we have $R(\hat{f}, Z) = \min_Z \min_{f \in \mathcal{U}} R(f, Z)$. Because the labeling is deterministic we have by Prop. 6 that the best achievable risk is $\min_Z \min_{f \in \mathcal{U}} R(f, Z) = 0$. From the recoverability property of $\mathcal{V}$-minimality and $\mathcal{V}$-sufficiency (Prop. 2) we have $\mathcal{M} \cap \mathcal{Z} = \mathcal{M}_{\mathcal{U}}$ so $Z \in \mathcal{M}_{\mathcal{U}} \iff Z \in \mathcal{M}_{\mathcal{U}} \implies \forall M \geq |\mathcal{Y}|, \, \mathcal{D} \overset{\text{i.i.d.}}{\sim} P_{X,Y}^M, \, \forall \hat{f} \in \hat{\mathcal{U}}(\mathcal{D})$ we have $R(\hat{f}, Z) = \min_Z \min_{f \in \mathcal{U}} R(f, Z) = 0$ as desired. $\square$

## C.5 Estimation Bounds

In this section we will prove and formalize Prop. 3, namely that $\hat{\mathcal{L}}_{\text{DIB}}(\mathcal{D})$ estimates $\mathcal{L}_{\text{DIB}}$ with PAC-style guarantees. First, let us formalize $\mathcal{L}_{\text{DIB}}$ and $\hat{\mathcal{L}}_{\text{DIB}}(\mathcal{D})$ described respectively in Eq. (8) and Fig. 2a. For simplicity, in the following we will use $\text{TDec}(X, y) := \{t \,|\, N = t(X), \forall N \in \text{Dec}(X, y)\}$ to denote the labeling that gave rise to $\text{Dec}(X, y)$. For notational convenience we use the following shorthands throughout this section:

$$
H_{\mathcal{V}}^Z[\mathcal{T}(X_y)] := \frac{1}{|\text{Dec}(X, y)|} \sum_{t \in \text{TDec}(X,y)} H_{\mathcal{V}}[t(X_y) \,|\, Z_y] \tag{10}
$$

$$
\hat{H}_{\mathcal{V}}^Z[Y] := \inf_{f \in \mathcal{V}} \frac{1}{|\mathcal{D}|} \sum_{x,y \in \mathcal{D}} -\log f[z \sim P_{Z|x}](y) \tag{11}
$$

$$
\hat{H}_{\mathcal{V}}^Z[t(X_y)] := \inf_{f \in \mathcal{V}} \frac{1}{|\mathcal{D}_y|} \sum_{x \in \mathcal{D}_y} -\log f[z \sim P_{Z|x}](t(x)) \tag{12}
$$

$$
\hat{H}_{\mathcal{V}}^Z[\mathcal{T}(X_y)] := \frac{1}{K} \sum_{t \in \text{TDec}(X,y;K)} \hat{H}_{\mathcal{V}}^Z[t(X_y)] \tag{13}
$$

**Definition 6** (DIB). Let $\beta \in \mathbb{R}_{>0}$ be a hyper-parameter controlling the importance of $\mathcal{V}$-minimality. The $\beta$-DIB criterion for the encoder $P_{Z|X}$ and predictions of Y from X is:

$$
\begin{aligned}
\mathcal{L}_{\text{DIB}}(X, Y, P_{Z|X}; \beta) :=& -I_{\mathcal{V}}[Z \to Y] + \beta * I_{\mathcal{V}}[Z \to \text{Dec}(X, \mathcal{Y})] \\
=& -H[Y] + H_{\mathcal{V}}[Y \,|\, Z] \\
&+ \sum_{y \in \mathcal{Y}} \frac{\beta}{|\mathcal{Y}|} \sum_{t \in \text{TDec}(X,y)} (H[t(X_y)] - H_{\mathcal{V}}[t(X_y) \,|\, Z_y]) \\
=& (const) + H_{\mathcal{V}}[Y \,|\, Z] - \sum_{y \in \mathcal{Y}} \frac{\beta}{|\mathcal{Y}|} \sum_{t \in \text{TDec}(X,y)} H_{\mathcal{V}}[t(X_y) \,|\, Z_y]
\end{aligned}
$$

$$= (const) + \inf_{f \in \mathcal{V}} \mathrm{E}_{x,y \sim P_{\mathrm{X,Y}}} \left[ \mathrm{E}_{z \sim P_{\mathrm{Z} \mid x}} [-\log f[z](y)] \right]$$

$$- \left( \sum_{y \in \mathcal{Y}} \frac{\beta}{|\mathrm{Dec}(\mathrm{X},y)||\mathcal{Y}|} \sum_{t \in \mathrm{TDec}(\mathrm{X},y)} \inf_{f \in \mathcal{V}} \mathrm{E}_{x \sim P_{\mathrm{X}|y}} \left[ \mathrm{E}_{z \sim P_{\mathrm{Z} \mid x}} [-\log f[z](t(x))] \right] \right)$$

$$= (const) + \mathrm{H}_{\mathcal{V}}[\mathrm{Y} \mid \mathrm{Z}] - \frac{\beta}{|\mathcal{Y}|} \mathrm{H}_{\mathcal{V}}^{\mathrm{Z}}[\mathcal{T}(\mathrm{X}_y)]$$

Where $(const)$ does not depend on $P_{\mathrm{Z}|X}$.

The empirical DIB is very similar but: (i) uses $\mathcal{D}$ to estimate all expectations over $P_{\mathrm{X,Y}}$; (ii) uses a single sample from Bob's encoder $z \sim P_{\mathrm{Z} \mid x}$; (iii) estimates the average over $t \in \mathrm{TDec}(\mathrm{X},y)$ using $K$ samples $\mathrm{TDec}(\mathrm{X},y;K) := \{t_i\}_{i=1}^{K}$ where $\forall i = 1, \dots, K, t_i \sim \mathrm{Unif}(\mathrm{TDec}(\mathrm{X},y))$ .

**Definition 7** (Empirical DIB). Let $\beta \in \mathbb{R}_{>0}$ be a hyper-parameter controlling the importance of $\mathcal{V}$-minimality, $\mathcal{D} \overset{\mathrm{i.i.d.}}{\sim} P_{\mathrm{X,Y}}^{M}$ be a training set of $M$ i.i.d. input-output pairs $(x,y)$, and $K \in \mathbb{N}_{>0}$ denote the number of r.v. to sample from each $y$ decomposition of X. The empirical (under $\mathcal{D}, K$) $\beta$-DIB criterion for the encoder $P_{\mathrm{Z}|\mathrm{X}}$ and predictions of Y from X is:

$$\hat{\mathcal{L}}_{\mathrm{DIB}}(\mathcal{D}, P_{\mathrm{Z}|\mathrm{X}}; \beta, K) := (const) + \inf_{f \in \mathcal{V}} \frac{1}{M} \sum_{x,y \in \mathcal{D}} -\log f[z \sim P_{\mathrm{Z}|x}](y)$$

$$- \left( \sum_{y \in \mathcal{Y}} \frac{\beta}{K|\mathcal{Y}||\mathcal{D}_y|} \sum_{t \in \mathrm{TDec}(\mathrm{X},y;K)} \inf_{f \in \mathcal{V}} \sum_{x \in \mathcal{D}_y} -\log f[z \sim P_{\mathrm{Z}|x}](t(x)) \right)$$

$$= (const) + \hat{\mathrm{H}}_{\mathcal{V}}^{\mathrm{Z}}[\mathrm{Y}] - \frac{\beta}{|\mathcal{Y}|} \hat{\mathrm{H}}_{\mathcal{V}}^{\mathrm{Z}}[\mathcal{T}(\mathrm{X}_y)]$$

Where we use $z \sim P_{\mathrm{Z}|x}$ to denote that z is one sample from $P_{\mathrm{Z}|x}$, $(const)$ is the same constant as in Def. 6 , and $\mathcal{D}_y := \{x \mid (x,y) \in \mathcal{D}\}$ is the subset of input examples labeled $y$.

Prop. 3 says that despite the previous approximations, $\hat{\mathcal{L}}_{\mathrm{DIB}}(\mathcal{D}, P_{\mathrm{Z}|\mathrm{X}}; \beta, K)$ still inherits $\mathcal{V}$-information's PAC estimation guarantees. More formally:

**Proposition 7** (PAC Estimation Guarantees). Let $\mathfrak{R}_M(\log \circ \mathcal{V})$ denote the $M$-samples Rademacher complexity of $\log \circ \mathcal{V} := \{g \mid g(z,y) = \log f[z](y), \forall f \in \mathcal{V}\}$, X, Y, Z be r.v.s, $\mathcal{D} \overset{\mathrm{i.i.d.}}{\sim} P_{\mathrm{X,Y}}^{M}$ be a dataset of $M$ i.i.d. input-output pairs $(x,y)$, $\beta \in \mathbb{R}_{>0}$, and $K \in \mathbb{N}_{>0}$. Assume that $\forall f \in \mathcal{V}$, $\forall z \in \mathcal{Z}$, $\forall y \in \mathcal{Y}$ we have $|\log f[z](y)| \leq C$, then for any $\delta \in ]0,1[$, with probability at least $1 - \delta$ we have that the estimation error[12] err $:= |\mathcal{L}_{\mathrm{DIB}}(\mathrm{X}, \mathrm{Y}, P_{\mathrm{Z}|\mathrm{X}}; \beta) - \hat{\mathcal{L}}_{\mathrm{DIB}}(\mathcal{D}, P_{\mathrm{Z}|\mathrm{X}}; \beta, K)|$ is bounded by:

$$\mathrm{err} \leq 2\mathfrak{R}_M(\log \circ \mathcal{V}) + \beta \log |\mathcal{Y}| + C\sqrt{\frac{2 \log \frac{1}{\delta}}{M}} \tag{14}$$

In order to prove Prop. 7, we need two key lemmas, one for PAC-estimation guarantees of the $\mathcal{V}$-sufficiency term and the other for estimation bounds of the $\mathcal{V}$-minimality term.

### C.5.1 Lemmas for Estimation Bounds

The estimation guarantees that we will use for the $\mathcal{V}$-sufficiency term essentially comes from Lemma 3 of Xu et al. [23], which we state here with a slight modification to incorporate the sampling from an encoder.

**Lemma 10** (Estimation error $\mathcal{V}$-sufficiency; Lemma 3 of Xu et al. [23]). Let $\mathcal{D} \overset{\mathrm{i.i.d.}}{\sim} P_{\mathrm{X,Y}}^{M}$ be a dataset of $M$ input-output pairs. Assume that $\forall f \in \mathcal{V}$, $\forall z \in \mathcal{Z}$, $\forall y \in \mathcal{Y}$ we have $|\log f[z](y)| \leq C$, then

for any $\delta \in ]0, 1[$, with probability at least $1 - \delta$, we have:

$$\left| \mathrm{H}_\mathcal{V}[\mathrm{Y} \mid \mathrm{Z}] - \hat{\mathrm{H}}_\mathcal{V}^\mathrm{Z}[\mathrm{Y}] \right| \leq \mathfrak{R}_M(\log \circ \mathcal{V}) + C\sqrt{\frac{2\log\frac{1}{\delta}}{M}} \tag{15}$$

*Proof.* For the bulk of the proof, we refer the reader to the proof in Xu et al. [23] which uses the standard Rademacher machinery (McDiarmid's inequality and a symmetrization argument for Rademacher random variables) to prove that with probability at least $1 - \delta$, we have:

$$\left| \mathrm{H}_\mathcal{V}[\mathrm{Y} \mid \mathrm{Z}] - \inf_{f \in \mathcal{V}} \frac{1}{|\mathcal{D}|} \sum_{z,y \in \mathcal{D}_{z,y}} - \log f[z](y) \right| \leq \mathfrak{R}_M(\log \circ \mathcal{V}) + C\sqrt{\frac{2\log\frac{1}{\delta}}{M}}$$

The only difference with Eq. (15) is that the dataset $\mathcal{D}_{z,y}$ consist of i.i.d. samples from $P_{\mathrm{Z,Y}}$, while our $\mathcal{D}$ consists of samples from $P_{\mathrm{X,Y}}$. Sampling a pair $(x, y) \sim P_{\mathrm{X,Y}}$ and then a single from $z \in P_{\mathrm{Z}|x}$ is nevertheless equivalent to sampling directly from $(x, y, z) \sim P_{\mathrm{X,Y,Z}}$ so $x, y \in \mathcal{D}$ and $z \sim P_{\mathrm{Z}|x}$ in Eq. (11) can be replaced by $z, y \in \mathcal{D}_{z,y}$ to get the desired Eq. (15). $\square$

We now provide a bound on the error of the $\mathcal{V}$-minimality.

**Lemma 11** (Estimation error $\mathcal{V}$-minimality). $\mathcal{D} \overset{\text{i.i.d.}}{\sim} P_{\mathrm{X,Y}}^M$ be a dataset of $M$ i.i.d. input-output pairs $(x, y)$, $\beta \in \mathbb{R}_{>0}$, and $K \in \mathbb{N}_{>0}$. We have:

$$\frac{\beta}{|\mathcal{Y}|} \sum_{y \in \mathcal{Y}} | \mathrm{H}_\mathcal{V}^\mathrm{Z}[\mathcal{T}(\mathrm{X}_y)] - \hat{\mathrm{H}}_\mathcal{V}^\mathrm{Z}[\mathcal{T}(\mathrm{X}_y)] | \leq \beta \log |\mathcal{Y}| \tag{16}$$

*Proof.* Suppose that at $y = \arg\max_{y \in \mathcal{Y}}(\mathrm{H}_\mathcal{V}^\mathrm{Z}[\mathcal{T}(\mathrm{X}_y)] - \hat{\mathrm{H}}_\mathcal{V}^\mathrm{Z}[\mathcal{T}(\mathrm{X}_y)])$ we have $\mathrm{H}_\mathcal{V}^\mathrm{Z}[\mathcal{T}(\mathrm{X}_y)] > \hat{\mathrm{H}}_\mathcal{V}^\mathrm{Z}[\mathcal{T}(\mathrm{X}_y)]$ then:

$$
\begin{aligned}
errMin &:= \frac{\beta}{|\mathcal{Y}|} \sum_{y \in \mathcal{Y}} | \mathrm{H}_\mathcal{V}^\mathrm{Z}[\mathcal{T}(\mathrm{X}_y)] - \hat{\mathrm{H}}_\mathcal{V}^\mathrm{Z}[\mathcal{T}(\mathrm{X}_y)] | & \\
&\leq \beta \max_{y \in \mathcal{Y}} | \mathrm{H}_\mathcal{V}^\mathrm{Z}[\mathcal{T}(\mathrm{X}_y)] - \hat{\mathrm{H}}_\mathcal{V}^\mathrm{Z}[\mathcal{T}(\mathrm{X}_y)] | & \text{Max > Mean} \\
&= \beta \max_{y \in \mathcal{Y}} (\mathrm{H}_\mathcal{V}^\mathrm{Z}[\mathcal{T}(\mathrm{X}_y)] - \hat{\mathrm{H}}_\mathcal{V}^\mathrm{Z}[\mathcal{T}(\mathrm{X}_y)]) & \text{Assumption} \\
&\leq \beta \max_{y \in \mathcal{Y}} \mathrm{H}_\mathcal{V}^\mathrm{Z}[\mathcal{T}(\mathrm{X}_y)] & \text{Non negativity} \\
&= \beta \max_{y \in \mathcal{Y}} \frac{1}{|\mathrm{Dec}(\mathrm{X}, y)|} \sum_{t \in \mathrm{TDec}(\mathrm{X}, y)} \mathrm{H}_\mathcal{V}[t(\mathrm{X}_y) \mid \mathrm{Z}_y] & \text{Eq. (10)} \\
&\leq \beta \max_{y \in \mathcal{Y}} \max_{t \in \mathrm{TDec}(\mathrm{X}, y)} \mathrm{H}_\mathcal{V}[t(\mathrm{X}_y) \mid \mathrm{Z}_y] & \text{Max > Mean} \\
&= \beta \max_{y \in \mathcal{Y}} \max_{t \in \mathrm{TDec}(\mathrm{X}, y)} \inf_{f \in \mathcal{V}} \mathbb{E}_{x \sim P_{\mathrm{X}|y}} \left[ \mathbb{E}_{z \sim P_{\mathrm{Z}|x}} [- \log f[z](t(x))] \right] & \text{Def.} \\
&\leq \beta \max_{y \in \mathcal{Y}} \max_{t \in \mathrm{TDec}(\mathrm{X}, y)} \mathbb{E}_{x \sim P_{\mathrm{X}|y}} \left[ \mathbb{E}_{z \sim P_{\mathrm{Z}|x}} [- \log P_{t(\mathrm{X}_y)}(t(x))] \right] & f[\cdot] = P_{t(\mathrm{X}_y)} \\
&= \beta \max_{y \in \mathcal{Y}} \max_{t \in \mathrm{TDec}(\mathrm{X}, y)} \mathbb{E}_{x \sim P_{\mathrm{X}|y}} \left[ - \log P_{t(\mathrm{X}_y)}(t(x)) \right] & \mathbb{E}[const] \\
&\leq \beta \max_{y \in \mathcal{Y}} \max_{t \in \mathrm{TDec}(\mathrm{X}, y)} \log \mathbb{E}_{x \sim P_{\mathrm{X}|y}} \left[ \frac{1}{P_{t(\mathrm{X}_y)}(t(x))} \right] & \text{Jensen's Ineq.} \\
&\leq \beta \max_{y \in \mathcal{Y}} \max_{t \in \mathrm{TDec}(\mathrm{X}, y)} \log \sum_{n=1}^{|\mathcal{Y}|} P_\mathrm{N}(n) \frac{1}{P_\mathrm{N}(n)} & t(\mathrm{X}_y) = \mathrm{N} \\
&= \beta \max_{y \in \mathcal{Y}} \max_{t \in \mathrm{TDec}(\mathrm{X}, y)} \log |\mathcal{Y}| & \\
&= \beta \log |\mathcal{Y}| &
\end{aligned}
$$

The fourth line uses the non-negativity of $\mathcal{V}$-entropy in the finite sample setting which can be shown using the non-negativity of entropy and the monotonicity of $\mathcal{V}$-entropy. The fact that $\exists f \in \mathcal{V}$, s.t. $\forall z \in \mathcal{Z}$ we have $f[z] = P_{t(\mathrm{X}_y)}$ comes from the arbitrary biasing assumption of $\mathcal{V}$. The third to last line uses the fact that $t(\mathrm{X}_y)$ t.v.i. the co-domain of $t$ which is $\mathcal{Y}$.

In the above we assumed that $\mathrm{H}_{\mathcal{V}}^{Z}[\mathcal{T}(\mathrm{X}_y)] > \hat{\mathrm{H}}_{\mathcal{V}}^{Z}[\mathcal{T}(\mathrm{X}_y)]$ at the arg max $y$. When $\mathrm{H}_{\mathcal{V}}^{Z}[\mathcal{T}(\mathrm{X}_y)] \leq \hat{\mathrm{H}}_{\mathcal{V}}^{Z}[\mathcal{T}(\mathrm{X}_y)]$, we have:

$$\begin{aligned}
errMin &\leq \beta \max_{y \in \mathcal{Y}} |\, \mathrm{H}_{\mathcal{V}}^{Z}[\mathcal{T}(\mathrm{X}_y)] - \hat{\mathrm{H}}_{\mathcal{V}}^{Z}[\mathcal{T}(\mathrm{X}_y)]\,| && \text{Max > Mean} \\
&= \beta \max_{y \in \mathcal{Y}} (\mathrm{H}_{\mathcal{V}}^{Z}[\mathcal{T}(\mathrm{X}_y)] - \hat{\mathrm{H}}_{\mathcal{V}}^{Z}[\mathcal{T}(\mathrm{X}_y)]) && \text{Assumption} \\
&\leq \beta \max_{y \in \mathcal{Y}} \hat{\mathrm{H}}_{\mathcal{V}}^{Z}[\mathcal{T}(\mathrm{X}_y)] && \text{Non negativity} \\
&= \beta \log |\mathcal{Y}|
\end{aligned}$$

Where we get the last line by applying the same steps as before to bound $\hat{\mathrm{H}}_{\mathcal{V}}^{Z}[\mathcal{T}(\mathrm{X}_y)]$ instead of $\mathrm{H}_{\mathcal{V}}^{Z}[\mathcal{T}(\mathrm{X}_y)]$.

$\square$

Note that the latter bound is loose and is not a PAC-style bound. To derive a tighter PAC-style bound one can use the fact that each $\mathrm{H}_{\mathcal{V}}[t(\mathrm{X}_y) \mid Z_y]$ term in $\mathrm{H}_{\mathcal{V}}^{Z}[\mathcal{T}(\mathrm{X}_y)]$ is $\frac{C}{\sqrt{M}}$-sub-Gaussian due to Lemma 10. We do not provide such bounds as the current looser bounds are more succinct and sufficient to show that $\mathcal{L}_{\mathrm{DIB}}$ is easier to estimate than $\mathcal{L}_{\mathrm{IB}}$ with finite samples.

### C.5.2  Proof for Estimation Bounds

We are now ready to prove Prop. 7

*Proof.* Due to the triangular inequality the error is:

$$\begin{aligned}
\mathrm{err} &:= \left| \mathcal{L}_{\mathrm{DIB}}(\mathrm{X}, \mathrm{Y}, P_{Z|X}; \beta) - \hat{\mathcal{L}}_{\mathrm{DIB}}(\mathcal{D}, P_{Z|X}; \beta, K) \right| \\
&= \left| (const) - (const) + \mathrm{H}_{\mathcal{V}}[\mathrm{Y} \mid Z] - \hat{\mathrm{H}}_{\mathcal{V}}^{Z}[\mathrm{Y}] - \frac{\beta}{|\mathcal{Y}|} \sum_{y \in \mathcal{Y}} \left( \mathrm{H}_{\mathcal{V}}^{Z}[\mathcal{T}(\mathrm{X}_y)] - \hat{\mathrm{H}}_{\mathcal{V}}^{Z}[\mathcal{T}(\mathrm{X}_y)] \right) \right| \\
&\leq |0| + \left| \mathrm{H}_{\mathcal{V}}[\mathrm{Y} \mid Z] - \hat{\mathrm{H}}_{\mathcal{V}}^{Z}[\mathrm{Y}] \right| + \frac{\beta}{|\mathcal{Y}|} \sum_{y \in \mathcal{Y}} \left| \mathrm{H}_{\mathcal{V}}^{Z}[\mathcal{T}(\mathrm{X}_y)] - \hat{\mathrm{H}}_{\mathcal{V}}^{Z}[\mathcal{T}(\mathrm{X}_y)] \right| && (17)
\end{aligned}$$

We can now compute the probability of not being approximately correct:

$$\begin{aligned}
\overline{\mathrm{PAC}} &:= \mathbb{P}\left( \mathrm{err} > 2\mathfrak{R}_M(\log \circ \mathcal{V}) + \beta \log |\mathcal{Y}| + C\sqrt{\frac{2 \log \frac{1}{\delta}}{M}} \right) \\
&\leq \mathbb{P}\left( \left| \mathrm{H}_{\mathcal{V}}[\mathrm{Y} \mid Z] - \hat{\mathrm{H}}_{\mathcal{V}}^{Z}[\mathrm{Y}] \right| + \frac{\beta}{|\mathcal{Y}|} \sum_{y \in \mathcal{Y}} \left| \mathrm{H}_{\mathcal{V}}^{Z}[\mathcal{T}(\mathrm{X}_y)] - \hat{\mathrm{H}}_{\mathcal{V}}^{Z}[\mathcal{T}(\mathrm{X}_y)] \right| \right. \\
&\qquad \left. > 2\mathfrak{R}_M(\log \circ \mathcal{V}) + \beta \log |\mathcal{Y}| + C\sqrt{\frac{2 \log \frac{1}{\delta}}{M}} \right) && Eq.\ (17) \\
&\leq \mathbb{P}\left( \left( \left| \mathrm{H}_{\mathcal{V}}[\mathrm{Y} \mid Z] - \hat{\mathrm{H}}_{\mathcal{V}}^{Z}[\mathrm{Y}] \right| > 2\mathfrak{R}_M(\log \circ \mathcal{V}) + C\sqrt{\frac{2 \log \frac{1}{\delta}}{M}} \right) \right.
\end{aligned}$$

$$\vee \left( \frac{\beta}{|\mathcal{Y}|} \sum_{y \in \mathcal{Y}} \left| \mathrm{H}_{\mathcal{V}}^{\mathrm{Z}}[\mathcal{T}(\mathrm{X}_y)] - \hat{\mathrm{H}}_{\mathcal{V}}^{\mathrm{Z}}[\mathcal{T}(\mathrm{X}_y)] \right| > \beta \log |\mathcal{Y}| \right) \right)$$

$$\leq \mathbb{P}\left( \left| \mathrm{H}_{\mathcal{V}}[\mathrm{Y} \mid \mathrm{Z}] - \hat{\mathrm{H}}_{\mathcal{V}}^{\mathrm{Z}}[\mathrm{Y}] \right| > 2\mathfrak{R}_M(\log \circ \mathcal{V}) + C\sqrt{\frac{2 \log \frac{1}{\delta}}{M}} \right)$$

$$+ \mathbb{P}\left( \frac{\beta}{|\mathcal{Y}|} \sum_{y \in \mathcal{Y}} \left| \mathrm{H}_{\mathcal{V}}^{\mathrm{Z}}[\mathcal{T}(\mathrm{X}_y)] - \hat{\mathrm{H}}_{\mathcal{V}}^{\mathrm{Z}}[\mathcal{T}(\mathrm{X}_y)] \right| > \beta \log |\mathcal{Y}| \right) \qquad \text{Union Bound}$$

$$\leq \delta + 0 \qquad \text{Lemma 10 and Lemma 11}$$

So, as desired, the probability of being approximately correct is:

$$\mathbb{P}\left( \mathrm{err} \leq 2\mathfrak{R}_M(\log \circ \mathcal{V}) + \beta \log |\mathcal{Y}| + C\sqrt{\frac{2 \log \frac{1}{\delta}}{M}} \right) = 1 - \overline{\mathrm{PAC}} = 1 - \delta$$

$\square$

## D  Reproducibility

In this section we provide further details of the hyperparameters chosen for the various experiments in the main text. Unless stated otherwise, all the models are trained for 300 epochs, using Adam [99] as the optimizer, a learning rate of $5e - 5$, at every epoch we decay all learning rates by $(1/100)^{(1/300)}$ (so that the learning rate is decayed by $100$ during the entire training), a batch-size of 256, without data augmentation, and using 5 and 3 random seeds respectively for experiments in the main text and appendices. We checkpoint and use the model which achieves the smallest *training* loss for evaluation.[13] Activation functions are $\mathrm{LeakyReLU}(x) = \max(x, 0.01 * x)$ while other unspecified parameters are PyTorch [100] defaults. The code can also be found at `github.com/YannDubs/Mini_Decodable_Information_Bottleneck`.

### D.1  $\mathcal{V}$-Minimality $\mathcal{V}$-Sufficiency

Bob's encoder is a neural network which maps the input X to a mean $\mu_z$ and standard deviation $\sigma_z$ used to parametrize a multivariate normal distribution with diagonal Gaussian: $P_{\mathrm{Z}|\mathrm{X}} = \mathcal{N}(\mathrm{Z}; \mu_z, \mathrm{softplus}(\sigma_z - 5))$, where $\mathrm{softplus}(\cdot) = \log(1 + \exp(\cdot))$. Note that we use $-5$ as done in VIB [19] to make the methods more comparable. During training we sample a single $z \sim P_{\mathrm{Z}|\mathrm{X}}$, while we sample 12 during evaluation (as done in VIB [19]). [14] The representation then goes through a batch normalization layer without trainable parameters (setting the mean to 0 and standard deviation to 1), which ensures that the representation cannot diverge as discussed in Appx. E.3.

The encoder is trained using two losses which are weighted by a hyperparameter $\beta$, $\hat{\mathcal{L}}_{\mathrm{DIB}}(\mathcal{D}) = \mathcal{L}_{\mathcal{V}\mathrm{suff}} - \beta \mathcal{L}_{\mathcal{V}\mathrm{min}}$:

- $\mathcal{V}$-**sufficiency** $\mathcal{L}_{\mathcal{V}\mathrm{suff}}$. The representation $z$ goes through a head of architecture $\mathcal{V}$. The last layer of this head goes through a softmax to parametrize a distribution over of labels, i.e., $f[z](y)$ corresponds to the $y^{th}$ neuron in that layer. The resulting loss $\mathcal{L}_{\mathcal{V}\mathrm{suff}}$ is the standard cross entropy. We then back-propagate to jointly minimize the loss with respect to the head and the encoder.

- $\mathcal{V}$-**minimality** $\mathcal{L}_{\mathcal{V}\mathrm{min}}$. In addition to being used for $\mathcal{L}_{\mathcal{V}\mathrm{suff}}$, the representation $z$ is used as input to $\mathcal{V}$-minimality heads that each predict a different $\mathrm{N} \in \mathrm{Dec}(\mathrm{X}, y)$ in the same way as how the $\mathcal{V}$-sufficiency head predicts the label Y. We get each N using Algorithm 1, i.e.,

assigning each example $x \in \mathcal{X}$ some index and then performing base $|\mathcal{Y}|$ expansion (see Appx. E.5). For the case of CIFAR10 this corresponds to: (i) assigning each image some index between $0$ and the number of examples ($\sim 6000$); (ii) having 4 nuisance labels N corresponding to each digit of the new index, e.g., the cat number 627 will have $N_1 = 0$, $N_2 = 6, N_3 = 2, N_4 = 7$.

Each $\mathcal{V}$-minimality head predicts the corresponding N. Having to treat every example differently based on their underlying label $y$ ("for loop" in Fig. 2a) is not amenable to batch GPU training, which assumes that every example in a batch is treated the same way. We thus use the same predictor for a set $\{\mathrm{Dec}(X, y)\}_{y \in \mathcal{Y}}$ (see Appx. E.6), i.e., instead of having one predictor for $\mathrm{Dec}(X, \mathrm{cat})$ and another for $\mathrm{Dec}(X, \mathrm{dog})$ where representations are $z \sim Z_y$ (as shown in Appx. D) we use a *single* head to predict both $\mathrm{Dec}(X, \mathrm{cat}), \mathrm{Dec}(X, \mathrm{dog})$ using representations $z \in Z$ from cats or dogs as inputs. By taking an average over the loss of each head we get the $\mathcal{L}_{\mathcal{V}\min}$ term of Fig. 2a. Throughout the paper we unroll the optimization of $\mathcal{V}$-minimality heads for 5 steps, i.e., for every batch $\mathcal{L}_{\mathcal{V}\min}$) is *minimized* by $\mathcal{V}$-minimality heads while the encoder *maximizes* it. We show in Appx. E.2 that, as seen in Fig. 2b, DIB can perform similarly well with joint gradient ascent descent — by reversing gradients which is more efficient and easier to implement.

Once the encoder is trained, we can train Alice's classifier by:

- **Standard (Avg, ERM)**. We freeze the trained encoder and use the representations as inputs to Alices head of architecture $\mathcal{V}$. Alice then trains her classifier by minimizing the usual cross-entropy.

- **Worst ERM**. In some cases we want to explicitly find a Classifier from Alice that will perform well on train but bad on test. To do so, we optimize $\arg \min_{f \in V} \hat{\mathrm{R}}(f, Z; \mathcal{D}) - 0.1 *$ $\mathrm{R}(f, Z)$ (see Appx. E.7), which corresponds to minimizing the training cross-entropy while directly *maximizing* the *test* cross-entropy.

Finally, Alice's classifier is then evaluated by its test log loss (risk).

### D.1.1 $\mathcal{V}$-Sufficiency

For the $\mathcal{V}$-Sufficiency experiments (Sec. 4.1), we use a ResNet18 for Bob's encoder and a single-hidden layer MLP for Alice with varying width (see Appx. E.1 for details and justification). For Fig. 3a we use a 2 dimensional Z and odd-even classification of CIFAR100. For Fig. 3b we use a 8 dimensional Z and full CIFAR100. The encoder is trained to be $\mathcal{V}_{Bob}$-sufficient and so we do do not use $\mathcal{L}_{\mathcal{V}\min}$. Alice uses an architecture $\mathcal{V}_{Alice}$ and is trained using standard cross-entropy. To support Prop. 1 we want to show that $\mathcal{V}$-sufficient representations are optimal when Bob and Alice *have access to the entire underlying distribution*. As a result, we evaluate Alice's classifier on the *training* set.

### D.1.2 $\mathcal{V}$-Minimality $\mathcal{V}$-Sufficiency

For the $\mathcal{V}$-minimality experiments (Sec. 4.2), our goal is to show that if Bob trains $\mathcal{V}$-minimal $\mathcal{V}$-sufficient representations, any ERM trained by Alice will perform well on test (supporting Theorem 1).

Since our theory does not impose any limitation on the possible representations Z, we need an encoder that is very flexible and as close as possible to a universal function approximator. Thus, we use a large MLP with three hidden layers each with 2048 hidden units, for a total of around 21M parameters. Furthermore, we use a 1024 dimensional Z in order to avoid constraints arising from a dimensionality bottleneck rather than from the criterion that Bob uses to train Z. Alice's predictive family $\mathcal{V}$ is a single hidden layer MLP with 128 hidden units. As the encoder is much larger than $\mathcal{V}$ we increase the learning rate of $\mathcal{V}$-minimality heads by a factor of 50 to make sure that they can "keep up" with the changing encoder.

For Fig. 4a and Fig. 4b (Effect of DIB on generalization), we use the CIFAR10 dataset, and train Alice's classifier in the "Worst ERM" setting. The same holds for Fig. 4c, but uses the CIFAR10+MNIST dataset (see Appx. D.2). In this case, the Bob's encoder is still trained using only CIFAR10. Once the encoder is trained and frozen, we evaluate how well Alice's (worst) ERM can predict the CIFAR10 labels (as before). In addition, we also train another classifier in $\mathcal{V}$ to predict the MNIST labels, using the same encoder.

For Table 1 (performance of Avg. and worst ERM for different regularizers) we train the encoder in different ways, and Alice's classifier in the "Worst ERM" (top row) and "Avg. ERM" (bottom row) settings. Importantly, each regularizer is used only during Bob's training, as we are interested to know how DIB performs compared to other regularizers for representation learning, when the Alice's downstream classifier is an empirical risk minimizer as in our problem formulation (Sec. 2.1). The regularizers are as follows (we tuned all models): (i) "No Reg." does not use any regularizer and the encoder directly outputs the representation $z$ rather than a distribution from which to sample (this is the only such deterministic encoder in these results); (ii) "Stoch Rep." does not use any regularizer but the encoder is the same as the one used in DIB, i.e., it parametrizes a Gaussian distribution from which 12 $z$ are sampled and the predictions are marginalized over these samples; (iii) "Dropout" uses 50% dropout after every layer in the encoder and is kept when training Alice's encoder; (iv) "Wt. Dec." uses 1e-4 weight decay during Bob's training; (v) "VIB" uses a KL-divergence "regularizer" to force the parametrized to be closer to a standard normal distribution (as described in [19]), the weight of the regularizer $\beta = 1e-1$. (vi) "$\mathcal{V}^-$-DIB" uses a one hidden layer MLP with 2 hidden units (instead of 128) and $\beta = 100$. (vii) "$\mathcal{V}^+$-DIB" uses a one hidden layer MLP with 8192 hidden units (instead of 8192) and $\beta = 0.01$. (viii) "$\mathcal{V}$-DIB" uses the correct one hidden layer MLP with 128 hidden units and $\beta = 10$.

## D.2   CIFAR10+MNIST Dataset

We follow Achille and Soatto [20] and overlay MNIST digits on top of CIFAR10 images to create the CIFAR10+MNIST dataset. Concretely, we pick a CIFAR10 image and on top of it overlay an MNIST image selected uniformly at random. The code used to generate the dataset as well as some samples can be found in `https://github.com/YannDubs/Overlayed-Datasets`.

## D.3   Correlation

For the correlation experiments in Sec. 4.3, we largely follow previous work by Jiang et al. [50] in the sweeps over hyperparameters to get an initial set of models with potentially different generalization errors.

Let `Conv(kernel_size, stride)` denote a convolutional layer. The basic block of the convolutional networks consist of (in order): `Conv(3, 2, padding=1)`, `BatchNorm`, `Relu`, `Conv(1, 1)`, `BatchNorm`, `Relu`, `Conv(1,1)`, `Relu`, `dropout`. The final networks consist of one `Conv(1,1)`,`Relu` used to set the correct number of channels, followed by "depth" number of blocks, followed by `Conv(1,1)` that set the number of channels to the dimensionality of the representation, followed by an average pooling over the spatial dimensions. The resulting output is a (deterministic) representation which will go through an MLP with 2 hidden layers of width 128 to perform classification.

In order to be comparable to Jiang et al. [50] we sweep over the following hyperparameters: (i) the learning rate (1e-3, 3e-4, 1e-4); (ii) the batch size (32, 64, 128); (iii) the dropout rate (0, 0.25, 0.5); (iv) the width/channel size (192, 384, 768); (v) the depth/number of blocks (2, 4, 8); (vi) the dimensionality of the representation (32, 128, 512). We train every models with combination of these parameters on CIFAR10 and stop once the train log likelihood is better than 0.01 (with a hard stop at 300 epochs if the model did not reach that threshold by then). The resulting subset of 562 models thus all (approximately) perform equally well on training, which enables to study the effect of hyperparameters on generalization in isolation without the influence of performance on training as an indicator for generalization. For each resulting model we compute the difference between performance on train and test (generalization gap), both in terms of accuracy and log likelihood. We then compare the rank correlation between the desired measure and the observed generalization gap.

The methods that we compare to are the best performing in each section of Jiang et al. [50], namely: (i) "Entropy" is the average entropy of the predicted probabilities; (ii) "Path Norm" takes an input of all ones and passes it through the network where all the parameters are squared and returns the square root of the sumed logits; (iii) "Var. Grad." computes the average gradients at the end of training; (iv) "Sharp. Mag" essentially finds the maximum (relative) perturbation that can be applied to the weights to get less than 0.1 log likelihood difference. We use $\frac{1}{\alpha'}$ version of sharpness magnitude as described in Jiang et al. [50].

The code of Jiang et al. [50] is not (currently) public but we did our best to follow their work on all but the following three points: (i) We use an MLP after the CNN, which was used to evaluate $\mathcal{V}^+$,$\mathcal{V}^-$ minimality as in the rest of the paper, (ii) We do not sweep over the weight decay and optimizer but instead we vary the size of the representation, to try to incorporate a representation-specific parameter, (iii) our implementation of the sharpness magnitude measures differences in log-likelihood instead of accuracy. [15]

# E  Additional Experiments

In the rest of the appendices we provide additional experiments and results. Specifically we investigate: (i) How to obtain meaningful nested predictive families $\mathcal{V}^- \subset \mathcal{V} \subset \mathcal{V}^+$ which we use throughout our paper (Appx. E.1); (ii) Different methods to deal with the min-max optimization in $\mathcal{L}_{\mathrm{DIB}}$ (Appx. E.2); (iii) An important "trick" that helps the min-max optimization of $\mathcal{L}_{\mathrm{DIB}}$ (Appx. E.3); (iv) The effect of using Monte Carlo estimation of $\mathrm{I}_{\mathcal{V}}[Z \to \mathrm{Dec}(X, \mathcal{Y})]$ (Appx. E.4); (v) How to improve $\mathcal{L}_{\mathrm{DIB}}$ by sampling approximately independent nuisance r.v.s from $\mathrm{Dec}(X, y)$ (Appx. E.5); (vi) How to efficiently implement $\mathcal{L}_{\mathrm{DIB}}$ for standard GPU batch training (Appx. E.6); (vii) How to obtain an ERM which performs well on train but bad on test ("Worst ERM";Appx. E.7); (viii) Whether our $\mathcal{V} - sufficiency$ results in Sec. 4.1 hold across various settings (Appx. E.8); (ix) Why $\mathcal{V}$-sufficiency is not as important in large networks trained with SGD (Appx. E.9); (x) The effect of $\beta$ on different $\mathcal{V}$-minimality terms (Appx. E.10); (xi) The performance of DIB as a standard regularizer (Appx. E.11); (xii) Whether the degree of $\mathcal{V}$-minimality is correlated with generalization across different neural networks and datasets (Appx. E.12).

## E.1  Sweeping over Predictive Families

A core set of our experiments involve using nested predictive families $\mathcal{V}^- \subset \mathcal{V} \subset \mathcal{V}^+$. In this appendix, we study different ways of "sweeping" over functional families, i.e. finding some parameter s.t. increasing the value it can take $k < k'$ means increasing the family $\mathcal{V}_k \subset \mathcal{V}_{k'}$. Using neural networks with varying architectures, we investigate the following possibilities:

**Width** Sweeping over the width ($w = 4^k$) of a single layer MLP ($d = 1$).

**Depth** Sweeping over the depth ($d = k$) of an MLP ($w = 128$) [16].

**Width and Depth** Simultaneously sweeping over the depth ($d = k$) and width ($w = 32 * 2^k$) of an MLP.

**Weight Pruning** Sweeping over the percentage of non-pruned weights ($\%_{\neq 0} = \frac{2^k}{2^8}$) of an MLP ($d = 3$,$w = 2048$). [17]

To see whether the aforementioned methods are effective ways of sweeping over functional families, we analyze their respective complexity by looking at how well they can fit arbitrary labelling, which was proposed by [37] as a measure of complexity intuitively similar to Rademacher complexity. Specifically, we train a $\mathcal{X} - 1024 - 1024 - \mathcal{Z} - 64 - \mathcal{Y}$ MLP on CIFAR10, then freeze the encoder $\mathcal{X} - 1024 - 1024 - \mathcal{Z}$, shuffle the labels $P_{Z \times Y} = P_Z \times P_Y$, and compute the training log likelihood achieved by the predictive family $\min_{f \in \mathcal{V}} \hat{\mathrm{R}}(f, Z; \mathcal{D})$. We do so for increasing dimensionality (2,16,1024) of the representations Z.

Figure 6 shows that each of the four aforementioned sweeping methods increases their respective complexities (besides for two dimensional representations which appears constant). Sweeping only over depth does not appear to significantly increase the complexity of the functional family [18]. Sweeping over weight pruning fraction is a very effective method to increase the functional family.

(a) Width

(b) Depth

(c) Width and Depth

(d) Weight Pruning

Figure 6: Sweeping over functional families. Each plot shows how the complexity of a functional family (measured by its ability to learn arbitrary CIFAR10 labels from a $2, 16, 1024$ dimensional representation) increases by sweeping over the following properties of an MLP: (a) Width, (b) Depth, (c) Width and Depth ($width = 32 * 2^{depth}$), (d) Weight Pruning. The results in all panels are averaged over 3 runs with 95% bootstrap confidence interval.

Sweeping over the width and depth together is also very effective to increase the complexity of the family, but we found that in some experiments the deepest MLPs were too difficult to optimize. Sweeping over the width of the MLP increases the complexity significantly. This last method is simple and effective, so we decided to use it as the sweeping method in the main text.

### E.2   Min-Max Optimization

As mentioned in Sec. 3.3, optimizing the DIB involves a min-max procedure that is hard to optimize. In all our experiments, we optimize over Z by using variants of stochastic gradient descent (SGD) to learn the parameters of the encoder and thus require the gradient of the DIB objective Eq. (8) with respect to the encoding model's parameters. To see where the issues arise, we show how to compute the gradients $\frac{\partial}{\partial Z} \mathcal{L}_{\mathrm{DIB}}$[19]:

$$
\begin{aligned}
\frac{\partial}{\partial Z} \mathcal{L}_{\mathrm{DIB}} &= -\frac{\partial}{\partial Z} I_{\mathcal{V}}[Z \to Y] + \frac{\partial}{\partial Z} \beta * I_{\mathcal{V}}[Z \to \mathrm{Dec}(X, \mathcal{Y})] \\
&= -\frac{\partial}{\partial Z} I_{\mathcal{V}}[Z \to Y] \\
&\quad + \frac{\partial}{\partial Z} \sum_{y \in \mathcal{Y}} \frac{\beta}{|\mathcal{Y}| * |\mathrm{Dec}(X, y)|} \sum_{N \in \mathrm{Dec}(X, y)} I_{\mathcal{V}}[Z_y \to N] \\
&= -\frac{\partial}{\partial Z} H[Y] + \frac{\partial}{\partial Z} H_{\mathcal{V}}[Y \mid Z]
\end{aligned}
$$

Figure 7: Effect of taking multiple inner optimization steps (over $\mathcal{V}$) on Alice's generalization gap during the min-max optimization in DIB. The left figure shows the gap when higher order gradients are computed through the unrolled internal optimization. The right figure is a baseline showing the result of taking the same number of internal optimization steps but not tracking gradients through the internal optimization. An inner optimization of 0 means joint optimization using gradient reversing.

$$
\begin{aligned}
&+ \sum_{y \in \mathcal{Y}} \beta' \sum_{\mathrm{N} \in \mathrm{Dec}(\mathrm{X}, y)} \left( \frac{\partial}{\partial \mathrm{Z}} \mathrm{H}[\mathrm{Y}] - \frac{\partial}{\partial \mathrm{Z}} \mathrm{H}_{\mathcal{V}}[\mathrm{N} \mid \mathrm{Z}_y] \right) \\
&= \frac{\partial}{\partial \mathrm{Z}} \min_{f \in \mathcal{V}} \mathrm{R}^{(\mathrm{Y})}(f, \mathrm{Z}) - \sum_{y \in \mathcal{Y}} \beta' \sum_{\mathrm{N} \in \mathrm{Dec}(\mathrm{X}, y)} \frac{\partial}{\partial \mathrm{Z}} \min_{f \in \mathcal{V}} \mathrm{R}^{(\mathrm{N})}(f, \mathrm{Z}) \qquad \text{Lemma 2} \\
&= \left( \frac{\partial}{\partial \mathrm{Z}} \min_{f \in \mathcal{V}} \mathrm{R}^{(\mathrm{Y})}(f, \mathrm{Z}) + \sum_{y \in \mathcal{Y}} \beta' \sum_{\mathrm{N} \in \mathrm{Dec}(\mathrm{X}, y)} \frac{\partial}{\partial \mathrm{Z}} \max_{f \in \mathcal{V}} \mathrm{R}^{(\mathrm{N})}(f, \mathrm{Z}) \right) \qquad (18)
\end{aligned}
$$

Where we used $\mathrm{R}^{(\mathrm{Y})}$ and $\mathrm{R}^{(\mathrm{N})}$ to make it explicit that the risk terms are for different predictions. For the first term in Eq. (18), we follow the *de facto* method of computing gradients, i.e., to treat the problem simply as joint optimization over Z and $\mathcal{V}$. Complications arise, however, because the $\mathcal{V}$-minimality term involves a maximization, thus giving rise to a min (over the encoding) - max (over classifiers) optimization. There exist at least three ways of estimating such gradients:

**Exact** Assuming that we can perform the inner optimization exactly $f_{\mathrm{Y}}^* = \arg \min_{f \in \mathcal{V}} \mathrm{R}^{(\mathrm{Y})}(f, \mathrm{Z})$ and $f_{\mathrm{N}}^* = \arg \max_{f \in \mathcal{V}} \mathrm{R}^{(\mathrm{N})}(f, \mathrm{Z})$, then we we know by the Envelop theorem [101] that the gradients are simply:

$$
\frac{\partial}{\partial \mathrm{Z}} \mathrm{R}^{(\mathrm{Y})}(f_{\mathrm{Y}}^*, \mathrm{Z}) + \beta' \sum_{y \in \mathcal{Y}} \sum_{\mathrm{N} \in \mathrm{Dec}(\mathrm{X}, y)} \frac{\partial}{\partial \mathrm{Z}} \mathrm{R}^{(\mathrm{Y})}(f_{\mathrm{N}}^*, \mathrm{Z})
$$

This exact method is very restrictive, as we can essentially only find the optimal functions if we add strong restrictions on $\mathcal{V}$ (e.g. linear classifiers).

**Joint Optimization** One could disregard the issues that arise from min-max optimization and optimize everything jointly. This can easily be implemented by reversing the sign (sometimes referred to as a gradient reversal layer [102]). This is what we show in Fig. 2b. Note that there are no guarantees of convergence, even to a local minimum.

**Unrolling Optimization** A third possibility consists in "unrolling" the inner optimization [39, 103–105] by taking a few SGD steps in the internal optimization loop (over the functions $f$) and computing the gradients with respect to the Z. Note that there are again no guarantees of converging even to a local minimum. Nevertheless, the gradients are better estimates of the true gradients than in the joint case. A key hyper-parameter then becomes the number of inner optimization steps to perform for each $Z$ update.

We experimented with the three aforementioned approaches to estimating the gradients. While preliminary results suggested that the "Exact" method is, unsurprisingly, better than the two other

(a) Absolute Value of Mean of Z  (b) Standard Deviation of Z

Figure 8: Consequences of not performing the internal optimization to convergence on (a) the average (across batches) absolute value of the mean of Z; (b) the average (across batches) of the standard deviation of Z. In both cases we plot "DIB Free", which consists of the naive DIB, and "DIB" which uses our batch normalization solution.

methods, we did not want to restrict the function families and thus opted for the other two approaches. In the following, we compare two performance of the other methods that do not necessitate any restriction on $\mathcal{V}$. For all of experiments, we employ an additional trick that arises due to the inner optimization not being run until convergence (see Appx. E.3).

Figure 7 shows the effect of the number of inner optimization steps on Alice's generalization. We see that she achieves best performance by either joint optimization (which is noted as 0 inner optimization steps) or unrolling optimization with multiple inner optimization steps. Although it is not clear from Fig. 7 using 5 inner optimization steps is significantly better than performing joint optimization. Indeed, at the best $\beta$ ($\beta = 10$ for 5 inner steps, and $\beta = 1000$ for joint optimization) taking 5 inner steps gives an average test log likelihood of $-1.56 \pm 0.03$ against $-1.65 \pm 0.00$ for joint optimization. We also see that increasing the number of steps results in an objective which appears more robust to the choice of $\beta$. This comes, however, at the cost of increased computational complexity. Throughout the paper we use five inner optimization steps, but note that for larger problems, it would be advisable to use joint optimization in order to decrease the computational complexity.

### E.3 Diverging Representation from Min-Max

As discussed in Appx. E.2, the DIB objective requires a minimax optimization, which we solve using 5 steps of inner optimization. A major issue that arises from with this approach is that the encoder can "cheat" because the inner optimization is not done until convergence. As a result the encoder can learn representations Z that are highly variable such that the decoder $f \in \mathcal{V}$, which tries to predict N, cannot adapt quickly enough.

To solve this issue we pass the sampled representations through a batch normalization layer [106] but without trainable hyper-parameters, i.e. we normalize each batch of representations to have a mean of zero and a variance of one. As this is simply a rescaling, it could easily be learned by any $f \in \mathcal{V}$ if the inner optimization were performed until convergence (it does not modify $\mathcal{V}$). Nevertheless, it does give much better results since it ensures that the encoder learns a meaningful representation, rather than taking advantage of the limited number of steps in the internal maximization. Note that the encoder has many more parameters than the classifier, allowing it to alter the representation such that the classifier cannot "keep up". Figure 8 shows that without this "trick" the mean and standard deviation in fact diverges as $\beta$ increases (labeled DIB Free). This is solved by the normalization trick (labeled DIB), which we use throughout the paper.

### E.4 Monte Carlo Estimation of $I_{\mathcal{V}}[Z \rightarrow \text{Dec}(X, \mathcal{Y})]$

Optimizing DIB involves the task of minimizing $I_{\mathcal{V}}[Z \rightarrow \text{Dec}(X, \mathcal{Y})]$, which requires (Eq. (7)) computing an average over all Y decompositions of X — of which there are $|\mathcal{Y}|^{\frac{|\mathcal{X}|}{|\mathcal{Y}|}+1}$. As a result,

(a) Generalization Gap

(b) Terms in DIB

Figure 9: Effect of number of Monte Carlo Samples on (a) the worst case generalization gap; (b) the terms estimated by DIB. In both (a) and (b) the left plot show [4, 12, 20] Monte Carlo samples per label. The right plot (labeled DIB Same) is a baseline that always has four Monte Carlo samples, but uses [1, 3, 5] predictors with different decoders with different initializations each predicting the *same* N (for a total of [4, 12, 20] predictors as in the left plots). All other hyperparameters are the same as for Fig. 4.

even though the $\mathcal{V}$-information terms are sample efficient (due to the estimation bounds given in [23]), estimating it directly is not computationally efficient. To estimate the $I_{\mathcal{V}}[Z \rightarrow \mathrm{Dec}(X, \mathcal{Y})]$ in a computationally efficient manner, we thus perform a Monte Carlo estimation of the average (corresponding to `random_choice` in Fig. 2a). In this section we show that in practice, we only require a very small number of Monte Carlo samples, allowing DIB to be implemented in a computationally efficient manner.

Figure 9 shows the result of using a different number of Monte Carlo samples (4, 12, or 20 r.v.s N per label $y$). All other hyperparameters are identical to those used in Fig. 4. In order to ensure that the gains come from sampling different Ns rather than from using a larger number of predictors, we also trained a model (labeled "DIB Same") which always uses four N labelings multiple predictors per labeling to match the total number of different predictors. For example, "DIB Same" with 20 predictors corresponds to sampling four N and then having 5 predictors (different initializations) per N that each try to predict the same arbitrary labels N. Indeed, increasing the number of predictors (even with the same N) might help as each will converge to a different local minimum. We see that increasing the number of predictors does seem to have an effect on DIB, but the number of Monte Carlo estimates does not seem to change much compared with using more predictors. Interestingly, the best test log likelihood comes from using the *fewest* number of predictors.

This finding that the number of Monte Carlo samples has little effect on DIB might seem surprising as we only use four instead of the $|\mathcal{Y}|^{\frac{|\mathcal{X}|}{|\mathcal{Y}|}+1} \approx 10000$ different N. But it is important to notice that many of these Y decompositions of X are redundant (i.e. they contain the same $\mathcal{V}$-information).

**Input** : All possible inputs $\mathcal{X}$, labels Y associated with each $\mathcal{X}$, all possible labels $\mathcal{Y}$
**Output**: A matrix N, where the $i^{th}$ column is the value of $\mathrm{N}_i$ for the corresponding $\mathcal{X}$
indices $\leftarrow$ zeros($|\mathcal{X}|$)
Ns $\leftarrow$ zeros($|\mathcal{X}|$, $\lceil \log_{|\mathcal{Y}|}(|\mathcal{X}|) \rceil - 2$)
**for** $y \in \mathcal{Y}$ **do**
   | idcs[Y == y] $\leftarrow$ range(0, len(Y == y))
**end**
**for** $i \leftarrow 0$ **to** $|\mathcal{X}|$ **do**
   | Ns[i,:] $\leftarrow$ base $|\mathcal{Y}|$ expansion of idcs[i]
**end**

**Algorithm 1:** Y decomposition of X through base expansion

(a) Generalization Gap        (b) Terms in DIB

Figure 10: Effect of using Base $|\mathcal{Y}|$ expansion (labeled "DIB") vs. randomly selecting N from the set of Y decompositions of X (labeled "DIB Random") on (a) the worst case generalization gap; (b) the terms estimated by DIB. All other hyper-parameters are the same as for Fig. 4.

For example, due to the invariance of $\mathcal{V}$ to permutations, minimizing $I_\mathcal{V}[Z \to N]$ also minimizes $I_\mathcal{V}[Z \to \pi N]$, for all permutations $\pi$ on $\mathcal{Y}$. Generally speaking, the larger the functional family $\mathcal{V}$, the more that $N \in \mathrm{Dec}(X, y)$ will be redundant in that minimizing the $\mathcal{V}$-information with respect to some subset of N will also minimize the $\mathcal{V}$-information of a different subset of $N \in \mathrm{Dec}(X, y)$.

### E.5 $y$ Decomposition of X Through Base Expansion

In the main paper and Appx. E.4, we discussed how estimate to estimate $I_\mathcal{V}[Z \to \mathrm{Dec}(X, \mathcal{Y})]$ by uniformly sampling $N \in \mathrm{Dec}(X, y)$. As previously mentioned, many $N \in \mathrm{Dec}(X, y)$ will actually be redundant and have the same $\mathcal{V}$-information. It thus makes sense to only using Ns which are (approximately) mutually independent so as to minimize redundancies. We do so by assigning to each $\mathcal{X}_y$ a certain index and then computing the base $|\mathcal{Y}|$ expansion of that index. For example, in the case of binary cat-dog classification, we would assign some index to all cats and have N be the binary expansion of that index. Using base $|\mathcal{Y}|$ indexing gives a set $\{\mathrm{N}_i\}_i$ of $\lceil \log_{|\mathcal{Y}|} |\mathcal{X}| - 2 \rceil$ elements, which ensures that (i) each of the N is a deterministic function from $\mathcal{X}_y \to \mathcal{Y}$ and thus part of $\mathrm{Dec}(X, y)$; (ii) each of the N are (approximately) uncorrelated and thus will not be redundant. The algorithm to compute the set of N from which we estimate $I_\mathcal{V}[Z \to \mathrm{Dec}(X, \mathcal{Y})]$ is described in Algorithm 1.

Figure 10 shows the effect of using the $y$ decomposition of X through base $|\mathcal{Y}|$ expansion, instead of randomly sampling labels $N \in \mathrm{Dec}(X, y)$. We see that although differences are not large, the base expansion is better. At the optimal $\beta = 10$, using base $|\mathcal{Y}|$ expansion gives a test log likelihood of $-1.41 \pm 0.05$ vs. $-1.66 \pm 0.09$. Note that the base $|\mathcal{Y}|$ expansion does not incur any additional computational costs nor does it have any other drawbacks that we know about.

(a) DIB for Batch Training

(b) Generalization Gap

Figure 11: (a) Schematic illustration of using the predictor / $\mathcal{V}$-minimality head for a set $\{\mathrm{Dec}(X, y)\}_{y \in \mathcal{Y}}$ which is more amenable to batch training than the standard way of one predictor / $\mathcal{V}$-minimality head for each $\mathrm{Dec}(X, y)$ shown in Fig. 2b. (b) Effect on Alice's log likelihood when sharing the predictors (labeled "DIB Shared Pred.") compared to no sharing (labeled "DIB.").

## E.6  Sharing Predictors of $\{\mathrm{Dec}(X, y)\}_{y \in \mathcal{Y}}$ for Batch Training

In Eq. (7) we see that every example has to be treated differently depending on its underlying label $y \in \mathcal{Y}$. Indeed, $\mathrm{Dec}(X, y)$ depends on the underlying label $y$. In practice this means having a "for loop" over $y \in \mathcal{Y}$ (see Fig. 2) and using a different $\mathcal{V}$-minimality head for each $\mathrm{Dec}(X, y)$. This makes DIB hard to take advantage of the standard batch GPU training, where all examples in a batch are assumed to go through the same predictor regardless of their underlying label. Here we investigate whether DIB can be modified to take advantage of batch training by having a *single* predictor for a set of nuisance r.v. $\{\mathrm{Dec}(X, y)\}_{y \in \mathcal{Y}}$ as seen in Fig. 11a, i.e. treating all representations $z \sim Z$ the same way instead of having to distinguish them based on their underlying label $z \sim Z_y$. This has the same under underlying computational complexity, but it has the advantage of being trainable in batches. Interestingly, Fig. 11b shows that sharing the predicors ("DIB shared Pred.") reaches a better test performance in practice. This is probably an artefact of the values $\beta$ we are sweeping over, but it nevertless shows that one can perform well by sharing the predictors and this take advantage of batch training.

## E.7  Searching for an ERM That Does Not Generalize

In Sec. 4.2 we briefly outlined a method to test Theorem 1, which states that *all* ERMs should generalize well when trained from $\mathcal{V}$-minimal $\mathcal{V}$-sufficient representations. In other words, no ERM should have a non-zero generalization gap. Since we can only approximate $\mathcal{V}$-minimal $\mathcal{V}$-sufficient representations, our aim is to show that no ERM predicting from such a representation will incur a large generalization gap. Of course, it is infeasible to train all possible ERMs and then check that each generalizes well. So instead we directly search for the ERM with the largest generalization gap (worst case). We expect from Theorem 1 that even this ERM will have a small gap. Specifically, we want to maximize the test loss under the empirical risk minimization constraint:

$$\arg\max_{f \in V} \quad \mathrm{R}(f, Z)$$
$$\text{s.t.} \quad \hat{\mathrm{R}}(f, Z; \mathcal{D}) = \min_{f \in V} \hat{\mathrm{R}}(f, Z; \mathcal{D}) \tag{19}$$

Using a Lagrangian relaxation of Eq. (19) and flipping the sign, our objective is then:

$$\arg\min_{f \in V} \hat{\mathrm{R}}(f, Z; \mathcal{D}) - \gamma \mathrm{R}(f, Z) \tag{20}$$

Figure 12: Sweeping over $\gamma$ to find a poorly generalizing ERM. As $\gamma$ increases, the test performance decreases without having much effect on the training performance, until approximately $\gamma = 0.1$. In these experiments, Bob learns representations with either joint ERM (labeled ERM) or DIB. Alice then trains a decoder from Bob's representation using Eq. 20. Left plot shows results on CIFAR100, right plot is for our CIFAR10+MNIST dataset.

We thus minimize the training loss as usual while maximizing the test loss times a factor $\gamma$. This can easily be optimized by training on training *and* test examples, but multiplying all the gradients of test examples by $-\gamma$. Note that this is the same loss used by [107].

In order to find an $f$ that is a poorly generalizing ERM, we sweep over values of $\gamma$ and select the largest such that $f$ is (approximately) an ERM. Figure 12 shows that $\gamma = 0.1$ seems to be a good value for both datasets, to ensure that $f$ is approximately an ERM but performs as poorly as possible on test. We thus use this value for all "worst case" experiments in the paper.

### E.8 Optimality of $\mathcal{V}$-Sufficiency in Various Settings

In Fig. 3b we have provided experimental evidence for the optimality of $\mathcal{V}$-sufficient representations for the considered setting (CIFAR100, 8-dimensional representations, ResNet18). Here we show that similar conclusions hold for CIFAR100 and SVHN, with 2- or 8-dimensional representations, and with ResNet18 or $\mathcal{X}$-1024-1024-$\mathcal{Z}$ MLP encoders.

Fig. 13 summarizes all the results under various settings. Similarly to Fig. 3b we see that for most $\mathcal{V}_{Alice}$ the empirical optimal representation is recovered by maximizing $I_{\mathcal{V}}[Z \to Y]$. The 3 exceptions (e.g. SVHN, MLP, 2 dimenional representation, width 2 $\mathcal{V}_{Alice}$) out of the 40 possible $\mathcal{V}_{Alice}$ in each setting are likely due to optimization issues. Notice that the results for SVHN with a ResNet18 encoder show that when the width of $\mathcal{V}_{Alice}$ and $\mathcal{V}_{Bob}$ are both larger, performance becomes less dependent on $\mathcal{V}_{Bob}$. We investigate this phenomenon in the following subsection Appx. E.9.

### E.9 The Surprising Effect of Large Neural Families Trained with SGD

Figure 14 shows the same results as Fig. 3 for much larger widths of $[4, 16, 64, 256, 1024]$ instead of $[1, 2, 4, 8, 16]$. Figure 14b shows that $\mathcal{V}_{Bob} = \mathcal{V}_{Alice}$ is still optimal but the difference for using $\mathcal{V}_{Alice}$ larger than $\mathcal{V}_{Bob}$ is much less pronounced and mostly disappears at the largest widths. For example, the difference in performance when $\mathcal{V}_{Bob}$ has width 4 and $\mathcal{V}_{Alice}$ has width 16 is much larger than the difference in performance when $\mathcal{V}_{Bob}$ has width 256 and $\mathcal{V}_{Alice}$ has width 1024.

This seems to imply that the larger the functional families, the more similar they become. However, Figure 14a suggests a simpler explanation. Notice that that when $\mathcal{V}_{Bob}$ is very large, the representation that is learned is more linearly decodable. Recall that from Fig. 6 larger functional families are indeed more powerful, however it seems that once networks are wide enough, SGD favors the learning of classifiers that are very simple, and thus the functional family does not need to be matched. This notion echoes the fact that SGD is known to learn simple classifiers first [108].

(a) MLP

(b) ResNet18

Figure 13: Optimality of $\mathcal{V}$-sufficiency for different hyperparameters. Both plots show the comparative training performance of $\mathcal{V}_{Bob}$-sufficient representations for classifiers in $\mathcal{V}_{Alice}$. As in the main text, the log likelihood is scaled to lie in the range $[0, \ldots, 100]$ for each column. The predictive families $\mathcal{V}_*$ are single MLPs with varying width. (a) shows $\mathcal{X}$-1024-1024-$\mathcal{Z}$ MLP encoders, where the left and right columns use 2 and 8 dimensional Z, respectively, and the rows are CIFAR100 and SVHN. (b) is the same as in (a) but with a ResNet18 encoder.

(a) 2D Visualization

(b) Scaling Up

Figure 14: Optimality of $\mathcal{V}$-sufficiency in larger functional families. Plots are the same as in Fig. 3 but for widths $[4, 16, 64, 256, 2014]$ instead of $[1, 2, 4, 8, 16]$. Notably, the representations learned by Bob become nearly linearly decodable for the largest $\mathcal{V}_{Bob}$.

We emphasize that this behavior does not contradict our theoretical prediction that the optimal setting for Bob is to set $\mathcal{V}_{Bob} = \mathcal{V}_{Alice}$. Instead, it simply illustrates that if both Alice and Bob use large neural networks with standard initialization and train with SGD, the difference in the full expressivity of the models is not accessed. It is still provably better to use the same family, and we never observe performance to be less than optimal when the families are chosen to match.

### E.10  Effect of $\beta$ on Different $*$-Minimality Terms

In Fig. 4a and Fig. 4b, we showed the effect of varying the $\beta$ of Bob's DIB objective on Alice's worst-case performance. We also plotted the estimated value of the individual $\mathcal{V}$-sufficiency and $\mathcal{V}$-minimality terms. Figure 15 shows the same plot but for different $*$-minimality terms in Bobs objective, in particular: $\mathcal{V}$-minimality (single-layer MLP with 128 hidden units), $\mathcal{V}^+$-minimality (single-layer MLP with 8192 hidden units), $\mathcal{V}^-$-minimality (single-layer MLP with 2 hidden units), and variational minimality (using VIB's bound [19]). For each objective, the best (over $\beta$) test performance is transcribed in Table 1.

Figure 15a shows that $\mathcal{V}^+$-minimality gives rise to an objective that may be more robust to the choice of $\beta$ but also exhibits higher variance (in line with the estimation bounds from [23]). Figure 15b shows that the representation can be $\mathcal{V}^-$-minimal (that is, $I_{\mathcal{V}^-}[Z \to \mathrm{Dec}(X, \mathcal{Y})]$ is close to zero) but have little effect on the $\mathcal{V}$-sufficiency term (large $I_{\mathcal{V}}[Z \to Y]$).

### E.11  $\mathcal{V}$-Minimality as a Regularizer

In the main text, we have seen that minimizing $I_{\mathcal{V}}[Z \to \mathrm{Dec}(X, \mathcal{Y})]$ is theoretically optimal (Theorem 1) and empirically outperforms other regularizers in our two stage setting (Table 1). It is thus natural to ask whether $I_{\mathcal{V}}[Z \to \mathrm{Dec}(X, \mathcal{Y})]$ can also perform well as a regularizer in a standard neural network setting.

Table 3: Evaluation of regularizers for permutation invariant classification (test accuracy).

|  | No Reg. | Stoch. Rep. | Dropout | Wt. Dec. | VIB | DIB |
|---|---|---|---|---|---|---|
| MNIST | $98.29 \pm {.05}$ | $98.33 \pm {.04}$ | $98.68 \pm {.04}$ | $98.49 \pm {.04}$ | $98.63 \pm {.04}$ | $\mathbf{98.69} \pm {.03}$ |
| CIFAR10MNIST | $46.49 \pm {.07}$ | $47.23 \pm {.13}$ | $\mathbf{48.86} \pm {.17}$ | $44.86 \pm {.06}$ | $46.38 \pm {.01}$ | $48.07 \pm {.10}$ |

(a) Generalization Gap

(b) Terms in DIB

(c) Terms in VIB

Figure 15: Effect of $\beta$ on the worst case generalization gap, as well as the terms estimated by DIB (using different $\mathcal{V}$s for the minimality term) and VIB. (a) Train log likelihood and worst case test log likelihood of the different predictors; (b) Estimated $I_\mathcal{V}[Z \to Y]$ and $I_\mathcal{V}[Z \to \mathrm{Dec}(X, \mathcal{Y})]$, $I_{\mathcal{V}^+}[Z \to \mathrm{Dec}(X, \mathcal{Y})]$, and $I_{\mathcal{V}^-}[Z \to \mathrm{Dec}(X, \mathcal{Y})]$; (c) Estimated $I_\mathcal{V}[Z \to Y]$ and (a variational estimate of) $I[Z; X]$ for VIB [19]. Similarly to Fig. 4 all experiments ran on cifar10 and the results show the average over 5 runs as well as $95\%$ bootstrap confidence interval.

We investigate this question in the same setting as Alemi et al. [19], where the neural network is an MLP, thus treating the pixels as permutation invariant. We use the same hyperparameters as [19]: 1e-4 learning rate with exponential decay of factor 0.984, Adam optimizer, 200 epochs, trained on the train and validation set, batch size 100, 256 dimensions for Z, $\mathcal{X} - 1024 - 1024 - \mathcal{Z}$ MLP encoder, logistic regression classifier $\mathcal{V}$. The only known difference being that we do not use exponential moving average (we did not test with it). We jointly train Bob and Alice ("1 Player, Avg ERM" in Appx. D.1).

We evaluate the model on MNIST (as done in Alemi et al. [19]) as well as on our CIFAR10+MNIST dataset. Table 3 compares the test accuracy of DIB with the same regularizers as in Table 1, the only difference being that all the regularizers are applied both on Bob and Alice (as it is now a single network trained jointly). For dropout the rate is droping rate 50%, for weight decay it uses a factor of 1e-4, VIB uses $\beta = 1e - 3$, DIB uses $\beta = 0.1$. We see that DIB performs best along with Dropout on MNIST, and performs second best after dropout on CIFAR10MNIST.

Although DIB performs well, it does not stand out as much as in the other settings we investigated in the main paper. We suggest a few potential explanations: (i) As is standard practice, we evaluate on accuracy, while our theory only speaks to log likelihood performance (although recall that Table 2 does show a strong correlation with accuracy generalization gap); (ii) With standard training methods (large learning rate and avg ERM), neural networks generalize relatively well without the need for regularizers, as seen by the strong performance of the stochastic representation baseline in Table 3. (iii) Our representations work well for a downstream ERM, they only regularize the model by the representation. In the single player game setting, other methods (such as dropout) regularize both the representation and the downstream classifier, which is not discussed in our theory.

### E.12 Additional Correlation Experiments

In this section, we expand on the results in Sec. 4.3 in the main paper and show how our approach compares to our implementation of sharpness (the best generalization measure from [50]) in a setting with heterogeneous model and dataset choices.

We hypothesize that $\mathcal{V}$-minimality should be a fairly model- and dataset- agnostic measure of generalization. Indeed, $I_\mathcal{V}[Z \to \mathrm{Dec}(X, \mathcal{Y})]$ has the advantage of being a measure in $[0, \log |\mathcal{Y}|]$, which is 0 when $\mathcal{V}$-minimal (Prop. 2) and seems to be monotonically decreasing with the generalization capacity of a model (Table 2).

In order to study this hypothesis we sweep over different hyper-parameters across two datasets (CIFAR-10 and SVHN) and two models (ResNet18 and a $\mathcal{X}$-2048-2048-2048-$\mathcal{Z}$ MLP) each followed by a $\mathcal{Z}$-128-128-$\mathcal{Y}$ MLP, and 5 seeds. The difference with the experiments in the main paper is that we do not run all possible combination of hyperparemters (computationally prohibitive as we already have 2*2*5=20 models for each seed,data,architecture) but rather sweep over one hyperparameter at the time and compute then average rank correlation. Here are the hyperparameters we sweep over: (i) learning rates (1e-3,1e-4,1e-5,1e-6); (ii) weight decay (1e-6,1e-5,1e-4,1e-3,1e-2,0.1); (iii) dropout (0.,0.1,0.2,0.3,0.4,0.5,0.6); (iv) Z dimensionality (8,32,128,512,2048). We additionally train a set of models using VIB with different $\beta$ values (100,10,1,1e-1,1e-2,1e-3). This gives a total of $5 * 2 * 2 * (4 + 6 + 7 + 5 + 6) = 560$ models, from which we only keep models that reach a training loss of 0.01. Similar to the main experiment in the paper, we compute the correlation between the probes and the observed generalization gap, with the difference being that the correlation is now computed across experiments with both the datasets as well as models while varying only one hyperparameter at a time.

Table 4: Evaluation of our probe and sharp mag. in settings with different datasets and architectures

|  |  | W. Dec | $|\mathcal{Z}|$ | VIB | Lr | Dropout |
|---|---|---|---|---|---|---|
| $\tau_{loglike.}$ | $\mathcal{V}$ | **0.67** | **0.50** | **0.45** | **0.62** | **0.09** |
|  | Sharp Mag. | 0.05 | $-0.27$ | $-0.16$ | 0.09 | $-0.36$ |
| $\tau_{acc.}$ | $\mathcal{V}$ | **0.63** | **0.52** | **0.49** | **0.56** | **0.06** |
|  | Sharp Mag. | $-0.03$ | $-0.09$ | $-0.03$ | 0.29 | $-0.19$ |

Table 4 shows the performance of $I_\mathcal{V}[Z \to \mathrm{Dec}(X, \mathcal{Y})]$ probe compared to sharpness (the best performing baseline in Table 2). We observe that our approach is significantly better correlated than sharpness, both in terms of generalization in terms of accuracy as well as log-likelihood. This seems to support that $\mathcal{V}$-minimality gracefully handles different datasets and model architectures, providing reliable estimates of generalization across the spectrum. In contrast, we see that the sharpness magnitude cannot be used to predict well generalization when sweeping over datasets and architectures. This suggests that the intuitive idea behind $I_\mathcal{V}[Z \to \mathrm{Dec}(X, \mathcal{Y})]$ (considering how easy it is to decode the training examples from a representation using the correct functional family) is useful and robust to predict generalization.

## Footnotes

[10] If both the labeling and predictors had been deterministic, the proof would be very simple and would go as follows: assume $f$ is not optimal on test performance, show that the labels $\hat{Y}$ predicted by $f$ and are in $\mathrm{Dec}(X, y)$, conclude that Z does not minimize $\mathrm{Dec}(X, y)$ as it perfectly predicts $\hat{Y}$ by construction.

[11] Indeed, by construction $f'$ predicts perfectly the training examples $\tilde{\mathrm{N}} = 1$, because $\hat{\mathrm{R}}_y(f', \mathrm{Z}, \mathcal{D}) = 0$ for $y = 1$. Unfortunately, its risk when predicting $\tilde{\mathrm{N}}$ is not always smaller than $\mathrm{H}\left[\tilde{\mathrm{N}}\right]$. For example, $f'$ would incur infinite loss if some test examples $x \in \mathcal{X}_y^e$ was encoded to some some $\mathcal{Z}_y^c$ because $f'$ would predict that it comes from the training set with probability of 1.

[12]Up to terms that are constant in $P_{\mathrm{Z}|\mathrm{X}}$. We can provide similar guarantees when incorporating these constants due to Lemma 4 of Xu et al.'s [23] but there is no reason to estimate these constants in our framework.

[13]Notice that we use a small learning rate, a large number of epochs, and checkpoint based on training loss because we are interested in studying the generalization ability of a model depending solely on the criterion being optimized over.

[14]Contrary to VIB, DIB does not require the use of an encoder that parameterizes a Gaussian distribution. We use a Gaussian to make it more comparable to VIB.

[15] We initially only wanted to consider generalization in terms of log-likelihood since our theory only talks about log-likelihood. For this reason, the correlation of sharpness magnitude in Table 2 is lower than in Jiang Jiang et al. [50], which is why we also transcribe their results.

[16] We also tried with a width of 32 but the differences due to depth was surprisingly less pronounced there.

[17] To implement that, we start with a usual MLP and then prune recursively 50% of the weights. The recursiveness ensures that every weight which were previously pruned will also be pruned in the next round.

[18] This is likely because the sweeping interval $[1, \ldots, 5]$ is quite small. When using a larger sweeping interval, the optimization was harder, often yielding smaller randomized log likelihood.

[19]We use the notation $\frac{\partial}{\partial Z} \mathcal{L}_{\mathrm{DIB}}$ to denote $\frac{\partial}{\partial \theta} \mathcal{L}_{\mathrm{DIB}}$ where $Z = \mathrm{encoder}_{\theta}(X, \epsilon)$, $\epsilon \sim \mathcal{U}(0, 1)$. In other words the encoder will take some noise $\epsilon$ and some input $x$ and will output a representation $z$, we are interested in the parameters of that encoder.