[Reviews · NeurIPS 2020]

Review 1

Summary and Contributions: The authors build upon a new information measure recently introduced and replace the mutual information with the latter in the original deep variational bottleneck formulation to form a new objective. In this respect, they adapt the notion of minimality and sufficiency to this measure, and prove, under several assumptions, that optimal representations (sufficient + minimal) induce the optimal test loss when used with downstream ERM.

Strengths: - The idea is well motivated and interesting. - The objective function is thoroughly analysed, in particular a result on generalization is derived from the V-sufficiency and V-minimality brought by the minimization of the objective - A series of controlled experiments are carried out to assess the theoretical statements of the paper and provide more insights on the method

Weaknesses: - Existence : In Theorem 1, you show that a sufficient condition for optimality of ERM is Z's V-minimality and V-sufficiency. Can we prove such representation exists ? - Lack of details : I acknowledge that 8 pages is short, especially for papers that ambition to derive some theory, but there are too many things that are omitted and impede reader's comprehension. For instance, I found part 3.2 unclear while this part seems crucial to the paper. For instance, what is N ? In which space does it live ? In practice, how do you compute the quantity Dec(X,y) ? Even after reading this part twice, I'm not sure I can wrap my head around it. - The optimization part is completely deferred to appendix but the introduction of min-max problems doesn't seem like a detail to me. After skimming through 26 pages of supplementary material, I was able to find that you need to use 5 unrolling steps. Does the method induce additional complexity introduced over the traditional IB, can you quantify it ? - Consistency between theory and experiments : If I'm correct, your optimal test loss (according to proposition 1.) should occur where Z is V-sufficient. As you mention it, the optimal loss on Fig. 3 occurs when beta=10, which the informativeness I_V(Z -> Y) is way below its maximum value. Am I missing some assumption ? - Usefulness of the method : While I appreciate and acknowledge the importance of controlled experiments to verify the theory, I feel the paper is missing at least one concrete and perhaps more challenging application where you would clearly demonstrate the superiority of this method over the traditional IB (eg. accuracy on simple classification, or invariance to nuisance factors, or even robustness of the classifier).

Correctness: - Empirical methodology looks correct, but hard to properly assess without carefully looking at the appendix

Clarity: Overall, paper well written, punctually unclear.

Relation to Prior Work: Yes

Reproducibility: No

Additional Feedback: ---- Post rebuttal ----- I have read the authors's rebuttal and summarize main points here: - Existence and clarifications of the definition of Dec(X,y) : The authors have properly addressed my question. The theoretical part now appears clearer to me. - Consistency of results : Not fully addressed. By Theorem 1, the representation that should achieve the best worst-case test loss should be the one that is minimal among the ones that are sufficient. Looking at Fig 3(b), this should occur around beta=0.01, while the best worst-case test loss actually occurs at beta=10. Regardless, it seems the consistency of such result is hard to assess, as the beta value that arises as the best representation seems highly dependent upon the optimization process (cf Fig.10). - Computational complexity of the method : L.909: authors clearly mention using 5 unrolling steps is "significantly better" than joint optimization and L. 913-915: authors confirms all results are produced with 5 inner optimization steps. Therefore, I agree with the authors that joint optimization is possible, making the computation efficient, but is neither what is used throughout the paper, nor what is suitable according the authors' own words. - Comparison on nuisance factor experiments : I am pleased to see such experiments were carried out in Table 3. Appendix D.10, which I had missed in my initial reading. The gains appear very marginal as compared to VIB though. - A positive point that I noted afterwards (which I'm not sure is disccused in the papaer) is that DIB seems more flexible than the common VIB, as the latter typically requires a Gaussian encoder to make the variational bound on I(X;Z) easy to estimate, while VIB is not constrained on the type of stochastic encoder to be used. Summary : The authors have properly clarified the definition of Dec(X,y). The theoretical results of the paper (especially Thm 1.) are now clearer to me. Besides, authors have pointed out to experiments I asked. I'm still not fully convinced about the consistency between experiments and theory, or whether the method will properly scale up to larger datasets (e.g number of possible combinations N growing with both data size and nb of classes), or to larger networks (how would the method compare to VIB if joint optimization is performed rather than costly unrolling steps ?), but I feel the novelty/relevance of the theoretical insights provided by the paper justify a NeurIPS publication. I will therefore raise my score to 7.


Review 2

Summary and Contributions: The paper presents an alternative to the Information Bottleneck (IB) criterion in which the mutual information terms of the IB are replaced by a predictive-family dependent version, introduced in a recent paper. Intuitively this criterion seeks a representation that is “sufficient” for the function family to make the optimal prediction, while it is “minimal”, in the sense that it cannot be used to make good predictions of the input using the same predictive family. Experiments evaluate the effect of the expressiveness of the function family in achieving sufficiency and minimality and demonstrate the use of this criterion for probing generalization, among other things.

Strengths: Building on recent ideas, this paper suggests an elegant approach in defining and quantifying task-dependent representations. The paper is very well-written, and the proposal is concisely put into context with nice use of figures. Empirical evaluation tests various claims both in the main text and the large body of supporting material that is presented in the appendix.

Weaknesses: I think the main text could benefit from a short discussion on the practicality of DIB compared to IB. The comment is inspired by skimming through the procedure used to estimate and optimize the proposed measure in the appendix. (addressed by the rebuttal)

Correctness: Yes, to the extent that I followed and understood the details. Unfortunately, I did not properly read most sections of the supplementary material and did not verify the proofs.

Clarity: Generally, the paper is very well-written. One place where the paper can be more clear, maybe through an example is in explaining equation 4. (addressed by the rebuttal)

Relation to Prior Work: That section is rather brief, but the related work is discussed throughout the paper. However, I cannot tell if a recent related work is missing.

Reproducibility: Yes

Additional Feedback: I've read the rebuttal and other reviews. Overall, I think the contributions are valuable and the modifications suggested in the rebuttal will further clarify the presentations and the computational concerns which are also raised by other reviews.


Review 3

Summary and Contributions: This paper introduces decodable information bottleneck (DIB), which generalizes the Information Bottleneck (IB) principle while taking into account the predictive family of the decoder. Under a series of assumptions, the authors prove that a V-sufficient and V-minimal representation achieves the best possible test loss with ERM on the training set, formally linking generalization with V-sufficiency and V-minimality. Numerical experiments show that optimal representation is achieved when the representation learner chooses the same predictive family as the machine learner; V-minimality relate closely to generalization; and V-minimality seems to be a good probe for generalization.

Strengths: Soundness: The theoretical results are sound, and empirical evaluation is convincing and thorough. Significance and novelty: The core theoretical result, Theorem 1, is a strong and novel result, which shows the connection of V-sufficiency and V-minimality with generalization. In addition, numerical experiments show convincingly the relation between V-minimality and generalization under DIB, as well as DIB is a good metric for probing generalization in joint training scenarios. This bring about a novel perspective about generalization, as well as the connection between information minimality, generalization and predictive family. Relevance: The paper will be particularly interesting to the Information Bottleneck community, as well as statistical machine learning community, and may bring about new perspectives and directions to the communities.

Weaknesses: Although Theorem 1 is a strong result, it assumes many assumptions. Among which finite sample space, deterministic labeling and pairwise universality seems quite limiting as compared to generic machine learning settings. This limits the broader applicability of the theoretical result. ================================================== # After Authors' Response: I'm glad to see that the authors have relaxed certain assumption. Overall the paper is novel and solid, and I remain my initial rating.

Correctness: The proposed method is reasonable, and I do not detect errors in theoretical claims. The empirical evaluation is reasonable and supports the paper's claim.

Clarity: The paper is well motivated, the logic is clear. Since the paper is theory-heavy, it would be more amenable to readers if it can provide more intuition when deriving and explaining the theoretical results.

Relation to Prior Work: The paper has discussed its relation with previous work, including their differences.

Reproducibility: Yes

Additional Feedback: (1) Typo in Line 199: should be "argmin" (2) The appendix following the main text has many unfinished parts as well as dialogue between authors (e.g. line 571, 657, etc.). The authors should have removed the appendix in the main submission.


Review 4

Summary and Contributions: This submission deals with the question of how to learn “optimal” latent representations that generalise well in the context of Information Bottleneck (IB) models. The authors address known deficiencies of the IB objective function (generally complex decision boundaries; hard estimation of mutual information; no optimality guarantees for minimal sufficient representations) and hypothesise that these problems are related to the IB objective not taking the prediction method into account, or more precisely, the predictive family V (e.g. linear classifiers or decoder networks). The submission proposes the decodable Information Bottleneck (DIB), an alternative approach which utilises V-information and thereby includes the predictive family V in learning a latent representation. On the basis of the introduced notions of V-sufficiency and V-minimality, it is proved that optimal latent representations can be obtained and empirical validations are provided. The main contributions consist in: 1. Decodable Information Bottleneck: Definition of minimal sufficient latent representations which take the predictive family V into account and showing their optimality both theoretically as well as experimentally. 2. Providing evidence that V-minimality can be related to generalisation capability.

Strengths: # Soundness: - The submission provides a solid theoretical grounding for their proposed approach which is based on V-information. The notion of minimal sufficient statistic or representation in the original IB formulation, which relies on mutual information, are replaced by the corresponding notions of V-minimality and V-sufficiency in the context of V-information and an analogous objective function for the DIB is derived. - In particular, it is elaborated that the proposed approach allows learning latent representation such that any empirical risk minimiser in the predictive family V generalises well, or more precisely, achieves the best possible test performance. The justification of the proposed model appears sound and adequate empirical validation on the basis of CIFAR-10/-100 and MNIST experiments is provided. # Significance & Novelty, Relevance: - Connecting the Information Bottleneck to the V-information appears to be a fairly novel idea. Especially the introduced notions of V-minimality and V-sufficiency potentially constitute a significant advancement in thinking about IB approaches, as it seems that the corresponding terms can be estimated more efficiently in the context of V-information from finite samples than mutual information terms in the context of Shannon information. - As increasingly variants to VAEs and GANs employ information-theoretic approaches like the Information Bottleneck, the theoretical insights might prove relevant for other models and a wider audience.

Weaknesses: - The choice of the appropriate predictive family V appears to be the most vulnerable aspect in the proposed model, as an inadequate choice can have a detrimental effect on the optimality of the learned latent representation (as shown in figure 2). To me it is not completely clear how to best choose the predictive family V. Could the authors comment on that? ###################################################### # After Authors' Response: I appreciate the authors’ response and I believe that the submission provides novel and relevant insights into (deep) IB models and might be relevant to a wider audience which employs information theoretic approaches. Therefore, I stand with my initial rating.

Correctness: The submission provides theoretical as well as empirical justification of the claims. The methodology appears sound.

Clarity: - Generally, the text is well-written and has a clear structure. However, the derivation is not always easy to follow. Especially, the part on the decomposition of the input X (lines 134 – 141) might need a revision, which explains eq. 4 in more detail. - L. 181 and figure 2: What is meant by odd / even CIFAR100?

Relation to Prior Work: In my opinion, the relationship to related work is adequately addressed.

Reproducibility: Yes

Additional Feedback: # Minor Remarks: - Footnote 4: “following” -> “Following” (capital “F”) - L. 230: “aren’t” -> “are not”.

[Author Response · NeurIPS 2020]

We thank the reviewers for their insightful feedback. We are pleased that the reviewers found our paper well-written
[R1,R2,R4], thorough [R1,R3], and recognized the strength of our theoretical results [R3,R4]. We are especially
encouraged by the comments of R3 and R4 concerning the possible impact of our work beyond the information
bottleneck (IB). We first address two recurrent questions and proceed by addressing reviewer-specific comments.

**G.1. Clarifying the definition of** $\mathrm{Dec}(X, y)$ [R1,R2,R4]. We simplified the notation and added an example to better
illustrate this concept. We summarize these changes here. Suppose we are classifying cats versus dogs, then the $y$ (say,
"cat") decomposition of images X, corresponds to taking all "cats" and labeling them with every possible binary labels.
Formally, let $X_y$ be a r.v. s.t. $P_{X_y} = P_{X|y}$ (the "cat" r.v.). Then $\mathrm{Dec}(X, y) := \{N | \exists t' : \mathcal{X} \to \mathcal{Y} \ s.t. \ N = t'(X_y)\}$. We
emphasize that: (i) N is a "labeling" of $X_y$ so it takes value in $\mathcal{Y}$ ( {"cat","dog"}) so we can predict it using a binary
classifier in $\mathcal{V}$; (ii) the formal definition does not assume that the underlying labeling of Y is deterministic.

**G.2. Practicality of our Decodable IB (DIB)** [R1,R2]. We extended the discussion about practical optimization
(summary of Appx. D6, D7) and estimation (summary of Appx. D4) in the main paper. We emphasize that: (i) the
min-max problem can be well optimized using joint gradient descent ascent (c.f. Fig. 10 and L. 907), it does not require
unrolling steps; (ii) we get good performance by only predicting a few (4) "labels" $N \in \mathrm{Dec}(X, y)$ (c.f. Appx. D6). As
a result, training a network with DIB is approximately as computationally efficient as training a standard network or the
variational IB — and much more efficient than IB. We will **release our code** to help practitioners use DIB.

**R.1.** We would like to thank you for the helpful review and useful comments which we adressed in the revised
manuscript. We are pleased that you found the idea interesting, well-motivated, and the analysis thorough.

• "*Can we prove such representation exists?*": **Yes**, they always exist because they are defined as maximizers/minimizers
of some quantity in a finite set. This is a very good point, which we now discuss in the paper.
• "*[...] I found part 3.2 unclear [...]*": Please refer to G.1. for $\mathrm{Dec}(X, y)$ and N. We also added an algorithm on the
optimization of $I_\mathcal{V}[Z \to \mathrm{Dec}(X, Y)]$ . For cat-dog classification: (i) assign each cat with a binary label; (ii) optimize
the encoder such that classifiers in $\mathcal{V}$ *cannot* predict these labels, i.e., distinguish between cats; (iii) repeat for dogs.
• "*[...] Does the method induce additional complexity [...]?*": **No**, please refer to G.2.
• "*Consistency between [...] proposition 1. [...] Fig. 3 [...]*": We would like to clarify that Prop. 1 only concerns the
*existence* of an optimal predictor. The generalization of *any* ERMs requires $\mathcal{V}$-minimality (Theorem 1). The best loss
in Fig. 3a occurs at approximate $\mathcal{V}$-minimality (Fig. 3b) suggesting that ensuring generalization is more valuable
than improving the best possible performance. We thus do not believe that Fig. 3 contradicts the theory.
• "*Usefulness of the method: [...] traditional IB [...]?*": Due to space constraints, we decided to highlight results
that validate the theory, and thank the reviewer for acknowledging the importance of such results. We nevertheless
emphasize that many of the results asked by the reviewer are in the appendices. Table 3 (Appx. D10) evaluates DIB
as a regularizer on CIFAR10MNIST (CIFAR10 with overlaid MNIST images to evaluate robustness to "nuisances")
and MNIST (to replicate the results from VIB's paper). We see that DIB outperforms VIB on both tasks, i.e., it gives
rise to a better classifier and is more robust to nuisances. We added a small discussion about this in the main text.

**R.2.** We thank you for the kind and helpful review, we are pleased that you found the paper well-written and the
approach elegant. We have put significant effort into incorporating both your suggestions into the revised manuscript.

• "*[...] the main text could benefit [...] practicality of DIB compared to IB [...]*": Please refer to G.2.
• "*[...] more clear, maybe through an example, is in explaining equation 4. [...]*": Please refer to G.1.

**R.3.** We thank the reviewer for reading the paper thoroughly and for providing a kind and detailed assessment. We are
encouraged that you found the idea novel, the theoretical results strong, and the empirical evaluation thorough.

• "*Although Theorem 1 is a strong result, it assumes [...]*": We believe that our assumptions **do not significantly limit**
**the applicability of our theory**: (i) Piece-wise universality: we now **relaxed this assumption** so that the proof
holds for any model using a softmax layer with biases on that layer; (ii) Finite sample spaces: although the continuous
setting is theoretically interesting, it (arguably) is less important in practice as everything must be discretized on a
computer; (iii) Deterministic labeling: examples are usually seen once per dataset, i.e., no two same images are given
different labels. In such cases the labeling is deterministic. We acknowledge that this is not necessarily true in the
real world and hope to extend our proof in future work. We emphasized this assumption in the revised paper.

**R.4.** We thank you for your helpful review, and are pleased you think the approach could constitute a significant
advancement for IB and be broadly relevant for methods based on information theory (e.g. variants of VAE and GAN).

• "*The choice of the appropriate predictive family $\mathcal{V}$ [...]?*": There is a trade-off. A larger $\mathcal{V}$ means a smaller best
achievable loss, but a more complex $\mathcal{V}$ makes the estimation of $\mathcal{L}_{\mathrm{DIB}}$ provably harder. This is a very good point,
which we now discuss in the revised paper.
• "*[...] the decomposition of the input X [...] might need a revision*": Please refer to G.1.
• "*What is [...] odd/even CIFAR100?*": Classifying the parity of the class index. This "CIFAR2" is easier to visualize.

[Meta-Review · NeurIPS 2020]

This paper aims at finding optimal representations for supervised learning from the perspective of information bottleneck (IB), and proposes an extension of IB, decodable information bottleneck (DIB), which considers the predictive family into account when learning representations. The original IB tries to learn representations that have sufficient information about the label but minimal information about the input features. The authors argue that DIB produces optimal representations, in the sense of achieving optimal expected test performance, and in addition, that DIB is easier to estimate than IB. Extensive experimental results are presented to verify the theoretical claims. According to the reviewers, the idea proposed in the paper is well motivated and interesting, and the empirical evaluation is convincing and thorough. All reviewers also agree that the authors properly addressed the concerns raised in the original reviews.